# TDP-43 loss and ALS-risk SNPs drive mis-splicing and depletion of UNC13A

Anna-Leigh Brown[1,91], Oscar G. Wilkins[1,2,91], Matthew J. Keuss[1,91], Sarah E. Kargbo-Hill[3,91], Matteo Zanovello[1], Weaverly Colleen Lee[1], Alexander Bampton[4,5], Flora C. Y. Lee[1,2], Laura Masino[2], Yue A. Qi[6], Sam Bryce-Smith[1], Ariana Gatt[4,5], Martina Hallegger[1,2], Delphine Fagegaltier[7], Hemali Phatnani[7], NYGC ALS Consortium*, Jia Newcombe[8], Emil K. Gustavsson[4,9], Sahba Seddighi[3,10], Joel F. Reyes[3], Steven L. Coon[11], Daniel Ramos[3,6], Giampietro Schiavo[1,12], Elizabeth M. C. Fisher[1], Towfique Raj[13,14,15,16], Maria Secrier[17], Tammaryn Lashley[4,5], Jernej Ule[1,2,18], Emanuele Buratti[19], Jack Humphrey[13,14,15,16], Michael E. Ward[3✉] & Pietro Fratta[1✉]

Variants of *UNC13A*, a critical gene for synapse function, increase the risk of amyotrophic lateral sclerosis and frontotemporal dementia[1–3], two related neurodegenerative diseases defined by mislocalization of the RNA-binding protein TDP-43[4,5]. Here we show that TDP-43 depletion induces robust inclusion of a cryptic exon in *UNC13A*, resulting in nonsense-mediated decay and loss of UNC13A protein. Two common intronic *UNC13A* polymorphisms strongly associated with amyotrophic lateral sclerosis and frontotemporal dementia risk overlap with TDP-43 binding sites. These polymorphisms potentiate cryptic exon inclusion, both in cultured cells and in brains and spinal cords from patients with these conditions. Our findings, which demonstrate a genetic link between loss of nuclear TDP-43 function and disease, reveal the mechanism by which *UNC13A* variants exacerbate the effects of decreased TDP-43 function. They further provide a promising therapeutic target for TDP-43 proteinopathies.

Amyotrophic lateral sclerosis (ALS) and frontotemporal dementia (FTD) are devastating adult-onset neurodegenerative disorders with shared genetic causes and common pathological aggregates[6]. Genome-wide association studies (GWAS) have repeatedly demonstrated a shared risk locus for ALS and FTD in the crucial synaptic gene *UNC13A*, although the mechanism underlying this association has remained unknown[1–3].

ALS and FTD are pathologically defined by cytoplasmic aggregation and nuclear depletion of TAR DNA-binding protein 43 (TDP-43) in more than 97% of ALS cases and 45% of FTD cases[4,5] (frontotemporal lobar degeneration (FTLD) due to TDP-43 proteinopathy (FTLD-TDP)). TDP-43 is an RNA-binding protein (RBP) that resides primarily in the nucleus and has key regulatory roles in RNA metabolism, including as a splicing repressor. Upon loss of nuclear TDP-43—an early pathological feature in TDP-43-associated ALS (ALS-TDP) and FTLD-TDP—non-conserved intronic sequences are de-repressed and erroneously included in

mature RNAs. These events are referred to as cryptic exons (CEs) and often lead to premature stop codons and transcript degradation, or premature polyadenylation[7]. One such CE occurs in the stathmin 2 (*STMN2*) transcript[8,9]. This *STMN2* CE is selectively expressed in affected tissue, and its level correlates with TDP-43 phosphorylation, enabling it to serve as a functional readout for TDP-43 proteinopathy[8–10]. However, a link between CEs and disease risk has not yet been established.

Here we report the presence of a CE in *UNC13A*, which is present at high levels in neurons from patients with ALS and FTLD-TDP. This CE promotes nonsense-mediated decay (NMD) and *UNC13A* transcript and protein loss. Notably, intronic risk-associated single nucleotide polymorphisms (SNPs) for ALS and FTD in *UNC13A* promote increased inclusion of this CE. Collectively, our findings reveal the molecular mechanism behind one of the top GWAS hits for ALS and FTD and provide a promising new therapeutic target for TDP-43 proteinopathies.

[1]UCL Queen Square Motor Neuron Disease Centre, Department of Neuromuscular Diseases, UCL Queen Square Institute of Neurology, UCL, London, UK. [2]The Francis Crick Institute, London, UK. [3]National Institute of Neurological Disorders and Stroke, NIH, Bethesda, MD, USA. [4]Queen Square Brain Bank, UCL Queen Square Institute of Neurology, University College London, London, UK. [5]Department of Neurodegenerative Disease, UCL Queen Square Institute of Neurology, University College London, London, UK. [6]Center for Alzheimer's and Related Dementias, National Institutes of Health, Bethesda, MD, USA. [7]Center for Genomics of Neurodegenerative Disease, New York Genome Center (NYGC), New York, NY, USA. [8]NeuroResource, Department of Neuroinflammation, UCL Queen Square Institute of Neurology, London, UK. [9]Great Ormond Street Institute of Child Health, Genetics and Genomic Medicine, University College London, London, UK. [10]Medical Scientist Training Program, Johns Hopkins University School of Medicine, Baltimore, MD, USA. [11]Molecular Genomics Core, Eunice Kennedy Shriver National Institute of Child Health and Human Development, NIH, Bethesda, MD, USA. [12]UK Dementia Research Institute, University College London, London, UK. [13]Nash Family Department of Neuroscience and Friedman Brain Institute, Icahn School of Medicine at Mount Sinai, New York, NY, USA. [14]Ronald M. Loeb Center for Alzheimer's Disease, Icahn School of Medicine at Mount Sinai, New York, NY, USA. [15]Department of Genetics and Genomic Sciences and Icahn Institute for Data Science and Genomic Technology, Icahn School of Medicine at Mount Sinai, New York, NY, USA. [16]Estelle and Daniel Maggin Department of Neurology, Icahn School of Medicine at Mount Sinai, New York, NY, USA. [17]Department of Genetics, Evolution and Environment, UCL Genetics Institute, University College London, London, UK. [18]Department of Molecular Biology and Nanobiotechnology, National Institute of Chemistry, Ljubljana, Slovenia. [19]Molecular Pathology Lab, International Centre for Genetic Engineering and Biotechnology (ICGEB), Trieste, Italy. [91]These authors contributed equally: Anna-Leigh Brown, Oscar G. Wilkins, Matthew J. Keuss, Sarah E. Kargbo-Hill. *A list of authors and their affiliations appears online. ✉e-mail: wardme@nih.gov; p.fratta@ucl.ac.uk

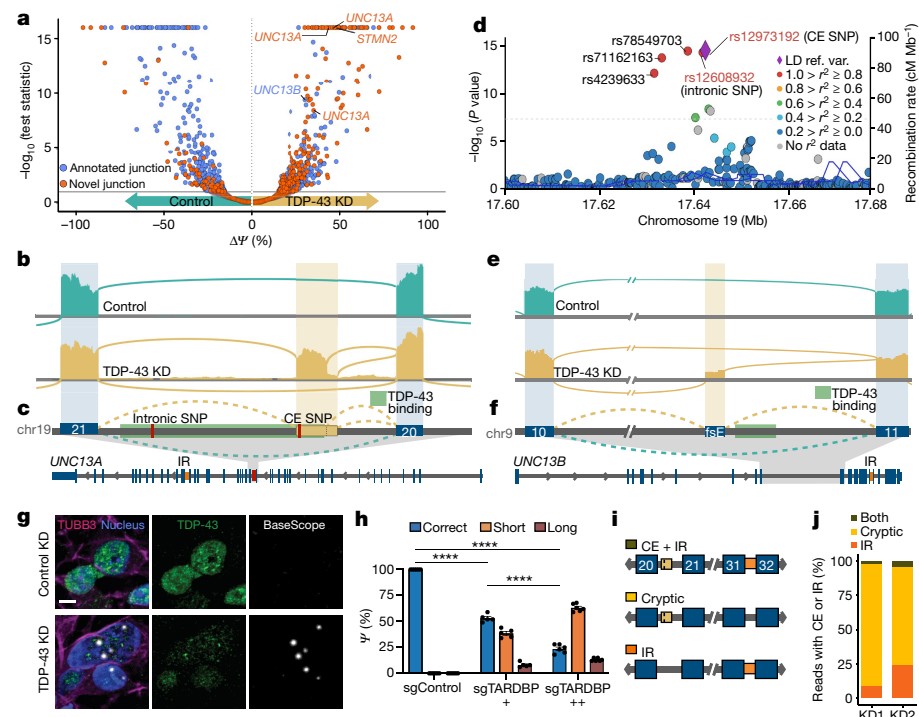

**Fig. 1 | TDP-43 depletion in neurons leads to altered splicing in synaptic genes *UNC13A* and *UNC13B*. a**, Differential splicing analysis by MAJIQ[33] in control (*n* = 4) and CRISPRi TDP-43 depleted (KD) (*n* = 3) iPS cell-derived cortical-like i[3]Neurons. Each point denotes a splice junction. **b**, Representative sashimi plots showing cryptic exon (CE) inclusion between exons 20 and 21 of *UNC13A* upon TDP-43 knockdown. **c**, **f**, Schematics showing intron retention (IR) (orange; bottom), TDP-43 binding region[22] (green), and two ALS- and FTLD-associated SNPs (red) in UNC13A (**c**) and UNC13B (**f**). **d**, LocusZoom plot of the *UNC13A* locus in the most recent ALS GWAS[15]; the dashed line indicates the risk threshold used in that study. Lead SNP *rs12973192* is plotted as a purple diamond, other SNPs are coloured by linkage disequilibrium (LD) with *rs12973192* in European individuals from 1000 Genomes. Ref. var., reference variant. **e**, Representative sashimi plot of *UNC13B* showing inclusion of the FSE upon TDP-43 knockdown. **g**, BaseScope detection of *UNC13A* CE (white puncta) in control (top) and TDP-43-knockdown (bottom) i[3]Neurons co-stained for TDP-43 (green), neuronal processes (stained for TUBB3, pink) and nuclei (blue). Scale bar, 5 μm. **h**, Quantification of RT–PCR products using iPS cell-derived neurons made from an independent iPS cell line, NCRM5, with a non-targeting control short guide RNA (sgRNA) (sgTARDBP−), an intermediate TDP-43 knockdown (sgTARDBP+) or stronger TDP-43 knockdown (sgTARDBP++). Data are mean ± s.e.m. sgControl, *n* = 6; sgTARDBP+, *n* = 5; sgTARDBP++, *n* = 6; one-way ANOVA with multiple comparisons. *$P$ < 0.05, **$P$ < 0.01, ***$P$ < 0.001, ****$P$ < 0.0001. **i**, Schematic of nanopore long reads quantified in **j**, Extended Data Figs. 2d, e, 5e, f. **j**, Percentage of targeted *UNC13A* long reads with TDP-43-regulated splice events that contain CE, intron retention or both in TDP-43-knockdown SH-SY5Y cells.

## *UNC13A* cryptic exon production on TDP-43 knockdown

To identify novel CEs promoted by TDP-43 depletion, we performed RNA sequencing (RNA-seq) on human induced pluripotent stem (iPS) cell-derived cortical-like i[3]Neurons, in which we reduced TDP-43 expression using CRISPR inhibition[11–13] (CRISPRi). Differential splicing and expression analyses identified 179 CEs, including several that have been reported previously, in genes including *AGRN*, *RAP1GAP*, *PFKP* and *STMN2*[7,8,14] (Fig. 1a, Supplementary Data 1, 2). We examined splicing, expression, ALS GWAS[15] risk genes and diagnostic panel genes for ALS and FTD[16]. Of the 179 CE-harbouring genes, only the synaptic gene *UNC13A* was also an ALS–FTD risk gene (Fig. 1b, c, Supplementary Table 1). *UNC13A* polymorphisms modify both disease risk and progression in ALS and FTLD-TDP[1–3,15,17–19], suggesting a potential functional relationship between TDP-43, *UNC13A* and disease risk. Inspection of *UNC13A* splicing revealed the presence of a CE, occurring in two forms distinguishable by their size, between exons 20 and 21 after TDP-43 knockdown (Fig. 1b), and increased intron retention between exons 31 and 32 (Extended Data Fig. 1a). One ALS-TDP and FTLD-TDP risk SNP—*rs12973192*[15]—lies 16 bp inside the CE (hereafter referred to as the CE SNP). Another SNP—*rs12608932*[1]—is located 534 bp downstream of the donor splice site of the CE within the same intron (hereafter referred to as the intronic SNP) (Fig. 1c). There are five polymorphisms associated with ALS risk in *UNC13A*[15]. All are in high linkage disequilibrium with both the CE and intronic SNPs in European populations, with an allele frequency of 0.3423 and 0.3651, respectively[20] (Fig. 1d). The proximity of the disease-associated SNPs to the *UNC13A* CE suggests that the SNPs may influence *UNC13A* splicing. Of note, we also observed robust mis-splicing in *UNC13B*, which encodes another member of the UNC13 synaptic protein family (Fig. 1e, f). TDP-43 knockdown led to the inclusion of an annotated frame-shift-inducing exon between exons 10 and 11 in *UNC13B*, hereafter referred to as the *UNC13B* frameshift exon (FSE), and increased intron retention between exon 21 and 22 (Fig. 1e, f, Extended Data Fig. 1b).

We validated the *UNC13A* CE in i[3]Neurons by in situ hybridization, which showed a primarily nuclear localization and occurred predominantly in TDP-43-knockdown neurons (Fig. 1g, Extended Data Fig. 1c). To confirm the CE was not restricted to neurons derived from a single iPS cell line, we performed TDP-43 knockdown in independent i[3]Neurons using two different guides leading to different levels of TDP-43 knockdown (Extended Data Fig. 1d, e). CE expression was restricted to cells with TDP-43 knockdown in both lines, and correlated with the level of TDP-43 knockdown (Fig. 1h, Extended Data Fig. 1f, g). We also detected these splicing changes in RNA-seq data we generated from TDP-43 depleted SH-SY5Y and SK-N-BE(2) neuronal lines, and publicly available RNA-seq from iPS cell-derived motor neurons[9] and SK-N-BE(2) datasets[21] (Extended Data Fig. 1h–k, Supplementary Table 2). We note

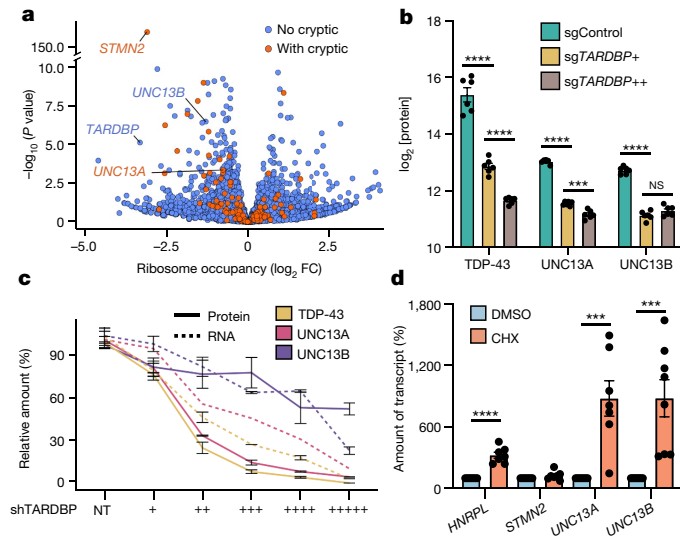

**a**, Ribosome profiling of TDP-43-knockdown i³Neurons shows a reduction in ribosome occupancy of *STMN2*, *UNC13A* and *UNC13B* transcripts. **b**, Mass spectrometry-based proteomic analysis shows dose-dependent reduction in protein abundance of UNC13A and TDP-43 upon TDP-43 knockdown in i³Neurons. $n = 6$ biological replicates. Two-sample $t$-test. **c**, Protein and RNA quantification of TDP-43, UNC13A and UNC13B in SH-SY5Y with varying levels of doxycycline-inducible TDP-43 knockdown. $n = 3$ biological replicates. **d**, Transcript expression upon treatment with CHX suggests that *UNC13A* and *UNC13B*, but not *STMN2*, are sensitive to NMD. *HNRNPL* is used as a positive control. $n = 7$ biological replicates (*UNC13A*, *HNRNPL* and *STMN2*) and 8 biological replicates (*UNC13B*). One-sample $t$-test. Data are mean ± s.e.m. (**b**–**d**).

that the expression of these events was lowest in the SH-SY5Y experiment, which also showed the weakest TDP-43 knockdown (Extended Data Fig. 1l). Using stronger TDP-43 knockdown, we validated the *UNC13A* CE by PCR with reverse transcription (RT–PCR) and Sanger sequencing in SH-SY5Y and SK-N-BE(2) cell lines (Extended Data Fig. 2a).

In support of a direct role for TDP-43 regulation of *UNC13A* and *UNC13B*, we found multiple TDP-43-binding peaks[22] both downstream and within the body of the *UNC13A* CE (Fig. 1c) and intron retention (Extended Data Fig. 1a). Additionally, TDP-43-binding peaks[22] were present near both splice events in *UNC13B* (Fig. 1f, Extended Data Fig. 1b). Additional iCLIP of endogenous TDP-43 in SH-SY5Y cells confirmed enhanced binding near the *UNC13A* CE and intron retention and *UNC13B* FSE and intron retention (Extended Data Fig. 2 b, c). We next tested whether the *UNC13A* intron retention and CE events co-occured in transcripts. Using targeted long-read sequencing, we determined that although co-regulated, *UNC13A* CE and intron retention occurred largely independently from each other (Fig. 1l, j; Extended Data Fig. 2d,e).

## *UNC13A* is downregulated on TDP-43 knockdown

Next, we examined whether incorrect splicing of *UNC13A* and *UNC13B* affected transcript levels in neurons and neuron-like cells. TDP-43 knockdown significantly reduced *UNC13A* RNA abundance in the three experiments with the highest levels of cryptic splicing (false discovery rate (FDR) < 0.0001; Extended Data Figs. 1h, 3a). Similarly, *UNC13B* RNA was significantly downregulated in four datasets (FDR < 0.0001) (Extended Data Fig. 3b). We confirmed these results by quantitative PCR

(qPCR) in i³Neurons, and SH-SY5Y and SK-N-BE(2) cell lines (Extended Data Figs. 1d, e, 3c, d). The number of ribosome footprints aligning to *UNC13A* and *UNC13B* was also reduced after TDP-43 knockdown (Fig. 2a, Extended Data Fig. 3e, Supplementary Data 3; FDR < 0.05). Notably, TDP-43 knockdown decreased expression of UNC13A and UNC13B at the protein level in a dose-dependent manner, as assessed by quantitative proteomics (Fig. 2b).

To assess the relation between TDP-43 reduction and UNC13 splicing, RNA and protein levels, we assayed SH-SY5Y cells with increasing amounts of TDP-43 knockdown. We found that changes in UNC13A paralleled those of TDP-43, whereas UNC13B levels were less affected (Fig. 2c, Extended Data Fig. 3f, Supplementary Fig. 1). *UNC13A* CE inclusion and intron retention increased on TDP-43 knockdown, with the CE being detected only after more than 50% TDP-43 loss, whereas *UNC13B* FSE and intron retention were not robustly detected until more than 90% of TDP-43 expression was lost (Fig. 2c, Extended Data Fig. 3g).

To assess whether the amount of *UNC13A* CE expression was underestimated owing to efficient transcript degradation, we investigated whether it promoted NMD, as predicted by the presence of a premature termination codon. Knockdown of the key NMD factor *UPF1* or cycloheximide (CHX) treatment—which stalls translation and impairs NMD—increased the amount of *UNC13A* CE and *UNC13B* FSE, which also leads to a PTC at the beginning of exon 11, confirming that both *UNC13A* and *UNC13B* were targeted by NMD (Fig. 2d, Extended Data Fig. 3h, i). Conversely, CHX treatment and *UPF1* knockdown did not alter levels of the aberrant *STMN2* transcript, which was not predicted to undergo NMD (Fig. 2d, Extended Data Fig. 3h). Of note, CHX treatment of SH-SY5Y cells with the least TDP-43 knockdown (Fig. 2c) enabled detection of the *UNC13A* CE, supporting the notion that its occurrence may be underestimated owing to efficient degradation (Extended Data Fig. 3j, k).

Together, these data suggest that TDP-43 has an essential role in ensuring the correct pre-mRNA splicing of *UNC13A* and *UNC13B*, thereby maintaining normal expression of these key presynaptic proteins.

## *UNC13A* cryptic exon in patient neurons

To test whether the *UNC13A* CE could be detected in tissues from patients affected by TDP-43 pathology, we first analysed RNA-seq data from neuronal nuclei sorted from frontal cortices of patients with ALS–FTLD[23]. We compared the levels of *UNC13A* CE to the levels of a CE in *STMN2* known to be regulated by TDP-43. Both *STMN2* and *UNC13A* CEs were found exclusively in TDP-43-depleted nuclei. Although the lack of NMD activity in the nucleus means that the nuclear splicing ratio may not reflect that of the whole cell, in some cases, the *UNC13A* CE percent spliced in (PSI ($\Psi$)) reached 100% (Fig. 3a). This suggests that there is a substantial loss of UNC13A expression in the subpopulation of neurons with TDP-43 pathology in human patients with ALS–FTLD.

Next, we quantified *UNC13A* CE inclusion in bulk RNA-seq data from the New York Genome Center (NYGC) ALS Consortium, which contains 1,349 brain and spinal cord tissues from a total of 377 individuals, including those with ALS or FTLD and controls. The *UNC13A* CE was detected exclusively in tissues from individuals with FTLD-TDP and ALS-TDP (89% and 38% of individuals, respectively), and not in individuals with non-TDP ALS (caused by *SOD1* and *FUS* mutations), FTLD associated with TAU (FTLD-TAU), FTLD associated with FUS (FTLD-FUS) or controls. There were no systematic differences across tissues between controls and non-TDP ALS or FTLD and ALS-TDP or FTLD-TDP in confounding factors such as library depth, RNA integrity number or cellular composition, which could explain the *UNC13A* CE specificity (Extended Data Fig. 4a–d). The lower detection rate in ALS compared with FTLD is possibly owing to the lower expression of *UNC13A* in the spinal cord (Extended Data Fig. 4a), although differences in NMD efficiency between cortical and spinal regions could also affect detection rate[24]. *UNC13A* CE was more likely to be detected in

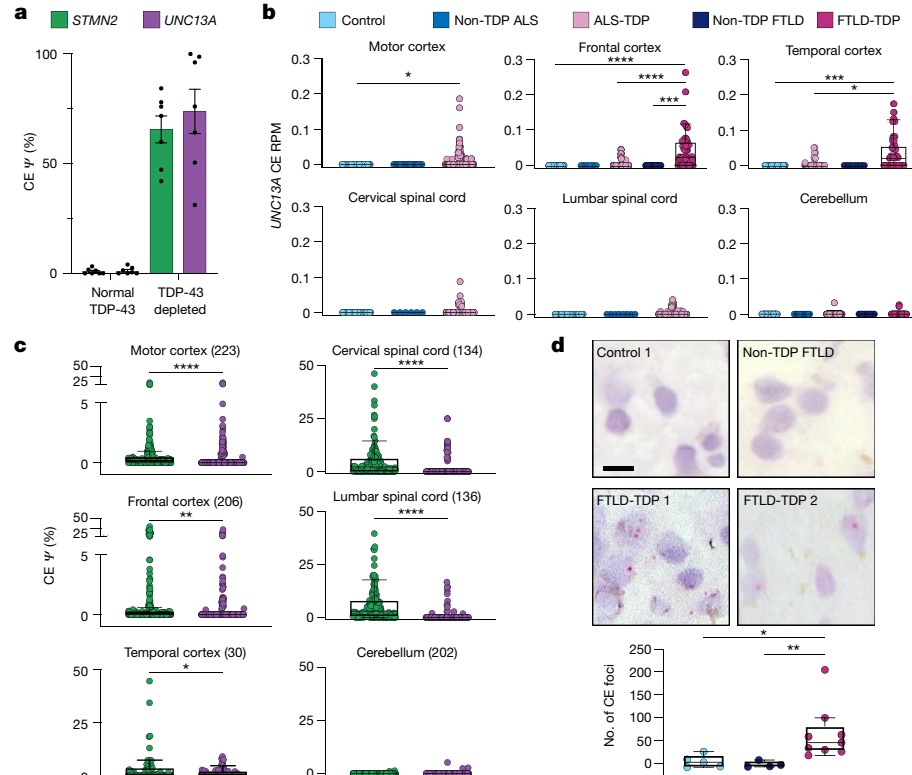

**Fig. 3 | *UNC13A* CE is highly expressed in tissues from patients with ALS or FTLD and correlates with known markers of TDP-43 loss of function.** **a**, *UNC13A* and *STMN2* CE expression from a published dataset of frontal cortex neuronal nuclei from patients with ALS or FTLD sorted according to TDP-43 expression[23]. **b**, *UNC13A* CE expression in bulk RNA-seq from the NYGC ALS Consortium data normalized by library size across disease and tissue samples. ALS cases are stratified by mutation status, FTLD cases are stratified by

pathological subtype. **c**, CE expression throughout ALS-TDP and FTLD-TDP cases across tissue, number of tissue samples in brackets. **d**, BaseScope detection of *UNC13A* CE (red foci) in FTLD-TDP (9 individuals) but not control (5 individuals) or non-TDP FTLD (FTLD-TAU) (4 individuals) frontal cortex samples and quantification of background-corrected foci frequency between groups. Scale bar, 10 μm. Data are mean ± s.e.m. (**b**–**d**); Wilcoxon test.

bulk samples that had been sequenced with 125-bp rather than 100-bp paired-end reads, but other technical factors did not systematically affect detection (Extended Data Fig. 5a–d).

*UNC13A* CE expression mirrored the known tissue distribution of TDP-43 aggregation and nuclear clearance[25]: it was specific to ALS-TDP spinal cord and motor cortex, as well as FTLD-TDP frontal and temporal cortices, but was absent from the cerebellum in both disease and control states (Fig. 3b). Furthermore, although the *UNC13A* CE induces NMD (unlike the *STMN2* CE) it was detected at similar levels to the *STMN2* CE in cortical regions, whereas *STMN2* CE was more abundant in the spinal cord (Fig. 3c). Targeted long-read sequencing of *UNC13A* in FTLD frontal cortex revealed that the CE and intron retention events can co-occur but, as in SH-SY5Y cells, they are mostly detected independently (Extended Data Fig. 5e, f). Thus, pathological *UNC13A* CEs occur in vivo and are specific to neurodegenerative disease subtypes in which mislocalization and nuclear depletion of TDP-43 occurs.

We next assessed expression of the *UNC13B* FSE across the NYGC dataset. We detected no increase in the *UNC13B* FSE in pathological ALS-TDP or FTLD-TDP tissues. However, the presence throughout control and ALS or FTD brains of a shorter isoform of the CE that included the FSE, which was absent in our in vitro experiments, may mask underlying changes (Extended Data Fig. 6a–c).

We also evaluated both *UNC13A* and *UNC13B* intron retention events from bulk RNA-seq. Unlike the CE, both intron retention events were also detected in control brains, making it difficult to determine whether TDP-43 pathology increased intron retention (Extended Data Fig. 7a, b).

We next investigated whether *UNC13A* CEs could be visualized by in situ hybridization in brains from patients with FTLD, using the same

probe used for iPS cell-derived neurons. We detected red foci in cortical neurons at a significantly higher frequency in FTLD-TDP relative to both neurologically normal controls (Kruskal–Wallis test, $P = 0.021$) and non-TDP FTLD (FTLD-TAU) ($P = 0.010$) (Fig. 3d).

To assess whether *UNC13A* CE levels in bulk tissue were related to the level of TDP-43 proteinopathy, we used the *STMN2* CE PSI as a proxy. The PSI of *STMN2* CE correlates with the cryptic PSI of other well-known TDP-43 induced CEs, such as those in *RAP1GAP* and *PFKP*[7,9,14] (Extended Data Fig. 7c,d) and correlates with the amount of phosphorylated TDP-43 in patient samples[10]. As expected, across the NYGC ALS Consortium samples we observed a significant positive correlation between the level of *STMN2* CE PSI and *UNC13A* CE PSI across the NYGC ALS Consortium samples (rho = 0.56, $P = 2.9 \times 10^{-7}$, $n = 72$ cortical samples) (Extended Data Fig. 7e).

Collectively, our analysis reveals a strong relationship between TDP-43 pathology and the level of *UNC13A* CE, supporting a model with direct regulation of *UNC13A* mRNA splicing by TDP-43.

## *UNC13A* risk SNPs exacerbate cryptic splicing

To test whether the ALS–FTD risk SNPs in *UNC13A* promote cryptic splicing, thereby explaining their link to disease, we assessed *UNC13A* CE levels across different genotypes. We found significantly increased *UNC13A* CE in cases homozygous for CE rs12973192 (G) and intronic rs12608932 (C) SNPs ($P = 0.028$, Wilcoxon test) (Extended Data Fig. 8a, Supplementary Table 4). To ensure that this was not owing to more severe TDP-43 loss of function in these samples, we normalized *UNC13A* CE by the level of *STMN2* cryptic splicing, a well-established product of TDP-43 loss of function. Again, we found significantly increased *UNC13A*

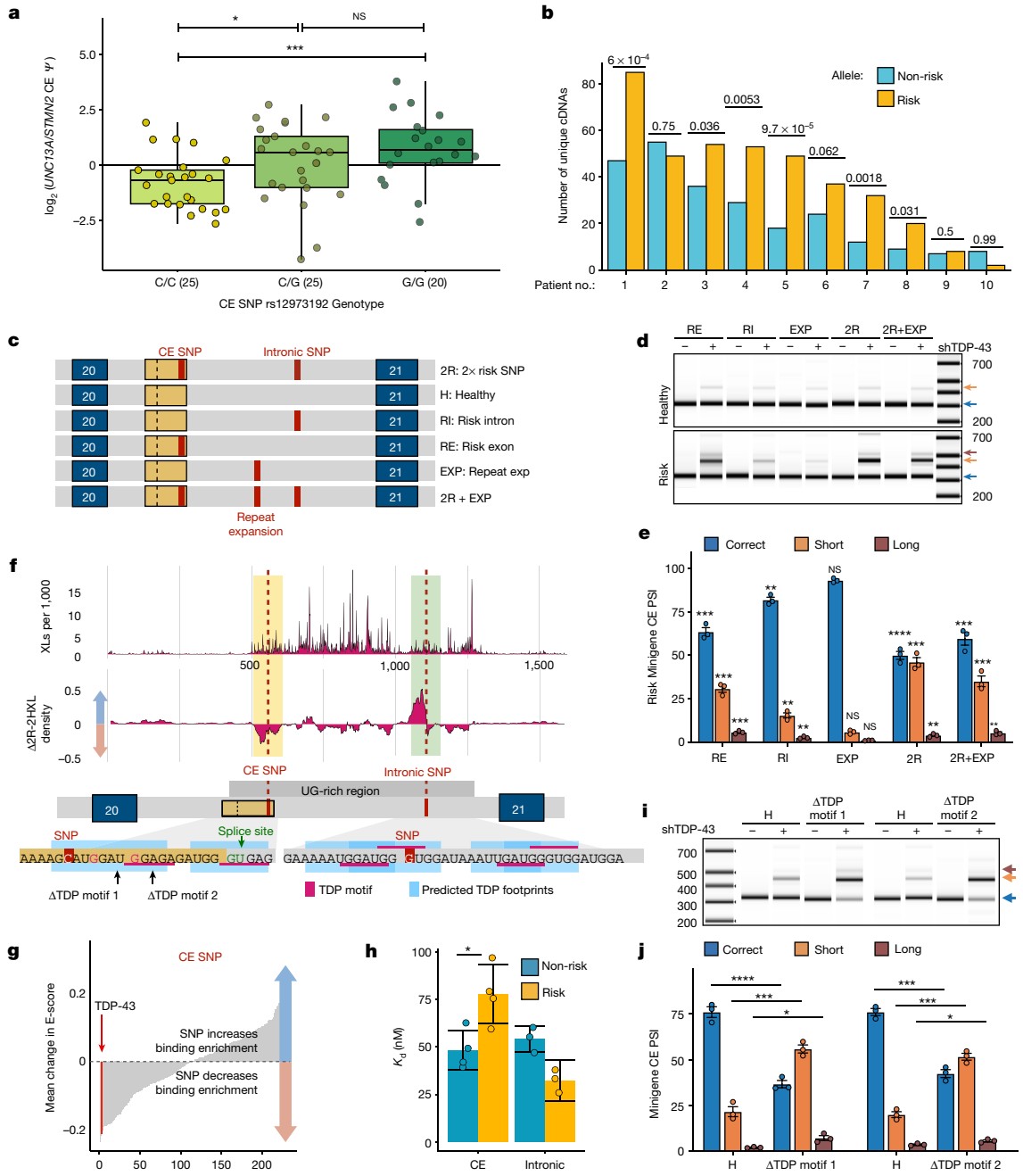

**Fig. 4 | _UNC13A_ ALS and FTD risk variants enhance _UNC13A_ CE splicing in patients and in vitro by altering TDP-43 pre-mRNA binding. a**, Ratio of _UNC13A_ and _STMN2_ CE Ψ in ALS-TDP and FTLD-TDP cortex, split by genotype for _UNC13A_ risk alleles. In box plots, the centre line shows the median, box edges delineate 25th and 75th percentiles and Tukey whiskers are plotted. **b**, Unique cDNAs from targeted RNA-seq in ten patients with FTLD-TDP who are heterozygous for the risk SNP within the _UNC13A_ cryptic exon. Single-tailed binomial tests. Patients 1, 5 and 7 carry the C9orf72 hexanucleotide repeat. **c**, Schematic of _UNC13A_ minigenes containing exon 20, intron 20 and exon 21 and combinations of _UNC13A_ alleles. **d**, **e**, Representative image (**d**) and quantification (**e**) of RT–PCR products from _UNC13A_ minigenes in SH-SY5Y cells with or without TDP-43 knockdown. Data are mean ± s.e.m. Each variant was compared with the healthy minigene with which it was co-transfected and results were compared with an unpaired _t_-test (_n_ = 3 biological replicates). **f**, TDP-43 iCLIP of SH-SY5Y cells containing 2R and 2H minigenes. Top, average crosslink density. Middle, average density change 2R for − 2H (20-nt rolling window, units are crosslinks per 1,000). Bottom, predicted TDP-43 binding footprints (UGNNUG motif). **g**, Average change in E-value (measure of binding enrichment) across proteins for heptamers containing risk or healthy CE SNP alleles. TDP-43 is indicated in red. **h**, Binding affinities between TDP-43 and 14-nt RNA containing the CE (_n_ = 4) or intronic (_n_ = 3) healthy or risk sequences measured by isothermal titration calorimetry. Data are mean ± s.d.; two-sample _t_-test. **i**, **j**, Representative image (**i**) and quantification (**j**) of RT–PCR products from _UNC13A_ minigenes with mutated UGNNUG TDP-43 binding motifs as shown in **f**. Data are mean ± s.e.m.; _n_ = 3 biological replicates; statistical analysis as in (**d**, **e**).

CE in cases with homozygous risk variants (_P_ < 0.001, Wilcoxon test) (Fig. 4a, Extended Data Fig. 8b). Next, we performed targeted RNA-seq on _UNC13A_ CE from temporal cortices of ten FTLD-TDP patients who were heterozygous in the risk allele and four controls (Supplementary Table 5). There was no detection of the CE in the control samples, and in the patient samples we detected significant biases towards reads

containing the risk allele ($P < 0.05$, single-tailed binomial test) in six samples, with a seventh sample approaching significance (Fig. 4b), suggesting that the two ALS- and FTLD-linked variants promote cryptic splicing in vivo.

To directly assess whether the risk SNPs increase CE inclusion, we performed minigene experiments. Using two minigenes containing *UNC13A* exon 20, intron 20 and exon 21, with and without the two ALS- and FTLD-linked variants, we determined that the risk variants enhanced CE upon TDP-43 loss (Extended Data Fig. 8c). To examine whether the CE SNP, intronic SNP or short tandem repeat expansion rs56041637—which is in linkage disequilibrium with the two SNPs[26]—are responsible for promoting the CE inclusion, we generated minigenes featuring different combinations of these genomic variants (Fig. 4c). Using quantitative analysis of RT–PCR products, we found that both the CE SNP and, to a lesser extent, the intronic SNP independently promoted CE inclusion, with the highest overall levels detected for the 2R minigene (Fig. 4d, e, Extended Data Fig. 8d).

TDP-43 can both inhibit and promote splicing by binding to pre-mRNA. We performed TDP-43 iCLIP in HEK 293T cells expressing either the 2R or the 2H minigene to fine map TDP-43 binding to *UNC13A* intron 20 and investigate whether the risk SNPs have an effect on this interaction. In agreement with our iCLIP data of endogenous *UNC13A* in SH-SY5Y cells (Extended Data Fig. 2b), we observed an enrichment of crosslinks within the approximately 800-nucleotide UG-rich region containing both SNPs in intron 20 (Fig. 4f). When comparing 2R with 2H, the largest fractional changes were near each SNP (Extended Data Fig. 8e). We detected a 21% decrease in total TDP-43 crosslinks centred around the CE SNP and a 73% increase upstream of the intronic SNP (Fig. 4f, Extended Data Fig. 8f; 50-nucleotide windows). These data suggest these two disease-risk SNPs distort the pattern of TDP-43–RNA interactions, decreasing TDP-43 binding near the CE donor splice site.

To explore whether these two SNPs directly influence TDP-43 binding, we analysed a dataset of in vitro RNA heptamer–RBP binding enrichments. We examined the effect of the SNPs on relative RBP enrichment[27] by comparing healthy versus risk SNP-containing heptamers. When investigating which RBPs were most affected in their RNA binding enrichment by the CE risk SNP, TDP-43 had the third largest decrease of any RBP, with only two non-mammalian RBPs showing a larger decrease (Fig. 4g, Extended Data Fig. 8g). The intronic SNP did not strongly affect TDP-43 binding, although data was only available for 2 out of 7 possible heptamers (Extended Data Fig. 8h, i). To verify that the CE SNP directly inhibited TDP-43 binding, we performed isothermal titration calorimetry using recombinant TDP-43 and short RNAs. We observed nanomolar binding affinity in all cases, with an increased dissociation constant ($K_d$) (lower binding affinity) for the CE SNP region ($P = 0.023$, two-sample $t$-test) and a trend of decreasing $K_d$ for the intronic SNP region ($P = 0.052$) when the risk variants were present (Fig. 4h, Extended Data Fig. 9a–d, Supplementary Data 4). Last, to test whether TDP-43 binding to the CE SNP region is critical for CE repression, we mutated the UGNNUG TDP-43-binding motif in this region, leading to significantly increased CE inclusion (Fig. 4i, j, Extended Data Fig. 9e). Together these data suggest that the risk SNPs modulate TDP-43 binding, in part via direct changes in binding affinity, exacerbating *UNC13A* CE inclusion.

## Discussion

Our results support a model in which *UNC13A* SNPs and TDP-43 loss synergistically drive cryptic exon inclusion in *UNC13A* transcripts, thereby reducing expression of a synaptic gene that is critical for normal neuronal function.

In this model, when nuclear TDP-43 levels are normal in healthy individuals, TDP-43 binds efficiently to *UNC13A* pre-mRNA, even in the presence of risk SNPs, thus preventing CE splicing. Conversely, severe nuclear depletion of TDP-43 in end-stage disease induces CE inclusion in all cases. However, in the setting of partial TDP-43 loss that occurs early in degenerating neurons, risk-associated intronic and CE risk SNPs alter TDP-43 binding to *UNC13A* pre-mRNA, exacerbating CE inclusion in these transcripts. The ensuing loss of UNC13A protein—which is critical for normal synaptic activity—at earlier disease stages may explain the associated risk effect of these SNPs. Notably, we found that both risk alleles for these SNPs independently and additively promoted cryptic splicing in vitro. Further, when the two variants are not co-inherited, as seen in individuals from east Asia with ALS, an attenuated effect is observed[19]. A similar phenomenon, in which SNP pairs both contribute to risk, has been widely studied at the APOE locus in Alzheimer's disease[28]. Clarification of single versus additive effects of co-inherited SNPs regarding effects on CE inclusion, as well as contributions of other RBPs, will require future investigation.

UNC13 family proteins are highly conserved across metazoans and are essential for calcium-triggered synaptic vesicle release[29]. In mice, single knockout of *Unc13a* blocks action potential-induced neurotransmitter release from the majority of glutamatergic hippocampal synapses[30]. Double knockout of *Unc13a* and *Unc13b* inhibits both excitatory and inhibitory synaptic transmission in hippocampal neurons and greatly impairs transmission at neuromuscular junctions[31,32]. In TDP-43-depleted neuronal nuclei derived from patients with ALS or FTLD, which reflect transcript expression before NMD, the *UNC13A* CE is present in up to 100% of transcripts, suggesting that expression of functional *UNC13A* is markedly reduced, which could affect normal synaptic transmission.

TDP-43 loss induces hundreds of splicing changes, several of which have also been detected in brains of patients with ALS or FTLD. However, it has remained unclear whether these events—even those that occur in essential neuronal genes—contribute to disease pathogenesis. The fact that genetic variation modulating *UNC13A* CE levels influences the rate of ALS progression strongly supports the role of *UNC13A* downregulation as an important effector of neurotoxicity mediated by TDP-43 loss. The *UNC13A* CE is thus a promising target for therapies that modulate splicing, potentially applicable to 97% of ALS cases and approximately half of FTD cases. These findings are also of interest to other neurodegenerative diseases—such as Alzheimer's disease, Parkinson's disease and chronic traumatic encephalopathy—in which TDP-43 depletion occurs in a substantial fraction of cases.

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

NYGC ALS Consortium

Hemali Phatnani[7], Justin Kwan[20], Dhruv Sareen[21,22], James R. Broach[23], Zachary Simmons[24], Ximena Arcila-Londono[25], Edward B. Lee[26], Vivianna M. Van Deerlin[26], Neil A. Shneider[27], Ernest Fraenkel[28], Lyle W. Ostrow[29], Frank Baas[30,31], Noah Zaitlen[32], James D. Berry[33,34], Andrea Malaspina[35,36,37], Pietro Fratta[1], Gregory A. Cox[38], Leslie M. Thompson[39,40], Steve Finkbeiner[41], Efthimios Dardiotis[42], Timothy M. Miller[43], Siddharthan Chandran[44], Suvankar Pal[44], Eran Hornstein[45], Daniel J. MacGowan[46], Terry Heiman-Patterson[47], Molly G. Hammell[48], Nikolaos. A. Patsopoulos[49], Oleg Butovsky[49], Joshua Dubnau[50], Avindra Nath[51], Robert Bowser[52,53], Matthew Harms[54], Eleonora Aronica[55], Mary Poss[56], Jennifer Phillips-Cremins[57], John Crary[13,14,58], Nazem Atassi[59], Dale J. Lange[60,61], Darius J. Adams[62,63], Leonidas Stefanis[64,65], Marc Gotkine[66], Robert H. Baloh[67,68], Suma Babu[34], Towfique Raj[13,14,15,16], Sabrina Paganoni[69], Ophir Shalem[70,71], Colin Smith[72,73], Bin Zhang[74], Brent Harris[75], Iris Broce[76], Vivian Drory[77], John Ravits[78], Corey McMillan[79], Vilas Menon[80], Lani Wu[81], Steven Altschuler[81], Yossef Lerner[82], Rita Sattler[83], Kendall Van Keuren-Jensen[84], Orit Rozenblatt-Rosen[85], Kerstin Lindblad-Toh[85], Katharine Nicholson[86], Peter Gregersen[87], Jeong-Ho Lee[88], Sulev Koks[89] & Stephen Muljo[90]

[20]Department of Neurology, Lewis Katz School of Medicine, Temple University, Philadelphia, PA, USA. [21]Cedars-Sinai Department of Biomedical Sciences, Board of Governors Regenerative Medicine Institute and Brain Program, Cedars-Sinai Medical Center, University of California, Los Angeles, CA, USA. [22]Department of Medicine, University of California, Los Angeles, CA, USA. [23]Department of Biochemistry and Molecular Biology, Penn State Institute for Personalized Medicine, The Pennsylvania State University, Hershey, PA, USA. [24]Department of Neurology, The Pennsylvania State University, Hershey, PA, USA. [25]Department of Neurology, Henry Ford Health System, Detroit, MI, USA. [26]Department of Pathology and Laboratory Medicine, Perelman School of Medicine, University of Pennsylvania, Philadelphia, PA, USA. [27]Department of Neurology, Center for Motor Neuron Biology and Disease, Institute for Genomic Medicine, Columbia University, New York, NY, USA. [28]Department of Biological Engineering, Massachusetts Institute of Technology, Cambridge, MA, USA. [29]Department of Neurology, Johns Hopkins School of Medicine, Baltimore, MD, USA. [30]Department of Neurogenetics, Academic Medical Centre, Amsterdam, The Netherlands. [31]Leiden University Medical Center, Leiden, The Netherlands. [32]Department of Medicine, Lung Biology Center, University of California, San Francisco, CA, USA. [33]ALS Multidisciplinary Clinic, Neuromuscular Division, Department of Neurology, Harvard Medical School, Boston, MA, USA. [34]Neurological Clinical Research Institute, Massachusetts General Hospital, Boston, MA, USA. [35]Centre for Neuroscience and Trauma, Blizard Institute, Barts Queen Mary University of London, London, UK. [36]The London School of Medicine and Dentistry, Queen Mary University of London, London, UK. [37]Department of Neurology, Basildon University Hospital, Basildon, UK. [38]The Jackson Laboratory, Bar Harbor, ME, USA. [39]Department of Psychiatry and Human Behavior, Department of Biological Chemistry, School of Medicine, University of California, Irvine, CA, USA. [40]Department of Neurobiology and Behavior, School of Biological Sciences, University of California, Irvine, CA, USA. [41]Taube/Koret Center for Neurodegenerative Disease Research, Roddenberry Center for Stem Cell Biology and Medicine, Gladstone Institute, San Francisco, CA, USA. [42]Department of Neurology and Sensory Organs, University of Thessaly, Thessaly, Greece. [43]Department of Neurology, Washington University in St Louis, St Louis, MO, USA. [44]Centre for Clinical Brain Sciences, Anne Rowling Regenerative Neurology Clinic, Euan MacDonald Centre for Motor Neurone Disease Research, University of Edinburgh, Edinburgh, UK. [45]Department of Molecular Genetics, Weizmann Institute of Science, Rehovot, Israel. [46]Department of Neurology, Icahn School of Medicine at Mount Sinai, New York, NY, USA. [47]Center for Neurodegenerative Disorders, Department of Neurology, the Lewis Katz School of Medicine, Temple University, Philadelphia, PA, USA. [48]Cold Spring Harbor Laboratory, Cold Spring Harbor, NY, USA. [49]Ann Romney Center for Neurologic Diseases, Brigham and Women's Hospital, Harvard Medical School, Boston, MA, USA. [50]Department of Anesthesiology, Stony Brook University, Stony Brook, NY, USA. [51]Section of Infections of the Nervous System, National Institute of Neurological Disorders and Stroke, NIH, Bethesda, MD, USA. [52]Department of Neurology, Barrow Neurological Institute, St Joseph's Hospital, Phoenix, AZ, USA. [53]Medical Center, Department of Neurobiology, Barrow Neurological Institute, St Joseph's Hospital and Medical Center, Phoenix, AZ, USA. [54]Department of Neurology, Division of Neuromuscular Medicine, Columbia University, New York, NY, USA. [55]Department of Neuropathology, Academic Medical Center, University of Amsterdam, Amsterdam, The Netherlands. [56]Department of Biology and Veterinary and Biomedical Sciences, The Pennsylvania State University, University Park, PA, USA. [57]New York Stem Cell Foundation, Department of Bioengineering, School of Engineering and Applied Sciences, University of Pennsylvania, Philadelphia, PA, USA. [58]Department of Pathology, Fishberg Department of Neuroscience, Icahn School of Medicine at Mount Sinai, New York, NY, USA. [59]Department of Neurology, Harvard Medical School, Neurological Clinical Research Institute, Massachusetts General Hospital, Boston, MA, USA. [60]Department of Neurology, Hospital for Special Surgery, New York, NY, USA. [61]Weill Cornell Medical Center, New York, NY, USA. [62]Medical Genetics, Atlantic Health System, Morristown Medical Center, Morristown, NJ, USA. [63]Overlook Medical Center, Summit, NJ, USA. [64]Center of Clinical Research, Experimental Surgery and Translational Research, Biomedical Research Foundation of the Academy of Athens (BRFAA), Athens, Greece. [65]1st Department of Neurology, Eginition Hospital, Medical School, National and Kapodistrian University of Athens, Athens, Greece. [66]Neuromuscular/EMG service and ALS/Motor Neuron Disease Clinic, Hebrew University-Hadassah Medical Center, Jerusalem, Israel. [67]Board of Governors Regenerative Medicine Institute, Los Angeles, CA, USA. [68]Department of Neurology, Cedars-Sinai Medical Center, Los Angeles, CA, USA. [69]Harvard Medical School, Department of Physical Medicine and Rehabilitation, Spaulding Rehabilitation Hospital, Boston, MA, USA. [70]Center for Cellular and Molecular Therapeutics, Children's Hospital of Philadelphia, Philadelphia, PA, USA. [71]Department of Genetics, Perelman School of Medicine, University of Pennsylvania, Philadelphia, PA, USA. [72]Centre for Clinical Brain Sciences, University of Edinburgh, Edinburgh, UK. [73]Euan MacDonald Centre for Motor Neurone Disease Research, University of Edinburgh, Edinburgh, UK. [74]Department of Genetics and Genomic Sciences, Icahn Institute of Data Science and Genomic Technology, Icahn School of Medicine at Mount Sinai, New York, NY, USA. [75]Department of Neuropathology, Georgetown Brain Bank, Georgetown Lombardi Comprehensive Cancer Center, Georgetown University Medical Center, Washington, DC, USA. [76]Neuroradiology Section, Department of Radiology and Biomedical Imaging, University of California, San Francisco, San Francisco, CA, USA. [77]Neuromuscular Diseases Unit, Department of Neurology, Tel Aviv Sourasky Medical Center, Sackler Faculty of Medicine, Tel-Aviv University, Tel-Aviv, Israel. [78]Department of Neuroscience, University of California San Diego, La Jolla, CA, USA. [79]Department of Neurology, University of Pennsylvania Perelman School of Medicine, Philadelphia, PA, USA. [80]Department of Neurology, Columbia University Medical Center, New York, NY, USA. [81]Department of Pharmaceutical Chemistry, University of California San Francisco, San Francisco, CA, USA. [82]Hadassah Hebrew University, Jerusalem, Israel. [83]Department of Translational Neuroscience, Barrow Neurological Institute, Phoenix, AZ, USA. [84]The Translational Genomics Research Institute (TGen), Phoenix, AZ, USA. [85]Broad Institute, Cambridge, MA, USA. [86]Massachusetts General Hospital, Boston, MA, USA. [87]Institute of Molecular Medicine, Feinstein Institutes for Medical Research, Northwell Health, Manhasset, NY, USA. [88]Korea Advanced Institute of Science and Technology (KAIST), Daejeon, South Korea. [89]Perron Institute for Neurological and Translational Science, Nedlands, Western Australia, Australia. [90]Integrative Immunobiology Section, National Institute of Allergy and Infectious Disease, NIH, Bethesda, MD, USA.

# Methods

## Human iPS cell culture

All policies of the NIH Intramural research program were followed for the procurement and use of iPS cells. For most studies, the iPS cells used were from the WTC11 line, derived from a healthy 30-year-old male, and obtained from the Coriell cell repository. Infomed consent was obtained from the donor. We confirmed the WTC11 line contained no ALS–FTD mutations in the ALS and FTD risk genes in Supplementary Table 1. For key experiments, an independent line was used, NCRM5. NCRM5 was derived from umbilical cord blood from NIH Center for Regenerative Medicine (CRM), Bethesda, MD, USA. Informed consent was obtained from the donor. All culture procedures were conducted as previously[11]. In brief, iPS cells were grown on tissue culture dishes coated with human embryonic stem cell-qualified Matrigel (Corning, catalogue no. 354277). They were maintained in Essential 8 Medium (E8; Thermo Fisher Scientific, catalogue (cat.) no. A1517001) supplemented with 10 μM ROCK inhibitor (RI; Y-27632; Selleckchem, cat. no. S1049) in a 37 °C, 5% $CO_2$ incubator. Medium was replaced every 1–2 days as needed. Cells were passaged with accutase (Life Technologies, cat. no. A1110501), 5–10 min treatment at 37 °C. Accutase was removed and cells were washed with PBS before re-plating. Following dissociation, cells were plated in E8 media supplemented with 10 μM RI to promote survival. RI was removed once cells grew into colonies of 5–10 cells.

The following cell line and DNA samples were obtained from the NIGMS Human Genetic Cell Repository at the Coriell Institute for Medical Research: GM25256.

## Data

Publicly available data were obtained from the Gene Expression Omnibus (GEO): iPS cell MNs[9], GSE121569; SK-N-BE(2)[b], GSE97262; FACS-sorted frontal cortex neuronal nuclei, GSE126543; Riboseq, E-MTAB-10235; targeted RNA-seq, E-MTAB-10237; minigene TDP-43 iCLIP, E-MTAB-10297; SH-SY5Y TDP-43 iCLIP, E-MTAB-11243; and *UNC13A*-targeted nanopore, E-MTAB-11244.

## CRISPRi knockdown in human iPS cells

The human iPS cells used in this study were previously engineered[11,13] to express mouse or human neurogenin-2 (NGN2) under a doxycycline-inducible promoter, as well as an enzymatically dead Cas9 (+/− CAG-dCas9-BFP-KRAB)[12]. For WTC11 these were integrated at the AAVS1 safe harbour and the CLYBL promoter safe harbour respectively, while for NCRM5, these were both integrated at the CLYBL promoter safe harbour.

To achieve knockdown, sgRNAs targeting either *TARDBP*/TDP-43, *UPF1* or a non-targeting control guide were delivered to iPS cells by lentiviral transduction. To make the virus, Lenti-X human embryonic kidney (HEK) cells were transfected with the sgRNA plasmids using Lipofectamine 3000 (Life Technologies, cat. no. L3000150), then cultured for 2–3 days in the following media: DMEM, high glucose GlutaMAX Supplement media (Life Technologies, cat. no. 10566024) with 10% FBS (Sigma, cat. no. TMS-013-B), supplemented with viral boost reagent (ALSTEM, cat. no. VB100). Virus was then concentrated from the media 1:10 in PBS using Lenti-X concentrator (Takara Bio, cat. no. 631231), aliquoted and stored at −80 °C for future use.

The sgRNAs were cloned into either pU6-sgRNA EF1Alpha-puro-T2A-BFP vector[12,37] (gift from J. Weissman; Addgene 60955) or a modified version containing a human U6 promoter, a blasticidin (Bsd) resistance gene, and eGFP. sgRNA sequences were as follows: non-targeting control:GTCCACCCTTATCTAGGCTA, *UPF1*: GGCCAGACGCAGACGCC CCC, and *TARDBP*: GGGAAGTCAGCCGTGAGACC (strong guide), and GCGGCCTAGCGGGTGAGTCG (weaker guide). The stronger *TARDBP* guide was used in all cases unless otherwise stated.

Virus was delivered to iPS cells in suspension following an accutase split. Cells were plated and cultured overnight. The following morning, cells were washed with PBS and media was changed to E8 or E8+RI depending on cell density. Two days after lentiviral delivery, cells were selected overnight with either puromycin (10 μg ml$^{-1}$) or blasticidin (50–100 μg ml$^{-1}$). iPS cells were then expanded 1–2 days before initiating neuronal differentiation. Knockdown efficiency was tested at iPS cell and neuronal stages using immunofluorescence, qPCR and observed in RNA-seq data.

## iPS cell-derived i³Neuron differentiation and culture

To initiate neuronal differentiation, 20–25 million iPS cells per 15 cm plate were individualized using accutase on day 0 and re-plated onto Matrigel-coated tissue culture dishes in N2 differentiation media containing: knockout DMEM/F12 media (Life Technologies Corporation, cat. no. 12660012) with N2 supplement (Life Technologies Corporation, cat. no. 17502048), 1× GlutaMAX (Thermofisher Scientific, cat. no. 35050061), 1× MEM nonessential amino acids (NEAA) (Thermofisher Scientific, cat. no. 11140050), 10 μM ROCK inhibitor (Y-27632; Selleckchem, cat. no. S1049) and 2 μg ml$^{-1}$ doxycycline (Clontech, cat. no. 631311). Media was changed daily during this stage.

On day 3 pre-neuron cells were replated onto dishes coated with freshly made poly-L-ornithine (PLO; 0.1 mg ml$^{-1}$; Sigma, cat. no. P3655-10MG), either 96-well plates (50,000 per well), 6-well dishes (2 million per well), or 15 cm dishes (45 million per plate), in i³Neuron Culture Media: Brain-Phys media (Stemcell Technologies, cat. no. 05790) supplemented with 1× B27 Plus Supplement (ThermoFisher Scientific, cat. no. A3582801), 10 ng ml$^{-1}$ BDNF (PeproTech, cat. no. 450-02), 10 ng ml$^{-1}$ NT-3 (Pepro-Tech, cat. no. 450-03), 1 μg ml$^{-1}$ mouse laminin (Sigma, cat. no. L2020-1MG), and 2 μg ml$^{-1}$ doxycycline (Clontech, cat. no. 631311). i³Neurons were then fed three times a week by half media changes. i³Neuron were then collected on day 7 or 17 after the addition of doxycycline or 4 or 14 days after re-plating.

## Generation of stable TDP-43-knockdown cell line

SH-SY5Y and SK-N-BE(2) cells were transduced with SmartVector lentivirus (V3IHSHEG_6494503) containing a doxycycline-inducible shRNA cassette for TDP-43. Transduced cells were selected with puromycin (1 μg ml$^{-1}$) for one week. For doxycycline dose–response experiments, the pool of TDP-43-knockdown SH-SY5Y cells were plated as single cells and expanded to obtain a clonal population.

## Depletion of TDP-43 from immortalized human cell lines

SH-SY5Y cells for RT–qPCR validations and western blots were grown in DMEM/F12 containing Glutamax (Thermo) supplemented with 10% FBS (Thermo). For induction of shRNA against TDP-43 cells were treated with 5 μg ml$^{-1}$ doxycyline hyclate (Sigma D9891). After 3 days medium was replaced with Neurobasal (Thermo) supplemented with B27 (Thermo) to induce differentiation. After a further 7 days, cells were collected for protein or RNA. For doxycycline dose response experiments, doxycycline was used at concentrations of 12.5 ng ml$^{-1}$, 18.75 ng ml$^{-1}$, 21 ng ml$^{-1}$, 25 ng ml$^{-1}$, and 75 ng ml$^{-1}$. SH-SY5Y and SK-N-BE(2) cells for RNA-seq experiments were treated with siRNA, as previously described[21].

## RNA sequencing, differential gene expression and splicing analysis

For RNA-seq experiments of i³Neurons, the i³Neurons were grown on 96-well dishes. For collection on day 17, media was completely removed, and wells were treated with tri-reagent (100 μl per well) (Zymo research corporation, cat. no. R2050-1-200). Then 5 wells were pooled together for each biological replicate: control ($n = 4$); TDP-43 knockdown ($n = 3$). To isolate RNA, we used a Direct-zol RNA miniprep kit (Zymo Research Corporation, cat. no. R2052), following manufacturer's instructions including the optional DNAse step. Note: one knockdown replicate did not pass RNA quality controls and so was not submitted for sequencing, resulting in a total of $n = 3$ samples for this condition. Sequencing libraries were prepared with polyA enrichment using a TruSeq Stranded mRNA Prep Kit (Illumina) and sequenced (2 × 75 bp) on an Illumina HiSeq 2500 machine.

Samples were quality trimmed using Fastp with the parameter "qualified_quality_phred: 10", and aligned to the GRCh38 genome build using STAR (v2.7.0f)[38] with gene models from GENCODE v31[39]. Gene expression was quantified using FeatureCounts[40] using gene models from GENCODE v31. Any gene which did not have an expression of at least 0.5 counts per million (CPM) in more than 2 samples was removed. For differential gene expression analysis, all samples were run in the same manner using the standard DESeq2[41] workflow without additional covariates, except for the Klim MNs dataset[9], where we included the day of differentiation. The DESeq2 median of ratios, which controls for both sequencing depth and RNA composition, was used to normalize gene counts. Differential expression was defined at a Benjamini–Hochberg false discovery rate < 0.1. Salmon (v1.5.1)[42] using an index built from GENCODE v34[39] was used to assess the isoform expression of *UNC13B*. Our alignment pipeline is implemented in Snakemake version 5.5.4[43] and available at: https://github.com/frattalab/rna_seq_snakemake.

STAR aligned BAMs were used as input to MAJIQ (v2.1)[33] for differential splicing analysis using the GRCh38 reference genome. A threshold of 10% $\Delta\Psi$ was used for calling the probability of significant change between groups. The results of the deltaPSI module were then parsed using custom R scripts to obtain $\Psi$ and probability of change for each junction. Cryptic splicing was defined as junctions with $\Psi < 5\%$ in control samples, $\Delta\Psi > 10\%$, and the junction was unannotated in GENCODE v31. Our splicing pipeline is implemented in Snakemake version 5.5.4 and available at: https://github.com/frattalab/splicing.

Counts for specific junctions were tallied by parsing the STAR splice junction output tables using bedtools[44]. Splice junction parsing pipeline is implemented in Snakemake version 5.5.4 and available at: https://github.com/frattalab/bedops_parse_star_junctions. $\Psi$ was evaluated using coordinates in Supplementary Table 6:

$$\psi = \frac{\text{Inclusion reads}}{\text{Inclusion reads} + \text{exclusion reads}}$$

Intron retention was assessed using IRFinder[36] with gene models from GENCODE v31.

**Analysis of published iCLIP data**

Cross-linked read files from TDP-43 iCLIP experiments in SH-SY5Y and human neuronal stem cells[22] were processed using iCount v2.0.1.dev implemented in Snakemake version 5.5.4, available at https://github.com/frattalab/pipeline_iclip. Sites of cross-linked reads from all replicates were merged into a single file using iCount group command. Significant positions of cross-link read density with respect to the same gene (GENCODE v34 annotations) were then identified using the iCount peaks command with default parameters.

**Western blot**

SH-SY5Y cells were lysed directly in the sample loading buffer (Thermo NP0008). Lysates were heated at 95 °C for 5 min with 100 mM DTT. If required lysates were passed through a QIAshredder (Qiagen) to shear DNA. Lysates were resolved on 4–12% Bis-Tris Gels (Thermo) or homemade 6% Bis-Tris gels and transferred to 0.45 μm PVDF (Millipore) membranes. After blocking with 5% milk, blots were probed with antibodies (Rb anti-UNC13A (Synaptic Systems 126 103) 1:2,000; Rb anti-UNC13B (abcam ab97924) 1:1,000; Rat anti-Tubulin (abcam ab6161 clone YOL1/34) 1:5,000, Mouse anti-TDP-43 (abcam ab104223 clone 3H8) 1:5,000) for 2 h at room temperature. After washing, blots were probed with HRP conjugated secondary antibodies (Goat anti-Rabbit HRP (Bio-Rad 1706515) 1:10,000; Goat anti-Mouse HRP (Bio-Rad 1706516) 1:10,000; Rabbit anti-Rat HRP (Dako P0450) 1:10,000) and developed with Chemiluminescent substrate (Merck Millipore WBKLS0500) on a ChemiDoc Imaging System (Bio-Rad). Band intensity was measured with ImageJ (NIH version 2.0.0-rc-69).

**RT–qPCR**

RNA was extracted from SH-SY5Y and SK-N-BE(2) cells with a RNeasy kit (Qiagen) or from i³Neurons on day 7 after the initiation of differentiation using a Direct-zol RNA miniprep kit (Zymo Research R2052) following the manufacturer's protocol including the on-column DNA digestion step. RNA concentrations were measured by Nanodrop and 500–1,000 ng of RNA was used for reverse transcription. First strand cDNA synthesis was performed with SSIV (Thermo 18090050), RevertAid (Thermo K1622) or High-Capacity cDNA Reverse Transcription Kit (Thermo 4368814) using random hexamer primers and following the manufacturer's protocol including all optional steps. Gene expression analysis was performed by qPCR using Taqman Multiplex Universal Master Mix (Thermo 4461882) or Taqman Universal PCR Master Mix (Thermo 4304437) and TaqMan assays (UNC13A-Fam Hs00392638_m1, UNC13B-Fam Hs01066405_m1, TDP-43-Vic Hs00606522_m1, GAPDH-Jun assay 4485713, TDP-43-FAM Hs00606522_m1, UPF1-FAM Hs00161289_m1, HPRT1-FAM Hs02800695_m1) on a QuantStudio 5 or a QuantStudio 6 Flex Real-Time PCR system (Applied Biosystems) and quantified using the $\Delta\Delta C_t$ method[45].

**RT–PCR**

RNA extraction and cDNA synthesis was performed as described under 'RT–qPCR'. *UNC13A* CE was amplified with a forward primer in exon 20 (5′-CAAGCGAACTGACAAATCTGCCGTGTCG-3′) and reverse primer in exon 21 (5′-GGCATCGTCACCCTTGGCATCTGG-3′). *UNC13A* intron retention was amplified with a forward primer in exon 30 (5′-ATGCCCTATTCTCCTGCTCC-3′) and a reverse primer that spans the exon 32–33 junction (5′-CATCCAGCTCCTTTCCTCCC-3′). *UNC13B* FSE was amplified with forward primer (5′-TCCGAGCAGTTACCAAGGTT-3′) and reverse primer (5′-GCTGTCAATGCCATAGAGCC-3′). *UNC13B* intron retention was amplified with a forward primer that spans the exon 19–20 junction (5′-CAGGCCATGACGCACTTTG-3′) and a reverse primer in exon 22 (5′-GATTTTAAGTCCTGAAGCCGTTC-3′). For Sanger sequencing, *UNC13A* CE was amplified with exon 19 forward primer (5′-GACATCAAATCCCGCGTGAA-3′) and exon 22 reverse primer (5′-CATTGATGTTGGCGAGCAGG-3′). Amplicons were resolved by agarose gel and the bands corresponding to the short and long form of the cryptic exon were excised and purified (NEB T1030L). The *UNC13A* exon 22 reverse primer (5′-ATACTTGGAGGAGAGGCAGG-3′) was used for sequencing reactions. PCR products were resolved on a TapeStation 4200 (Agilent) and bands were quantified with TapeStation Systems Software v3.2 (Agilent).

**Nonsense-mediated decay inhibition**

For the SH-SY5Y experiment, 10 days after the induction of shRNA against TDP-43 with 1 μg ml⁻¹ doxycyline hyclate (Sigma D9891-1G), cells were treated either with 100 μM CHX or DMSO[46] for 6 h before collecting the RNA with a RNeasy Minikit (Qiagen). Reverse transcription was performed using RevertAid cDNA synthesis kit (Thermo), and transcript levels were quantified by qPCR (QuantStudio 5 Real-Time PCR system, Applied Biosystems) using the $\Delta\Delta C_t$ method[45]. Using RefFinder (https://www.heartcure.com.au/reffinder/), we identified *GAPDH* as the most stable endogenous control across our conditions of interest; the forward GAPDH primer used was 5′-CACCAGGGCTGCTTTTAACT-3′, and the reverse primer was 5′-GACAAGCTTCCCGTTCTCAG-3′. Since it has been shown to undergo NMD[47], *HNRNPL* NMD transcript was used as a positive control. The UNC13B experiment was subsequently performed, following the same method.

For the TDP-43-UPF1 double siRNA knockdown, SH-SY5Y cells were transfected with 40 pM TDP-43 siRNA and either 40 pM control or 40 pM UPF1 siRNAs, and collected after 96 h. Similarly to our experiment with CHX, we used a qPCR approach with GAPDH as endogenous control and *HNRNPL* as positive control. To assess TDP-43 and UPF1 levels, we used the following primers: TDP-43 forward, 5′-GATGGTGTGACTGCAAACTTC-3′;

TDP-43 reverse, 5′-CAGCTCATCCTCAGTCATGTC-3′; *UPF1* forward, 5′-TCGAGGAAGATGAAGAAGACAC-3′, and UPF1 reverse, 5′-TCCGTTGCAGAACCACTTC-3′.

For both experiments in SH-SY5Y cells, *UNC13A* CE was amplified with a forward primer in exon 20 (5′-CAAGCGAACTGACAAATCT GCCGTGTCG-3′) and reverse primer within the CE (5′-CCTGGAAA GAACTCTTATCCCCAGGAACTAGTTTGTTG-3′); UNC13B FSE was amplified with a forward primer in exon 10 (5′-TCCGAGCAGTTAC CAAGGTT-3′) and reverse primer within the FSE (5′-GAAAAGCGAG GAGCCCTTCAG-3′); *STMN2* CE was amplified with a forward primer in exon 1 (5′-GCTCTCTCCGCTGCTGTAG-3′) and reverse primer within the cryptic exon (5′-CTGTCTCTCTCTCTCGCACA-3′); *HNRNPL* NMD transcript was amplified with a forward primer in the NMD-inducing exon (5′-GGTCGCAGTGTATGTTTGATG-3′) and reverse primer in exon 7 (5′-GGCGTTTGTTGGGGTTGCT-3′).

For i³Neuron experiments, iPS cells were infected sequentially, first with either control or a TDP-43 targeting sgRNA in the human pU6-sgRNA EF1A-Bsd-T2A-eGFP backbone, and then second with either a control or UPF1-targeting sgRNA in the bovine pU6-sgRNA EF1A-puro-T2A-BFP backbone for a total of 4 groups: control/control, control/UPF1, TDP43/control, TDP43/UPF1. Two days following each infection, iPS cells were selected with either blasticidin (first infection) or puromycin and blasticidin (second infection) (see 'CRISPRi knockdown in human iPS cells' for further details). iPS cells were then differentiated and neurons were collected in tri-reagent on day 7 after differentiation. Then RNA was isolated and cDNA was made (see 'RT–qPCR'). Then samples were analysed for differential gene expression and splicing by qPCR or PCR followed by Agilent bioanalyzer measurements to assess differences in band sizes resulting from cryptic exon splicing. PCR products were diluted 1:10 in nuclease-free water and resolved on a Bioanalyzer 2100 (Agilent). Bands were quantified with Agilent 2100 Software (Version B.02.08.SI648) using High sensitivity DNA Assay (Version 1.03). UNC13A primers are listed under RT–PCR.

## Quantification of TDP-43, UNC13A and UNC13B using quantitative proteomics

i³Neurons were collected from 6-well plates on day 17 after the initiation of differentiation. One or two wells were pooled for each biological replicate, $n = 6$ for each control and TDP-43-knockdown neurons. To collect cells, wells were washed with PBS, and then SP3 protein extraction was performed to extract intracellular proteins. In brief, we collected and lysed using a very stringent buffer (50 mM HEPES, 50 mM NaCl, 5 mM EDTA 1% SDS, 1% Triton X-100, 1% NP-40, 1% Tween 20, 1% deoxycholate and 1% glycerol) supplemental with cOmplete protease inhibitor cocktail at 1 tablet/10 ml ratio. The cell lysate was reduced by 10 mM dithiothreitol (30 min, 60 °C) and alkylated using 20 mM iodoacetamide (30 min, dark, room temperature). The denatured proteins were captured by hydrophilic magnetic beads, and tryptic on-beads digestion was conducted for 16 h at 37 °C. We injected 1 μg resulting peptides to a nano liquid chromatography for separation, and subsequently those tryptic peptides were analyzed on an Orbitrap Eclipse mass spectrometer coupled with a FAIMS interface using data-dependent acquisition (DDA) and data-independent acquisition (DIA) controlled by Xcalibur v4.3. The peptides were separated on a 120 min LC gradient with 2-35% solvent B (0.1% FA, 5% DSMO in acetonitrile), and FAIMS's compensation voltages were set to −50, −65 and −80. For DDA, we used MS1 resolution at 12,000 and cycle time was selected for 3 s, MS2 fragments were acquired by linear ion trap. For DIA, we used 8 $m/z$ isolation windows (400–1,000 $m/z$ range), cycle time was set to 3 s, and MS2 resolution was set to 30,000. The DDA and DIA MS raw files were searched against Uniprot-Human-Proteome_UP000005640 database with 1% FDR using Proteome Discoverer (v2.4) and Spectronaut (v14.1), respectively. The raw intensity of quantified peptides was normalized by total peptides intensity identified in the same sample. The DDA quantified TDP-43- and UNC13A-derived unique and sharing peptides were parsed out

and used for protein quantification. Specifically, we visualized and quantified the unique peptides of UNC13A using their MS/MS fragment ion intensity acquired by DIA.

## Nanopore sequencing and analysis

RNA from four FTLD-TDP patient samples and four SHSY-5Y samples (two with doxycycline-induced TDP-43 knockdown and two untreated controls) was reverse transcribed using Superscript IV (Thermo Fisher Scientific) using a specific reverse transcription primer following the manufacturer recommendations, but with the volumes halved. Following heat inactivation of the reverse transcriptase, the samples were treated with RNase H (NEB) for 20 min at 37 °C, then diluted fourfold with Phusion HF mastermix (Thermo Fisher Scientific). Two rounds of nested PCR were performed to generate pure amplicons spanning the exon upstream of the CE and the exon downstream of the TDP-43 regulated intron retention, with thermolabile exoI treatment in between (NEB). To ensure complete amplification of amplicons, a 10 min extension time was used (approximately 10× longer than recommended by the manufacturer's protocol). Nanopore-compatible overhangs were then added by PCR and the products were validated by agarose electrophoresis, followed by barcode addition using primers 5–12 from the Nanopore PCR barcoding kit (SQK-PBK004). Following ligase-free rapid adaptor addition (SQK-PBK004) the products were loaded onto and sequenced with a MinION. Demultiplexing and basecalling was performed in real time using the GUPPY basecaller.

Raw fastqs were aligned to a section of chromosome 19 containing the entire UNC13A gene (17690344-17599328; GRCh38.p13) using Minimap2[48] with settings "-ax splice". Downstream analysis was performed using a custom R script (https://github.com/frattalab/unc13a_cryptic_splicing) that quantified alignment to the regions of interest (the CE, the intron retention and their flanking exons), filtering for reads that were long enough to contain both the CE and intron retention so as not to bias the analysis against reads containing both events. Correct assignment was verified manually by visualizing differently classified reads.

Reverse transcription primer, CACATTGCCTGTGCCCTTAAC; nested PCR 1 forward, GACGTGTGGTACAACCTGGA; nested PCR 1 reverse, CACTCTTCAATGTGCGGCTG; nested PCR 2 forward, CTGACAAATCT GCCGTGTCG; nested PCR 2 reverse, GAAGCTGGTAGCAAACACCC; add overhang forward, TTTCTGTTGGTGCTGATATTGC CTGACAAATCT GCCGTGTCG; add overhang reverse, ACTTGCCTGTCGCTCTATCTTC GAAGCTGGTAGCAAACACCC.

## Ribosome profiling

For ribosome-profiling experiments, i³Neurons were grown on 15 cm plates, one plate per biological replicate for control ($n = 4$) and TDP-43-knockdown ($n = 4$) neurons. On day 17, i³Neuron culture medium was replaced 90 min before collecting the neurons to boost translation. Then the medium was removed, cells were washed with cold PBS, PBS was removed and 900 μl of cold lysis buffer (20 mM Tris pH 7.4, 150 mM NaCl, 5 mM MgCl₂, 1 mM DTT (freshly made), 100 μg ml⁻¹ CHX, 1% TX100; 25 U ml⁻¹ Turbo DNase I) was added to each 15 cm plate. Lysed cells were scraped and pipetted into microcentrifuge tubes on ice. Cells were then passed through a 26-gauge needle 10 times, and then centrifuged twice at 19,000g at 4 °C, for 10 min, each time moving the supernatant to a fresh tube. Tubes containing supernatant were flash frozen in liquid nitrogen and stored at −80 °C until processing.

Ribosome footprints from three biological replicates of both TDP-43-knockdown control samples were generated and purified as described, using a sucrose cushion[49] and a customized library preparation method based on revised iCLIP[50]. No ribosomal RNA depletion step was performed, and libraries were sequenced on an Illumina Hi-Seq 4000 machine (SR100). Reads were demultiplexed and adaptor/quality trimmed using Ultraplex[51], then aligned with Bowtie2[52] against a reference file containing abundant ncRNAs that are common contaminants

of ribosome profiling, including rRNAs. Reads that did not pre-map were then aligned against the human genome with STAR[38] and the resulting BAM files were deduplicated with UMI-tools[53]. Multi-mapping reads were discarded and reads 28–30 nt in length were selected for analysis. FeatureCounts[44] was used to count footprints aligning to annotated coding sequences, and DESeq2[41] was used for differential expression analysis, using default parameters in both cases. Periodicity analysis was performed using a custom R script, using transcriptome-aligned bam files. Raw data have been uploaded to E-MTAB-10235.

## ALS and FTD panel genes

To find ALS and FTD 'green' panel genes—those with diagnostic level of evidence that have been approached for testing by NHS in England—'Amyotrophic lateral sclerosis/motor neuron disease (Version 1.33)' and 'Early onset dementia (encompassing fronto-temporal dementia and prion disease) (Version 1.48)' were downloaded from PanelApp[16].

## Genome-wide association study data

Harmonized summary statistics for the latest ALS GWAS[15] were downloaded from the NHGRI-EBI GWAS catalogue[54] (accession GCST005647). Locus plots were created using LocusZoom[55], using linkage disequilibrium values from the 1000 Genomes European superpopulation[56].

## NYGC ALS Consortium RNA-seq cohort

Our analysis contains 377 patients with 1,349 neurological tissue samples from the NYGC ALS dataset, including non-neurological disease controls, FTLD, ALS, FTD with ALS (ALS-FTLD), or ALS with suspected Alzheimer's disease (ALS-AD). Patients with FTD were classified according to a pathologist's diagnosis of FTD with TDP-43 inclusions (FTLD-TDP), or those with FUS or Tau aggregates. ALS samples were divided into the following subcategories using the available Consortium metadata: ALS with or without reported SOD1 or FUS mutations. All non-SOD1 or FUS ALS samples were grouped as ALS-TDP in this work for simplicity, although reporting of postmortem TDP-43 inclusions was not systematic and therefore not integrated into the metadata. Confirmed TDP-43 pathology postmortem was reported for all FTLD-TDP samples.

Sample processing, library preparation, and RNA-seq quality control have been extensively described in previous papers[10,57]. In brief, RNA was extracted from flash-frozen postmortem tissue using TRIzol (Thermo Fisher Scientific) chloroform, and RNA-Seq libraries were prepared from 500 ng total RNA using the KAPA Stranded RNA-Seq Kit with RiboErase (KAPA Biosystems) for ribosomal RNA depletion. Pooled libraries (average insert size: 375 bp) passing the quality criteria were sequenced either on an Illumina HiSeq 2500 (125 bp paired end) or an Illumina NovaSeq (100 bp paired end). The samples had a median sequencing depth of 42 million read pairs, with a range between 16 and 167 million read pairs.

Samples were uniformly processed, including adapter trimming with Trimmomatic and alignment to the hg38 genome build using STAR (2.7.2a)[38] with indexes from GENCODE v30. Extensive quality control was performed using SAMtools[58] and Picard Tools[59] to confirm sex and tissue of origin.

Uniquely mapped reads within the UNC13A locus were extracted from each sample using SAMtools. Any read marked as a PCR duplicate by Picard Tools was discarded. Splice junction reads were then extracted with RegTools[60] using a minimum of 8 bp as an anchor on each side of the junction and a maximum intron size of 500 kb. Junctions from each sample were then clustered together using LeafCutter[61] with relaxed junction filtering (minimum total reads per junction = 30, minimum fraction of total cluster reads = 0.0001). This produced a matrix of junction counts across all samples.

The CE was considered detected in a sample if there was at least one uniquely mapped spliced read supporting either the short CE acceptor or the CE donor. As the long CE acceptor was detected consistently in control cerebellum samples, as part of an unannotated cerebellum-enriched 35 bp exon containing a stop codon between exons 20 and 21 (Extended Data Fig. 10 a, b), we excluded the long CE acceptor for quantification of UNC13A CE Ψ in patient tissue. Only samples with at least 30 spliced reads at the exon locus were included for correlations. In Fig. 4a, only cortical samples that were concordant for genotypes at rs12973192 and rs12608932, had both STMN2 and UNC13A CE detected, and had at least 30 spliced reads at the exon locus were included in the analysis. Cell-type deconvolution was performed using the top 100 most specific marker genes from neurons, astrocytes, oligodendrocytes, endothelial cells and microglia derived by single-cell RNA sequencing[62] with the dtangle[63]. The NYGC ALS Consortium samples presented in this work were acquired through various IRB protocols from member sites and the Target ALS postmortem tissue core and transferred to the NYGC in accordance with all applicable foreign, domestic, federal, state, and local laws and regulations for processing, sequencing, and analyses. The Biomedical Research Alliance of New York (BRANY) IRB serves as the central ethics oversight body for NYGC ALS Consortium. Ethical approval was given and is effective until 22 August 2022. Informed consent has been obtained from all participants.

## Ethics

Brains were donated to the Queen Square Brain Bank (QSBB) for Neurological Disorders (QSBB) and the NeuroResource tissue bank (UCL Queen Square Institute of Neurology). All tissue samples were donated with the full informed consent. Accompanying clinical and demographic data of all cases used in this study were stored electronically in compliance with the 20181998 Data Protection Act and are summarized in Supplementary Table 5. Ethical approval for the study was obtained from the NHS research ethics committee (RNEC) and in accordance with the Human Tissue Authority's codes of practice and standards under license number 12198. We have conformed with all relevant ethical regulations related to informed consent and anonymization of patient data analysed in the manuscript.

## Gene transcript model harmonization

To ensure consistency between RNA-seq, re-analysis of published iCLIP data, and the NYGC ALS Consortium RNA-seq cohort, we confirmed that both the ENSEMBL gene minor version and transcripts for UNC13A and UNC13B are identical between the three GENCODE annotations used across our team.

## BaseScope assay

To validate a BaseScope assay for UNC13A cryptic exons, we first performed the assay in i³Neurons with CRISPRi depletion of control or a non-targeting guide. Neurons were plated on 8-well IBIDI slides, 0.2 million per well and then fixed with 4% paraformaldehyde for 10 min on day 7 after the initiation of differentiation. Neurons were then dehydrated and stored for ~1 week at −20C. Neurons were then rehydrated and pretreated following the recommendations of the RNAscope® Assay for Adherent Cells, using 30% hydrogen peroxide for 8 min and a 1:15 dilution of the RNAscope Protease III. Then the BaseScope v2-RED assay was performed using our UNC13A CE target probe (BA-Hs-UNC13A-O1-1zz-st) according to manufacturer guidelines (Advanced Cell Diagnostics). Following fast red solution, wells were washed 2× with PBS, and incubated overnight at 4 °C in 0.5% Triton-X and 3% BSA containing primary antibodies: rabbit TDP43 (proteintech 12892-1-AP, 1:1,000 dilution) and mouse TUBB3 (Biolegend 801201, 1:5,000 dilution). The next morning, wells were washed three times with PBS and treated with secondary antibodies Alexa Fluor 488 anti-rabbit (Jackson Immuno 711-545-152) and Alexa Fluor 647 anti-mouse (Jackson Immuno 715-605-151), and Hoechst 33342 (Thermo Scientific) at 1:10,000 dilution for 1 h at room temperature. Wells were then washed 3× with PBS and imaged on an inverted spinning disk confocal microscope (Nikon Eclipse T1), using a 60× 1.40 NA oil-immersion objective. Confocal images were then processed in FIJI.

Frozen tissue from the frontal cortex of FTLD-TDP ($n = 9$), FTLD-TAU ($n = 4$) and control ($n = 5$) cases were sectioned at 10 µm thickness onto Plus+Frost microslides (Solmedia). Immediately prior to use, sections were dried at room temperature and fixed for 15 min in pre-chilled 4% paraformaldehyde. Sections were then dehydrated in increasing grades of ethanol and pre-treated with RNAscope hydrogen peroxide (10 min, room temperature) and protease IV (30 min, room temperature). The BaseScope v2-RED assay was performed using our *UNC13A* CE target probe (BA-Hs-UNC13A-O1-1zz-st) according to manufacturer guidelines with no modifications (Advanced Cell Diagnostics,). Sections were nuclei counterstained in Mayer's haematoxylin (BDH) and mounted (VectaMount). Slides were also incubated with a positive control probe (Hs-PPIB-1 ZZ) targeting a common housekeeping gene and a negative control probe (DapB-1 ZZ) which targets a bacterial gene to assess background signal (<1–2 foci per approximately 100 nuclei). Representative images were taken at ×60 magnification.

Hybridized sections were imaged and analysed blinded to disease status. Slides were scanned using an Olympus VS120 slide scanner at ×20 magnification and equal sized (34.5 mm$^2$) regions of interest were extracted from the centre of each section. The total number of red foci, which should identify single transcripts harbouring the *UNC13A* CE event, were manually counted in ImageJ (v1.52p). Foci frequency was background-corrected by subtracting the signal obtained with the negative control probe in the same experiment.

### UNC13A genotypes in the NYGC ALS Consortium
Whole-genome sequencing was carried out for all donors, from DNA extracted from blood or brain tissue. Full details of sample preparation and quality control will be published in a future manuscript. In brief, paired-end 150-bp reads were aligned to the GRCh38 human reference using the Burrows-Wheeler Aligner (BWA-MEM v0.7.15)[64] and processed using the GATK best-practices workflow. This includes marking of duplicate reads by the use of Picard tools[59] (v2.4.1), followed by local realignment around indels, and base quality score recalibration using the Genome Analysis Toolkit[65,66] (v3.5). Genotypes for *rs12608932* and *rs12973192* were then extracted for the samples.

### Targeted RNA-seq
RNA was isolated from temporal cortex tissue of 10 FTLD-TDP and 4 control brains (6 male, 4 female, average age at death 70.6 ± 5.8 yr, average disease duration 10.98 ± 5.9 yr) full metadata provide in Supplementary Table 5. Fifty milligrams of flash-frozen tissue was homogenized in 700 µl of Qiazol (Qiagen) using a TissueRuptor II (Qiagen). Chloroform was added and RNA subsequently extracted following the spin-column protocol from the miRNeasy kit with DNase digestion (Qiagen). RNA was eluted off the column in 50 µl of RNAse-free water. RNA quantity and quality were evaluated using a spectrophotometer.

Purified RNA was reverse transcribed with Superscript IV (Thermo Fisher Scientific) using either sequence-specific primers containing sample-specific barcodes or random hexamers, following the manufacturer recommendations. Unique molecular identifiers (UMIs) and part of the P5 Illumina sequence were added either during first- or second-strand-synthesis (with Phusion HF 2× Master Mix) respectively. Barcoded primers were removed with exonuclease I treatment (NEB; 30 min) and subsequently bead–size selection of RT–PCR products (TotalPureNGS, Omega Biotek). Three rounds of nested PCR using Phusion HF 2× Master Mix (New England Biolabs) were used to obtain highly specific amplicons for the *UNC13A* cryptic, followed by gel extraction and a final round of PCR in which the full length P3/P5 Illumina sequences were added. Samples were sequenced with an Illumina HiSeq 4000 machine (SR100).

Raw reads were demultiplexed, adaptor/quality trimmed and UMIs were extracted with Ultraplex[51], then aligned to the hg38 genome with STAR[38]. To control for mapping biases, a VCF containing rs12973192 was used and alignments that failed to pass WASP filtering were ignored.

Reads were deduplicated via analysis of UMIs with a custom R script; to avoid erroneous detection of UMIs due to sequencing errors, UMI sequences with significant similarity to greatly more abundant UMIs were discarded–this methodology was tested using simulated data, and final results were manually verified. Raw reads for targeted RNA-seq are available at E-MTAB-10237.

Primers used are listed in Supplementary Table 7.

### Splicing reporters
One variant of the *UNC13A* exon 20, intron 20 and exon 21 sequence was synthesized and cloned into a pIRES-EGFP vector (Clontech) by BioCat. The repeat expansion, containing four extra copies of the CATC repeats (ten instead of the six found in the reference genome), was added via Gibson assembly of a PCR-linearized plasmid and a dsDNA insert generated by annealing two synthesized ssDNA oligos (oligos used: unc13mg_bb_FWD: AATGGGTGGGTGGATGAATGGAAGGATG, unc13mg_bb_REV: TCTACCCATCTGACTATCAACAAATTCACC, Unc13_Repeat_add_AntiSense: CCCACCCATTCATCCATTTGTCCATCTGCCTATACATCCATCCATCCATCCATCCATCCATCCATCCATCCATCTACCTATCTACCCATC, Unc13_Repeat_add_Sense: GATGGGTAGATAGGTAGATGGATGGATGGATGGATGGATGGATGGATGGATGGATGTATAGGCAGATGGACAAATGGATGAATGGGTGGG). Plasmids with all four possible combinations of the SNPs were then generated by PCR-based site directed mutagenesis (primers used: healthy_exon_SNP_REV: CTTTTATCTACTCATCACTCATTC, healthy_exon_SNP_FWD: GATGGATGGAGAGATGGG, healthy_intron_SNP_REV: CCATCCATTTTTCGTCTGTC, healthy_intron_SNP_FWD: TTGGATAAATTGATGGGTGGATG. risk_exon_SNP_FWD: CATGGATGGAGAGATGGG, risk_exon_SNP_REV: CTTTTATCTACTCATCACTCATTC). Plasmids were propagated in Stbl3 bacteria (Thermo Fisher Scientific) grown at 30 °C due to the observed instability of the plasmids in DH5alpha cells grown at 37 °C. Similarly, the two UG/UC mutants were generated by PCR-based site directed mutagenesis of the 'healthy' plasmid (primers used: UG_UC_1_F: CGATGGAGAGATGGGTGAG, UG_UC_1_R: ATCCTTTTATCTACTCATCAC, UG_UC_2_F: CGAGAGATGGGTGAGTAC, UG_UC_2_R: ATCCATCCTTTTATCTACTC). All plasmids were verified by Sanger sequencing.

To reduce the impact of sample-to-sample variation on our analysis, we generated (via PCR site-directed mutagenesis) a modified healthy minigene with an alternative primer binding site downstream of the *UNC13A* sequence, before the polyA site, which had no detectable impact on CE splicing. This enabled co-transfection of 1. a minigene featuring a specific combination of variants and 2. the modified control (healthy) minigene into the same population of cells; the cryptic splicing level of each could then be determined by specific RT–PCR amplification of each minigene from the same cDNA, thus ensuring that the observed differences between variants did not simply reflect differences between cells grown in different dishes.

TDP-43 inducible knockdown SH-SY5Y cells were electroporated with 1.5 µg each of the variant and healthy minigene DNA with the Ingenio electroporation kit (Mirus) using the A-023 setting on an Amaxa II nucleofector (Lonza). The cells were then left untreated or treated for 6 days with 1 µg ml$^{-1}$ doxycycline before RNA extraction. Reverse transcription was performed with RevertAid (Thermo Scientific) and cDNA was amplified by nested PCR with minigene-specific primers (5′-TCCTCACTCTCTGACGAGG-3′ and 5′-CATGGCGGTCGACCTAG-3′ or 5′-TGGTCGCCATACTGTCATG-3′ (for the healthy cotransfection control)) followed by *UNC13A*-specific primers 5′-CAAGCGAACTGACAAATCTGCCGTGTCG-3′ and 5′-CGACACGGCAGATTTGTCAGTTCGCTTG-3′. PCR products were resolved on a TapeStation 4200 (Agilent) and bands were quantified with TapeStation Systems Software v3.2 (Agilent).

### Heptamer analysis
Binding enrichment *E*-scores were downloaded from Ray et al. (2013)[27]. Seven-nucleotide sequences that overlapped with either the exonic or

intronic SNPs were extracted using a sliding-window approach. Using a custom R script (https://github.com/frattalab/unc13a_cryptic_splicing/), the average *E*-scores for each RBP were calculated for each set of 7-mers, and the RBPs were ranked by effect size of the SNPs on average *E*-score.

## TDP-43 protein purification

His-tagged human TDP-43 (amino acids 102 to 269) was expressed in BL21-DE3 Gold *Escherichia coli* (Agilent) as previously described[67]. Bacteria were lysed by 2 h of gentle shaking in lysis buffer (50 mM sodium phosphate pH 8, 300 mM NaCl, 30 mM imidazole, 1 M urea, 1% v/v Triton X-100, 5 mM β-mercaptoethanol, with Roche EDTA-free cOmplete protease inhibitor) at room temperature. Samples were centrifuged at 16,000 rpm in a Beckman 25.50 rotor at 4 °C for 10 min, and the supernatant was clarified by vacuum filtration (0.22 μm).

The clarified lysate was loaded onto a 5 ml His-Trap HP column (Cytiva) equilibrated with buffer A (50 mM sodium phosphate pH 8, 300 mM NaCl, 20 mM imidazole) using an AKTA Pure system, and eluted with a linear gradient of 0-100% buffer B (50 mM sodium phosphate pH 8, 300 mM NaCl, 500 mM imidazole) over 90 column volumes. The relevant fractions were then analysed by SDS–PAGE and then either extensively dialysed (3.5 kDa cutoff) against isothermal titration calorimetry (ITC) buffer (50 mM sodium phosphate pH 7.4, 100 mM NaCl, 1 mM TCEP) at 4 °C, or flash frozen in liquid nitrogen.

## Isothermal titration calorimetry

RNAs with sequences 5′-AAGGAUGGAUGGAG-3′ (CE SNP healthy), 5′-AAGCAUGGAUGGAG-3′ (CE SNP risk), 5′-AAAAAUGGAUGGUUGGAU-3′ (intron SNP healthy) and 5′-AAAAAUGGAUGGGUGGAU-3′ (intron SNP risk) were synthesized by Merck, resuspended in Ultrapure water, then dialysed against the same stock of ITC buffer used for TDP-43 dialysis (above) overnight at 4 °C using 1 kDa Pur-a-lyzer tubes (Merck). Protein and RNA concentrations after dialysis were calculated by A280 and A260 absorbance respectively. ITC measurements were performed on a MicroCal PEAQ-ITC calorimeter (Malvern Panalytical). Titrations were performed at 25 °C with TDP-43 (9.6–12 μM) in the cell and RNA (96–120 μM) in the syringe. Data were analysed using the MicroCal PEAQ-ITC analysis software using nonlinear regression with the One set of sites model. For each experiment, the heat associated with ligand dilution was measured and subtracted from the raw data.

## iCLIP of SH-SH5Y and minigene-transfected HEK 293T cells

SH-SY5Y cells were grown to 80% confluence in two 10 cm dishes. HEK 293T cells were grown to 80% confluence and transfected with either the 2× healthy or 2× risk minigenes using Lipofectamine 3000 (Thermofisher Scientific). Each replicate consisted of 2× 3.5 cm dishes, with two replicates per sample, for eight dishes total. Plasmid (1.25 μg) was used for each dish, measured via Nanodrop (Thermo Fisher Scientific), combined with 2.5 μl of Lipofectamine 3000 and P3000 reagent diluted in 250 μl (2×125 μl) of Opti-MEM I following the manufacturer protocol (Thermo Fisher Scientific). Cells were UV crosslinked on ice and subjected to iCLIP analysis following the iiCLIP protocol[50]. In brief, medium RNase I was added to cell lysate for RNA fragmentation. Immunoprecipitations were performed with 4 μg of TDP-43 antibody ((Proteintech, Rabbit anti-TDP-43 cat. no. 10782-2-AP) coupled with 100 μl of protein A or G dynabeads (for SH-SY5Y or HEK 293T, respectively) per sample. The complexes were then size-separated with SDS–PAGE and visualized by Odyssey scanning. cDNA was synthesized with Superscript IV Reverse Transcriptase (Life Technologies). cDNA was then circularized. After PCR amplification, libraries were removed from primers with Ampure beads and QCed for sequencing. Libraries were sequenced on an Illumina HiSeq4000 machine (SR100).

For SH-SH5Y iCLIP, downstream analysis was performed with the iMAPS server. For data from HEK 293T cells, after demultiplexing the reads with Ultraplex, we initially aligned to the human genome using STAR[38], which showed that >5% of uniquely aligned reads mapped solely to the genomic region that is contained in the minigene. Given the high prior probability of reads mapping to the minigene, we therefore instead used Bowtie2 to align to the respective minigene sequences alone, thus minimizing mis-mapping biases that could be caused by the SNPs[52] with settings "--norc --no-unal --rdg 50,50 --rfg 50,50 --score-min L,−2,−0.2 --end-to-end -N 1", then filtered for reads with no alignment gaps, and length >25 nt. Due to the exceptional read depth and high library complexity, we did not perform PCR deduplication to avoid UMI saturation at signal peaks. All downstream analysis was performed using custom R scripts; to avoid biases due to differing transfection efficiencies, crosslink densities were normalized by the total number of minigene crosslinks for each sample. Raw data are available at E-MTAB-10297.

## Reporting summary

Further information on research design is available in the Nature Research Reporting Summary linked to this paper.

## Data availability

A minimum dataset to reproduce analyses is freely available at https://github.com/frattalab/unc13a_cryptic_splicing/tree/main/data. RNA-seq data for i3Neurons, SH-SY5Y and SK-N-BE(2)[a] are available through the European Nucleotide Archive (ENA) under accession PRJEB42763. NYGC ALS Consortium RNA-seq: RNA-seq data generated through the NYGC ALS Consortium in this study can be accessed via the NCBI GEO database (GSE137810, GSE124439, GSE116622 and GSE153960). To request immediate access to new data generated by the NYGC ALS Consortium and for samples provided through the Target ALS Postmortem Core, complete a genetic data request form at CGND_help@nygenome.org. NYGC ALS Consortium genotypes for the common SNPs in this study *rs129731921* and *rs12608932* are available at https://github.com/frattalab/unc13a_cryptic_splicing/blob/main/data/nygc_junction_information.csv. Source data are provided with this paper.

## Code availability

Analysis code and data to reproduce figures are freely available at https://github.com/frattalab/unc13a_cryptic_splicing/. The tool for demultiplexing iCLIP reads is freely available at https://github.com/ulelab/ultraplex. Snakemake pipelines to perform RNA-seq alignment, splicing and parsing splice junction files are freely available at https://github.com/frattalab/rna_seq_snakemake/, https://github.com/frattalab/splicing/ and https://github.com/frattalab/bedops_parse_star_junctions/. The Snakemake pipeline for analysing publicly available iCLIP is available at https://github.com/frattalab/pipeline_iclip.

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

**Acknowledgements** We thank F. Allain for the His-tagged TDP-43 plasmid; C. Stuani, F. Weissmann, M. Watson and K. Stott for guidance on TDP-43 purification and ITC; A. Isaacs and P. Whiting for support with shRNA experiments; N. Seyfried for input on proteomic experiments; and J. Vargas for his scientific insights and engaging conversations. This work was supported by grants from UK Medical Research Council MR/R005184/1 (E.M.C.F. and P.F.), FC001002 (J.U.); NIH U54NS123743 (P.F.); UK Motor Neurone Disease Association (P.F.); Rosetrees Trust (P.F. and A.G.); Chan Zuckerberg Initiative (M.E.W.); The Robert Packard Center for ALS Research (M.E.W., P.F. and E.M.C.F.); AriSLA (E.B.); Alzheimers Society (A.G.); NIH T32 GM136577 (S.S.); NIH National Institute of Aging R56-AG055824 and U01-AG068880 (J.H. and T.R.) European Union's Horizon 2020 research and innovation programme 835300 (J.U.); Cancer Research UK FC001002 (J.U.); Wellcome Trust FC001002 (J.U.); Collaborative Center for X-linked Dystonia-Parkinsonism (W.C.L. and E.M.C.F.). P.F. is supported by a UK Medical Research Council Senior Clinical Fellowship and Lady Edith Wolfson Fellowship (MR/M008606/1 and MR/S006508/1), the UCLH NIHR Biomedical Research Centre; M.E.W. and S.E.K.-H. are supported by the NIH Intramural Research Program of the National Institutes of Neurological Disorders and Stroke; O.G.W. is supported by a Wellcome Trust Studentship; M.Z. is supported by the Neurological Research Trust; S.C. is supported by NIH Intramural Research Program of the Eunice Kennedy Shriver National Institute of Child Health and Human Development; A.B. is supported by Eisai and the Wolfson Foundation; S.E.K.-H. is supported by a Brightfocus Foundation postdoctoral research fellowship; T.L. is supported by an Alzheimer's Research UK senior fellowship; G.S. is supported by a Wellcome Trust Investigator Award (107116/Z/15/Z) and UK Dementia Research Institute Foundation award (UKDRI-1005); M.S. is supported by a UKRI Future Leaders Fellowship (MR/T042184/1); S.B.-S. is supported by a UK Motor Neurone Disease Association and Masonic Charitable Foundation PhD Studentship (893-792); M.H. is supported by a Lady Edith Wolfson Senior Non-Clinical Fellowship (959-799); S.S. is supported by the NIH Oxford–Cambridge Scholars Program.

**Author contributions** Conceptualization: A.-L.B., O.G.W., M.J.K., S.E.K.-H., J.H., M.E.W. and P.F. Data curation: A.-L.B., O.G.W., M.Z., W.C.L. and S.B.-S. Formal analysis: A.-L.B., O.G.W., M.J.K., M.Z., W.C.L., S.B.-S., A.B., M.H., E.K.G., S.S., J.F.R., S.L.C. and D.R. Funding acquisition: P.F., M.E.W. and E.B. Investigation: A.-L.B., O.G.W., M.J.K., S.E.K.-H., M.Z., W.C.L., F.C.Y.L., L.M., Y.A.Q., S.B.-S., A.B., A.G., M.H., E.K.G., S.S., J.F.R., S.L.C., D.R., E.K.G. and S.L.C. Methodology: A.-L.B., O.G.W., M.J.K., S.E.K.-H., J.H., M.E.W. and P.F. Project administration: P.F. and M.E.W. Resources: H.P., T.L., E.B., D.F. and J.N. Software: A.-L.B., O.G.W., M.Z., S.B.-S., J.H. and M.J.K. Supervision: P.F., M.E.W., J.H., J.U., M.S., T.R., T.L., E.M.C.F., G.S. and T.F. Visualization: A.-L.B., O.G.W., M.J.K., W.C.L. and S.E.K.-H. Writing, original draft: A.-L.B., O.G.W., M.J.K., M.E.W. and P.F. Writing, review and editing: S.E.K.-H., W.C.L., E.B., J.U., J.H., M.E.W., P.F. A.-L.B., O.G.W. M.J.K. and S.E.K.-H. contributed equally; therefore each may place their name first in author order when referencing this manuscript in personal communications.

**Competing interests** A patent application related to this work has been filed. The technology described in this work has been protected in the patent PCT/EP2021/084908 and UK patent 2117758.9 (patent applicant, UCL Business Ltd and NIH; status pending), in which A.-L.B., O.G.W., M.J.K., S.E.K.-H., M.E.W. and P.F. are named as inventors. The other authors declare no competing interests.

**Additional information**
**Correspondence and requests for materials** should be addressed to Andrea Malaspina or Pietro Fratta.

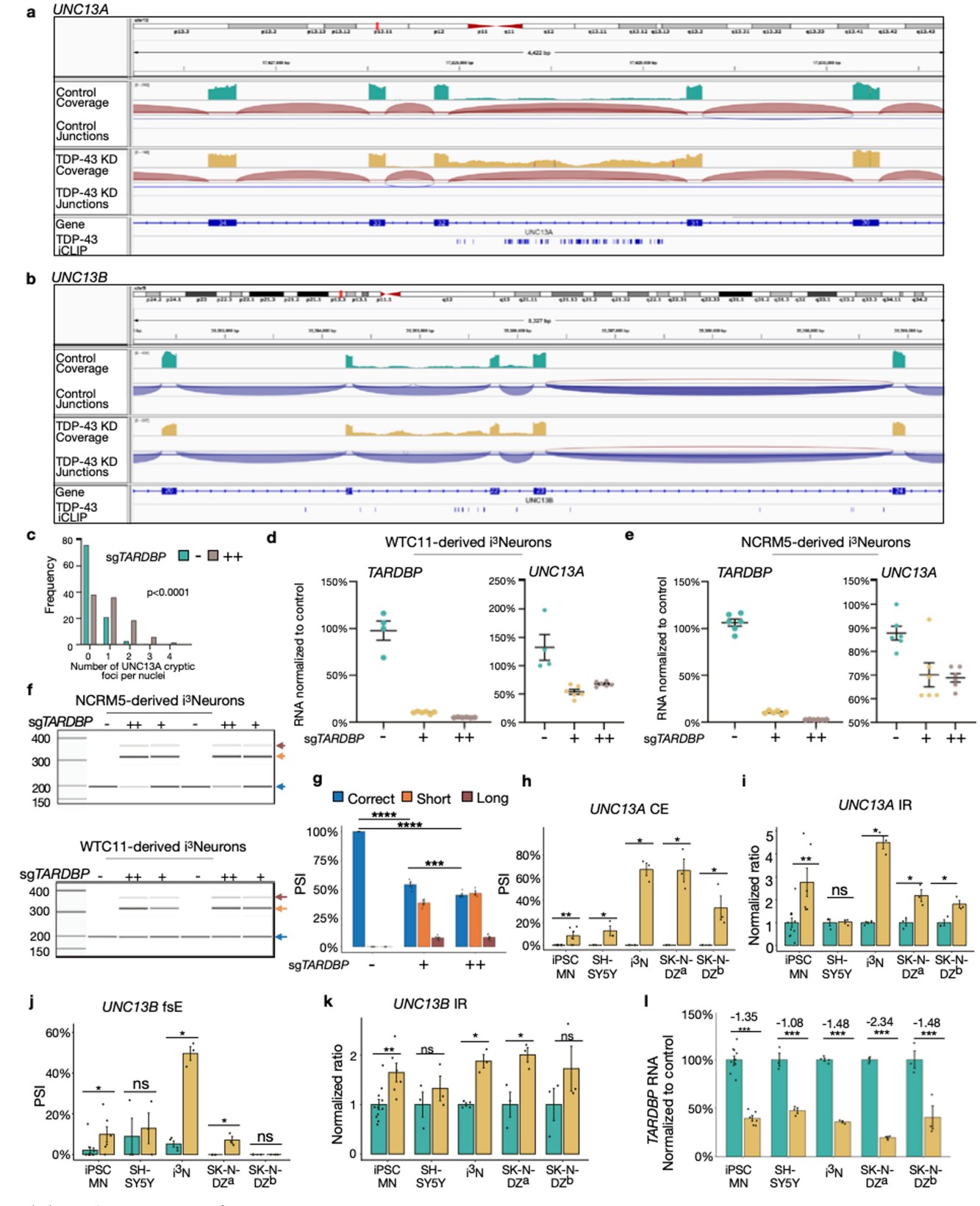

**Extended Data Fig. 1** | See next page for caption.

**Extended Data Fig. 1 | *UNC13A* and *UNC13B* are misspliced after TDP-43 knockdown across neuronal lines.** (**a, b**) RNA-seq traces from IGV[34] of representative samples from control (top) and *TARDBP* KD (bottom) in i[3]Neurons showing intron retention in *UNC13A* (A) (mean 4.50 ± 1.50 increased IR in KD) and *UNC13B* (mean 1.86 ± 0.63 increased IR in KD))(B), overlaid with published TDP-43 iCLIP peaks[22] (**c**) Histogram showing number of basescope cryptic foci per nuclei in control (blue) and TDP-43 KD (grey) in WTC11-derived i[3]Neurons, p < 0.0001 unpaired t-test. (**d, e**) RT-qPCR levels of *TARDBP* and *UNC13A* with a non-targeting control sgRNA (sgTARDBP −), an intermediate TDP-43 KD (sgTARDBP +) or a higher TDP-43 KD (sgTARDBP ++) in WTC11-derived (**d**) and NCRM-5-derived i[3]Neurons (**e**). n = 4 biological replicates sgTARDBP − (**d**), n = 6 biological replicates sgTARDBP − (**e**), sgTARDBP + (**d, e**) and ++ (**d, e**). plotted as means ± SEM. (**f**) Representative images of *UNC13A* CE RT-PCR products (**g**) Quantification of the lower gel in (**f**) plotted as means ± SEM, n = 6 biological replicates non-targeting control sgRNA (sgTARDBP −), sgTARDBP +, sgTARDBP ++. Upper gel is quantified in Fig. 1h. One-way ANOVA with multiple comparisons. (**h−k**) Expression of TDP-43 regulated splicing in *UNC13A*(**h, i**) and *UNC13B*(**j, k**) across neuronal datasets[9,21] in control (blue) and TDP-43 KD (yellow). Intron retention (IR)(**i, k**) and CE and fsE PSI (**h, j**) significantly increase after TDP-43 depletion in most experiments, Wilcoxon test (**l**) Relative gene expression levels for *TARDBP* across neuronal datasets[9,21]. Normalized RNA counts are shown as relative to control mean. Numbers show log₂ fold change calculated by DESeq2. Significance shown as adjusted p-values from DESeq2. For (**h−l**) biological replicates are: iPSC MN Ctrl KD n = 12, TDP-43 KD n = 6; i[3]N Ctrl KD n = 4, TDP-43 KD n = 3; SH-SY5Y, SK-N-BE(2)[a], and SK-N-BE(2)[b] Ctrl KD n = 3, TDP-43 KD n = 3, Significance levels reported as * (p < 0.05) ** (p < 0.01) *** (p < 0.001) **** (p < 0.0001).

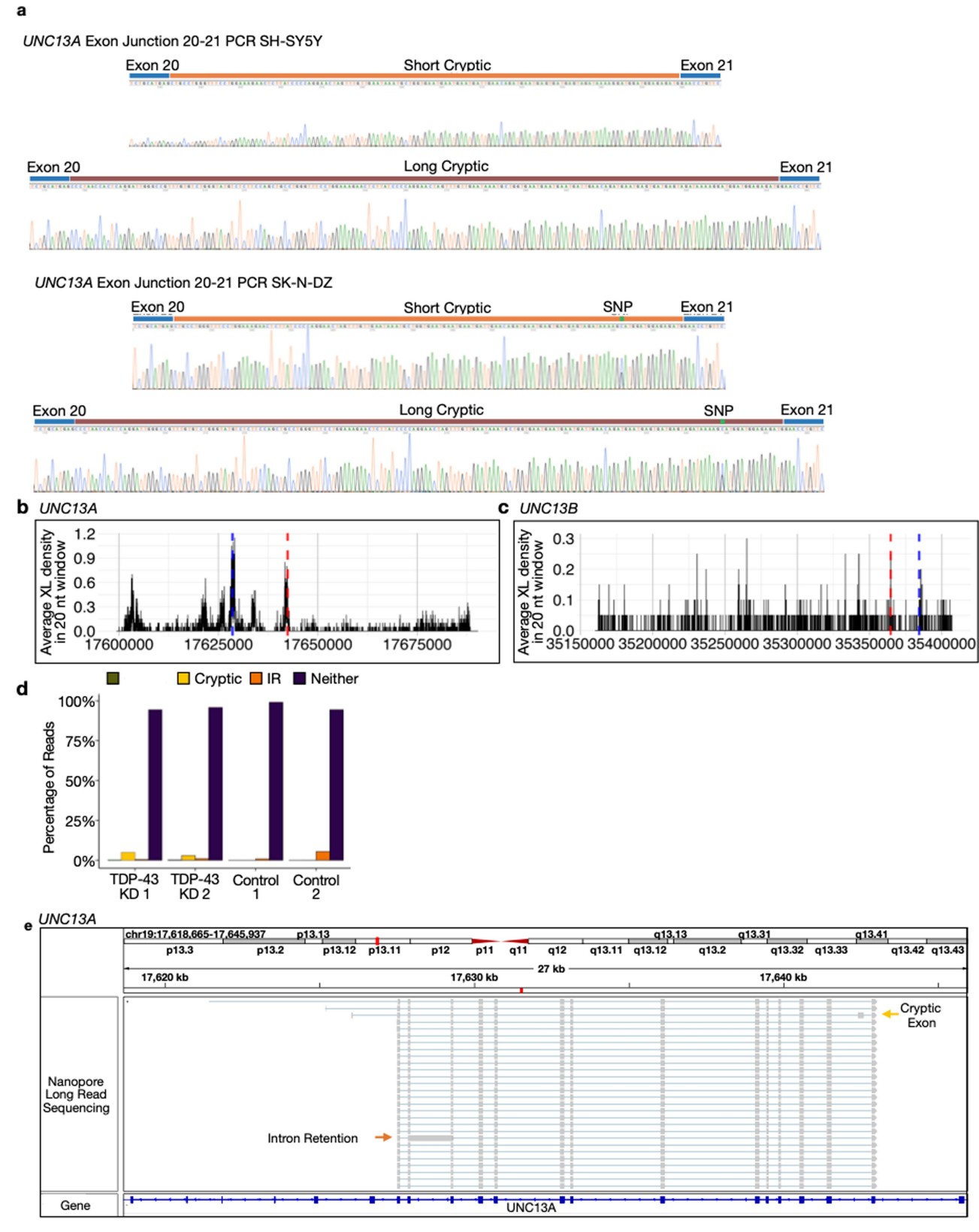

**Extended Data Fig. 2 | Validation of *UNC13A* and *UNC13B* misplicing after TDP-43 KD across multiple neuronal cell lines.** Targeted nanopore sequencing reveals *UNC13A* CE and IR events occur largely independently in-vitro. (**a**) Sanger sequencing of cryptic bands in both SH-SY5Y and SK-N-BE(2) cells confirm the CE splice junctions. (**b, c**) Crosslink density across *UNC13A* (chr19) (b) and *UNC13B* (chr9) (c) genomic loci from novel iCLIP on endogenous TDP-43 in SH-SHY5Y cells. Crosslink densities for both genes show peaks at the CE/fsE (red) and retained introns (blue). Coordinates shown in hg38. (**d**) Percentage of all targeted *UNC13A* long reads in SH-SY5Y cells containing either neither CE nor IR, both, or either CE or IR. Most reads in both control and TDP-43 KD contain neither event, and while IR event is present in controls, CE is only detected in TDP-43 KD. (**e**) Representative trace in TDP-43 KD of *UNC13A* targeted long reads showing transcript containing either the CE or IR, and transcripts with neither.

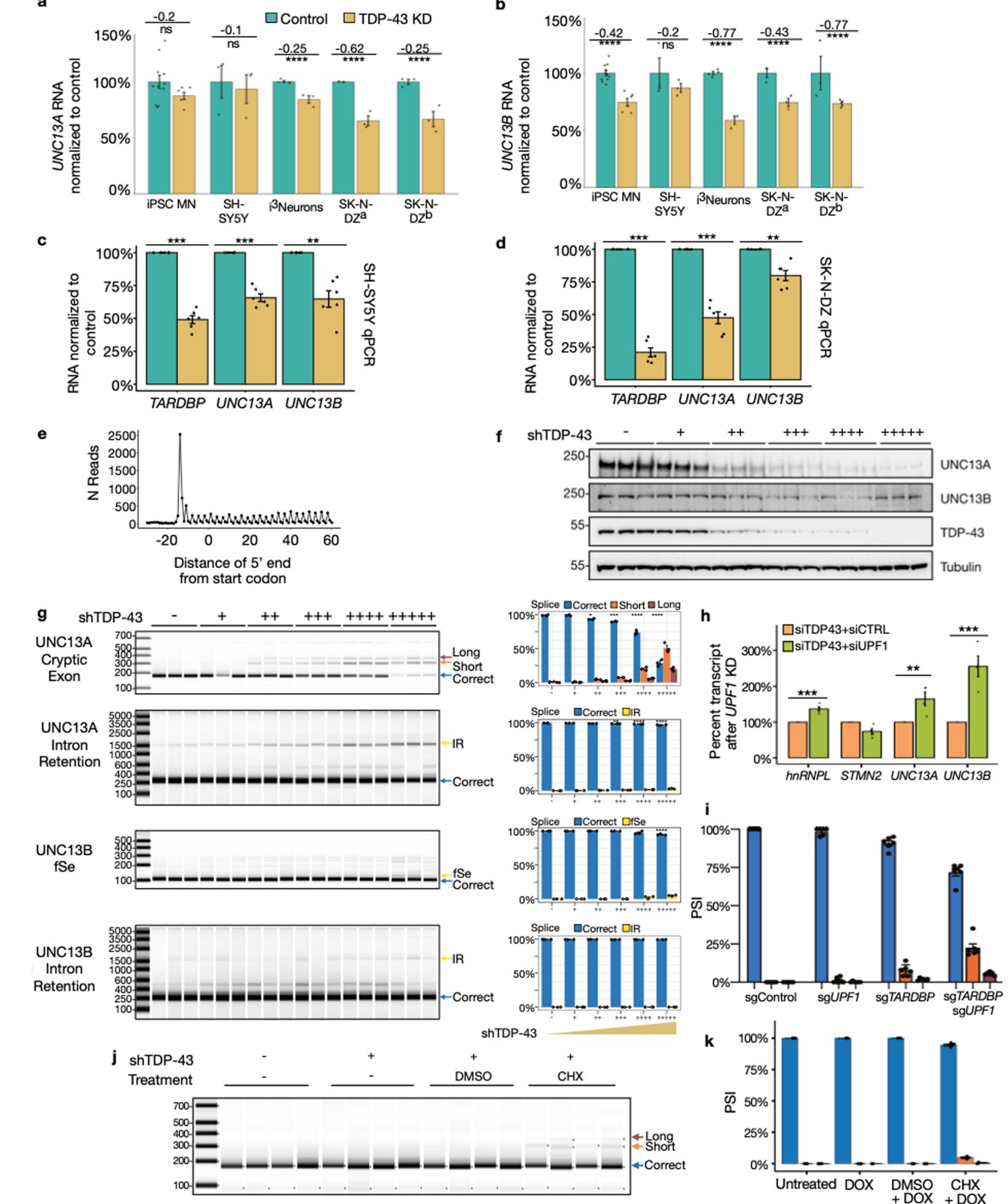

**Extended Data Fig. 3** | See next page for caption.

**Extended Data Fig. 3 | Reduction of *UNC13A* and *UNC13B* after TDP-43 knockdown correlates with TDP-43 levels and is caused by nonsense-mediated decay.** Relative gene expression levels for *UNC13A* (**a**) and *UNC13B* (**b**) after TDP-43 knockdown across neuronal cell lines[9,21]. Normalized RNA counts are shown as relative to control mean. Numbers show log fold change calculated by DESeq2. Significance shown as adjusted p-values from DESeq2. Number of replicates as in Extended data Fig. 1 H-L (**c, d**) RT-qPCR analysis shows *TDP-43, UNC13A* and *UNC13B* gene expression is reduced by *TARDBP* shRNA knockdown in both SH-SY5Y and SK-N-BE(2) human cell lines. Graphs represent the means ± SEM, n = 6 biological replicates, one sample t-test. (**e**) The 5' ends of 29 nt reads relative to the annotated start codon from a representative ribosome profiling dataset (TDP-43 KD replicate B). As expected, we detected strong three-nucleotide periodicity, and a strong enrichment of reads across the annotated coding sequence relative to the upstream untranslated region. (**f**) UNC13A, UNC13B, and TDP-43 protein levels, measured by Western Blot, with varying levels of DOX-inducible TDP-43 knockdown in SH-SY5Y cells. Tubulin is used as endogenous control, n = 3. For gel source data, see Supplementary Figure 1. (**g**) Quantification of RT-PCR products from the transcripts containing *UNC13A* CE, *UNC13A* intron retention, *UNC13B* fsE, and *UNC13B* intron retention, with varying levels of DOX-inducible TDP-43 knockdown in SH-SY5Y cells. Graphs represent the means ± SEM n = 3 biological replicates. (**h**) UPF1 siRNA knock-down led to the rescue of *hnRNPL* (positive control), *UNC13A*, and *UNC13B* transcripts, but not *STMN2*. Graphs represent the means ± SEM, n = 4 biological replicates, one-sample t-test. (**I**) *UNC13A* CE containing-transcript PSI is increased after UPF1 knockdown in i³Neurons. Graphs represent the means ± SEM, n = 6 biological replicates. (**j**) RT-PCR products from *UNC13A* in the setting of mild TDP-43 knockdown ("+", as for Figure 2C and S4G) with the addition of either DMSO (control) or CHX (NMD inhibition). (**k**) Quantification of (**j**) Graphs represent the means ± SEM, n = 4 biological replicates. Significance levels reported as * (p < 0.05) ** (p < 0.01) *** (p < 0.001) **** (p < 0.0001).

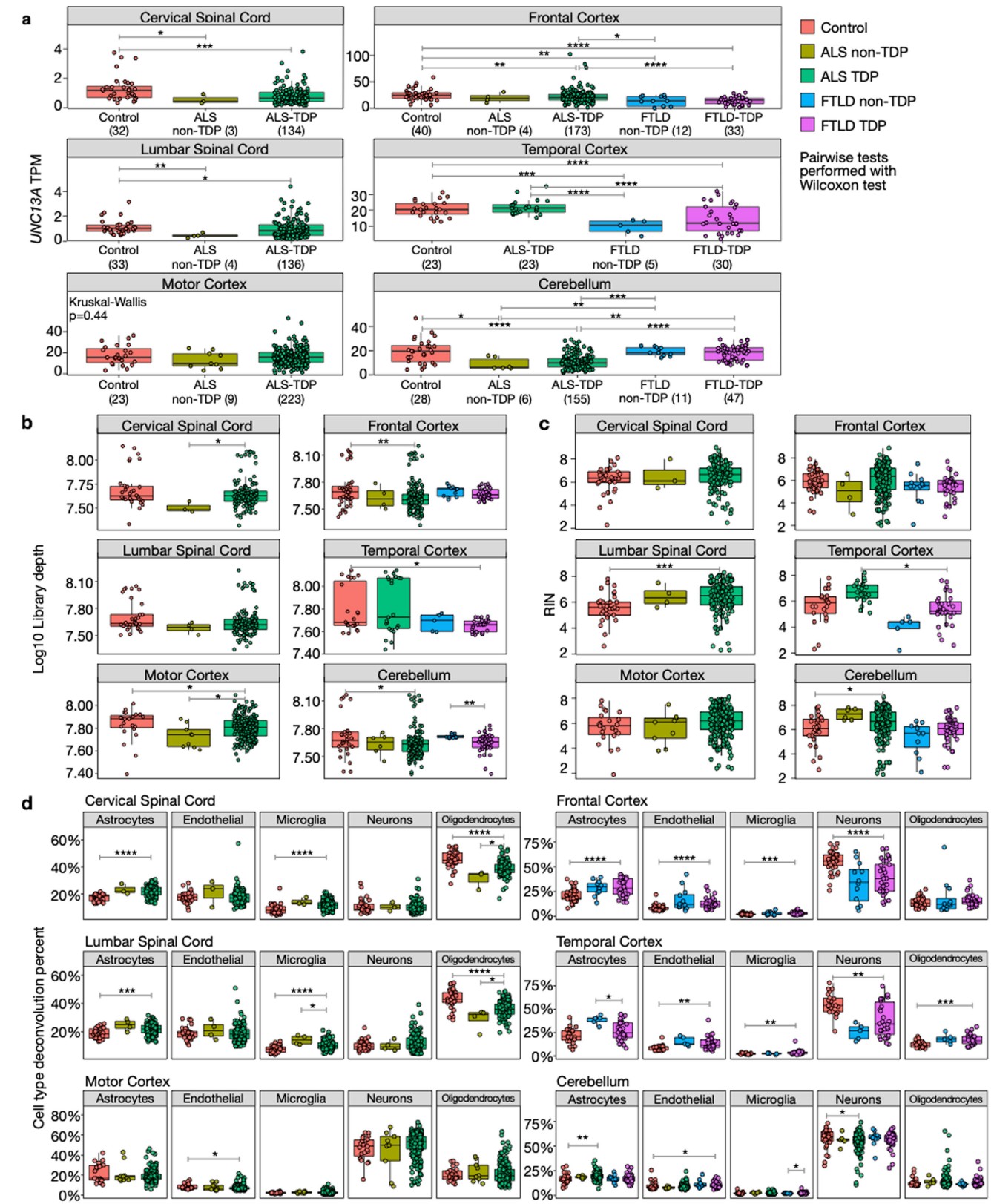

**Extended Data Fig. 4 | Sample technical factors in NYGC tissue samples do not vary in a systematic way. (a)** *UNC13A* expression across tissues and disease subtypes in the NYGC ALS Consortium RNA-seq dataset. Expression normalised as transcripts per million (TPM). Cortical regions have noticeably higher *UNC13A* expression than the spinal cord. **(b)** total RNA-seq library size (log10 scaled) **(c)** RNA integrity score (RIN) **(d)** Cell type decomposition across NYGC ALS Consortium RNA-seq dataset. While there are differences between tissues and disease-subtypes on these technical factors, specificity of *UNC13A* CE detection to tissues presumed to contain TDP-43 proteinopathy cannot be explained by these technical factors. Box plots **(a–d)**: boundaries 25–75th percentiles; midline, median; whiskers, Tukey style. Wilcoxon test, significance levels reported as * (p < 0.05) ** (p < 0.01) *** (p < 0.001) **** (p < 0.0001).

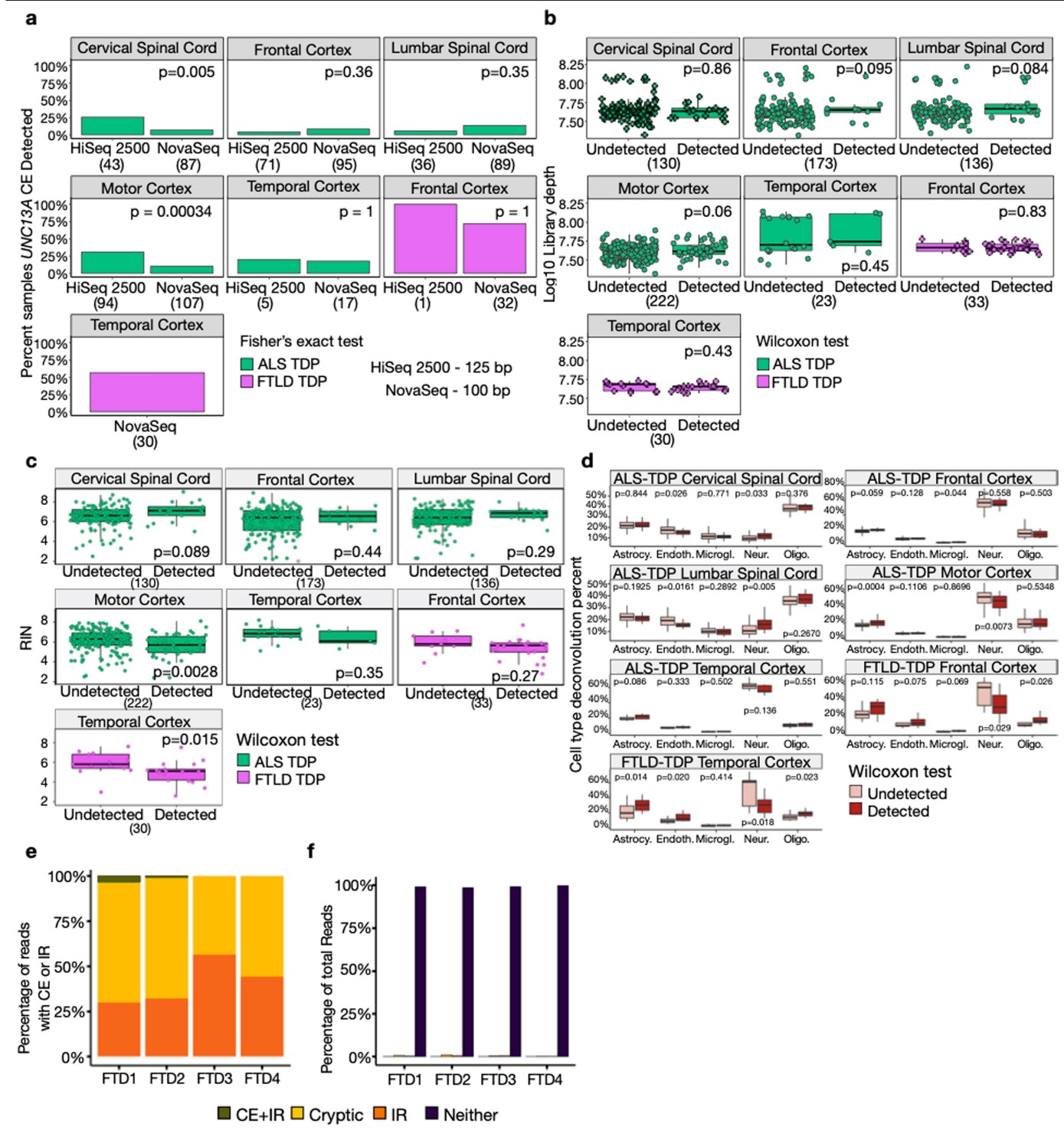

**Extended Data Fig. 5 | Differences in sample technical factors where UNC13A CE was detected and undetected vary between cortical and spinal tissues.** Targeted long reads in FTLD frontal cortex show that UNC13A CE and IR occur independently in-vivo. (**a**) Detection rate of UNC13A CE across tissues by RNA sequencing platform and read length. UNC13A CE was more likely to be detected in cervical spinal cord and motor cortex when sequenced on machines with 125 bp compared to 100 bp. (**b**) No significant differences in total RNA-seq library size (log10 scaled). (**c**) RNA integrity score (RIN) was significantly lower in motor and temporal cortices in samples where UNC13A was detected. (**d**) Cell type decomposition revealed that samples with UNC13A CE detected had a higher proportion of neurons in cervical and lumbar spinal cord, whereas in frontal, temporal, and motor cortex samples with UNC13A CE detected had a lower proportion of neurons, and in motor and temporal cortex samples with UNC13A CE detected had a higher proportion of astrocytes. Astrocy. - Astrocytes, Endothi. - Endothelial, Microgl. - Microglia. Neur. - Neurons, Oligiodendr. - Oligiodendrycytes. P-values shown are from Fisher's exact test (**a**) or Wilcoxon test (**b**–**d**). N tissue samples show below in brackets. Box plots (**a**–**d**): boundaries 25-75th percentiles; midline, median; whiskers, Tukey style. (**e**) Percentage of targeted UNC13A long reads with TDP-43 regulated splice events that contain either both, CE, or IR in four in FTLD frontal cortices. (**f**) Percentage of all targeted UNC13A long reads in (**a**) containing neither CE nor IR, both, or either CE or IR.

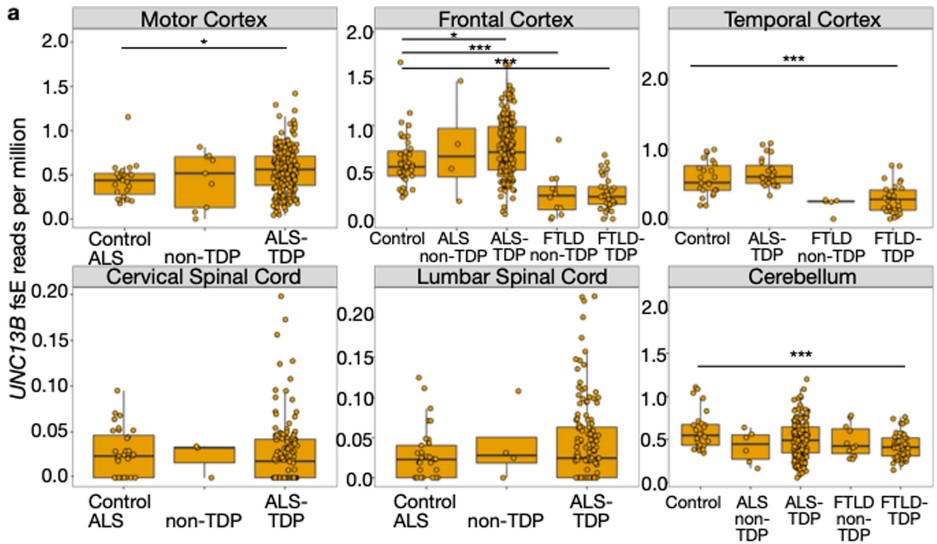

**Extended Data Fig. 6 | Expression of shorter *UNC13B* isoform in human neuronal tissue masks detection of *UNC13B* fsE across NYGC tissue samples.** (**a**) Expression of splice junction reads supporting the *UNC13B* fsE across tissues and disease subtypes. Junction counts are normalised by library size in millions (junctions per million). Expression of UNC13B fsE is present across controls and ALS/FTLD-non-TDP tissues. Wilcoxon test, significance levels reported as * (p < 0.05) ** (p < 0.01) *** (p < 0.001) **** (p < 0.0001). (**b**) Diagram showing three of the *UNC13B* transcripts, including the APPRIS[35]

principal isoform *UNC13B-207* (blue), the NMD sensitive isoform *UNC13B-208* (green), and the shorter isoform *UNC13B-210* which shares the fsE (light green highlight) and one of the splicing junctions supporting the fsE as *UNC13B-208*. (**c**) Expression of three *UNC13B* isoforms across NYGC cohort and in the five in vitro TDP-43 knockdowns experiments[9,21]. *UNC13B-210* is expressed across in vivo human tissues, whereas there is almost no expression of *UNC13B-210* in any of the in vitro experiments. Box plots (**a, c**): boundaries 25–75th percentiles; midline, median; whiskers, Tukey style.

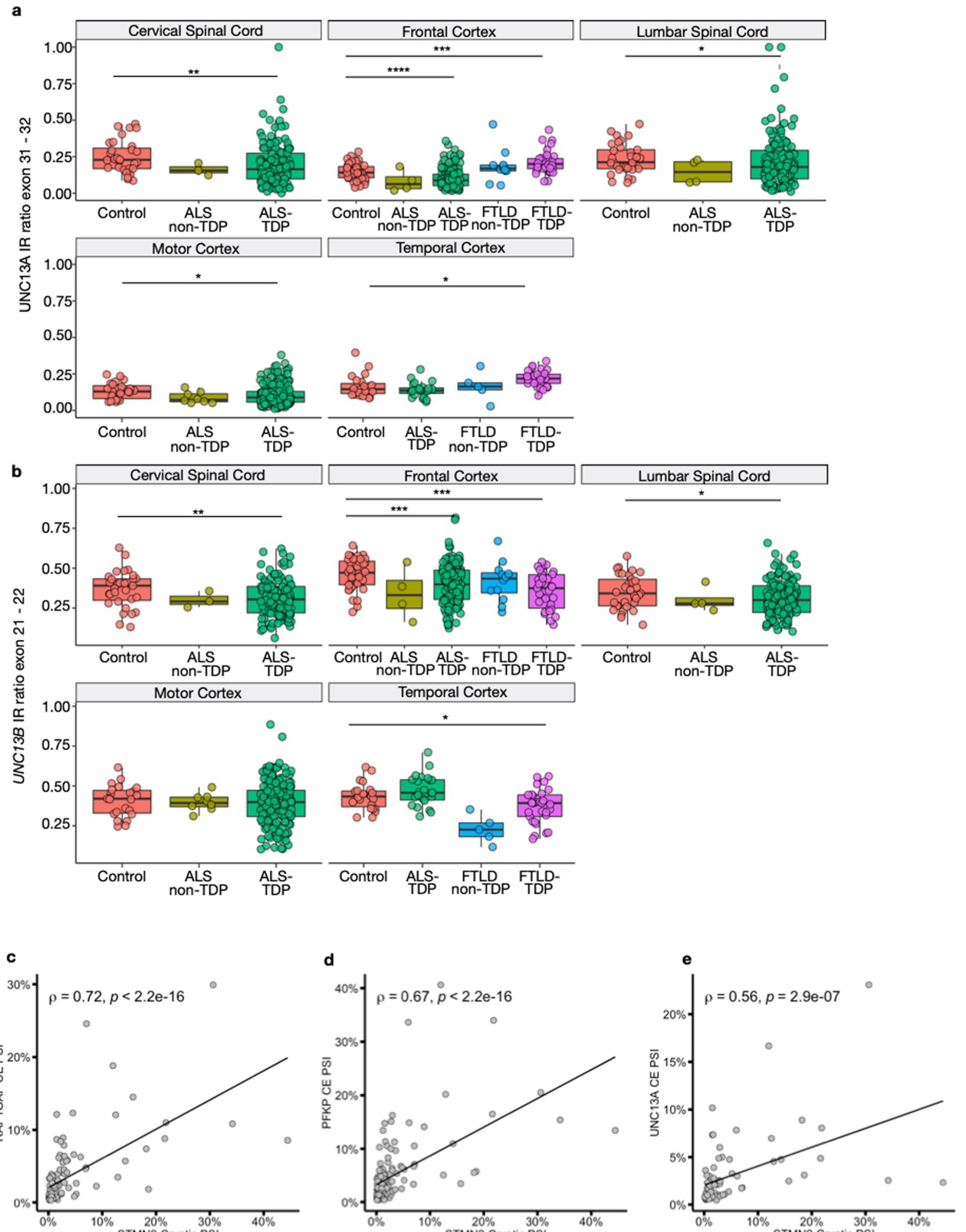

**Extended Data Fig. 7 | TDP-43 regulated *UNC13A* and *UNC13B* introns are expressed across human neuronal tissues in NYGC tissue samples.** *STMN2* CE PSI correlates with TDP-43 regulated cryptics across NYGC RNA-seq dataset. IRratio[36] in *UNC13A* exon 31–32 (**a**) and *UNC13B* exon 21–22 (**b**) across NYGC tissue samples. *UNC13A* IR was lower in ALS-TDP cases than in controls in cervical spinal, frontal and motor cortices, and higher in FTLD-TDP cases than controls in frontal and temporal cortices. Possibly this reflects differences in the effects of cell type composition in disease state. Box plots (**a, b**): boundaries 25–75th percentiles; midline, median; whiskers, Tukey style.. Wilcoxon test, significance levels reported as * (p < 0.05) ** (p < 0.01) *** (p < 0.001) **** (p < 0.0001). (**c–e**) Correlation in ALS/FTLD-TDP cortex between *RAP1GAP* CE (**c**), *PFKP* CE (**d**), and *UNC13A* CE (**e**) with *STMN2* CE PSI in patients with at least 30 spliced reads across the CE locus. Spearman's correlation.

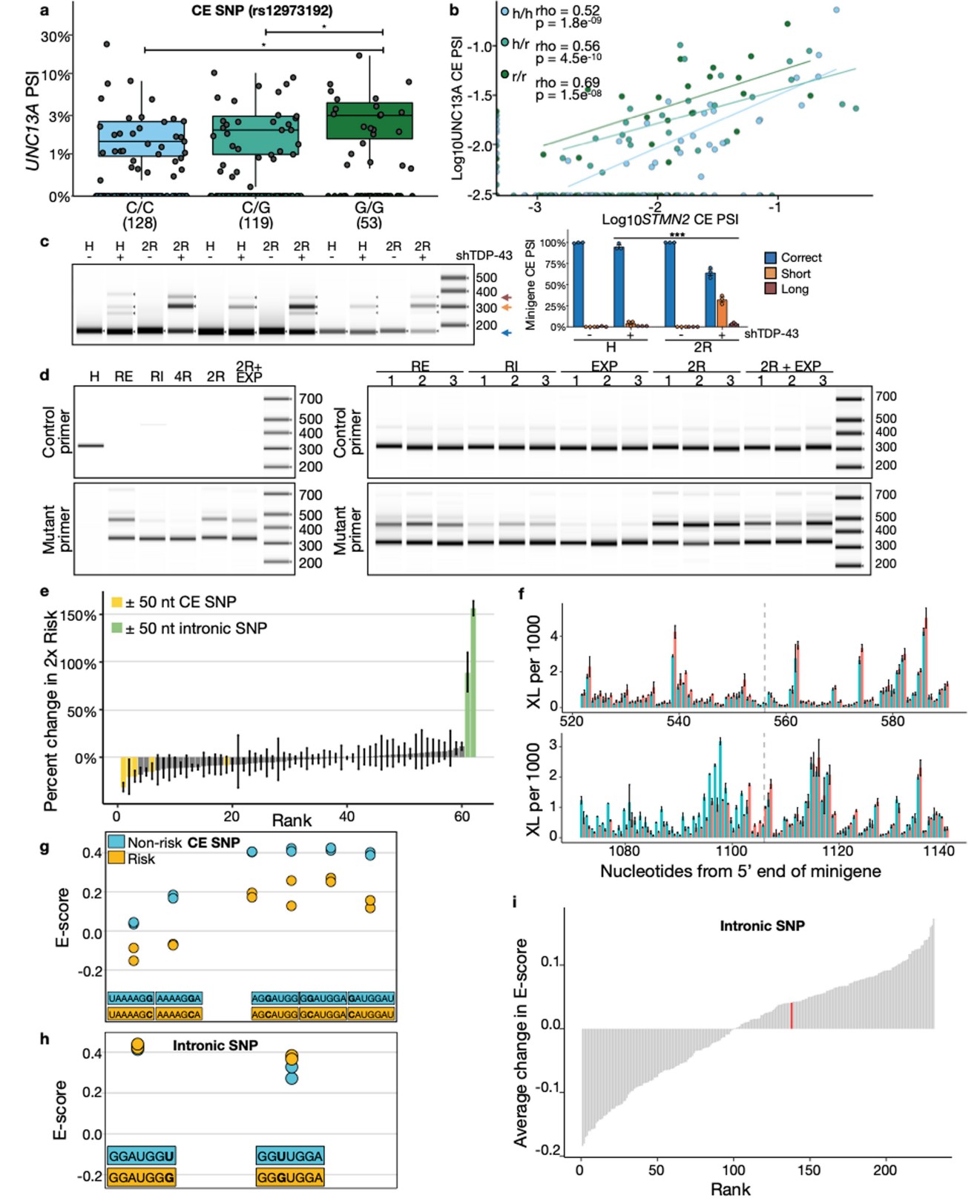

**Extended Data Fig. 8 |** See next page for caption.

**Extended Data Fig. 8 | *UNC13A* risk alleles increase *UNC13A* CE expression after TDP-43 depletion by altering TDP-43 binding affinity across the UNC13A CE-containing intron.** (**a**) *UNC13A* CE PSI by genotype (Wilcoxon test) Box plots: boundaries 25-75th percentiles; midline, median; whiskers, Tukey style. (**b**) Effect of CE or intronic SNP on the correlation between STMN2 and UNC13A CE PSI in ALS/FTD cortex in samples with at least 30 junction reads across the CE locus. Spearman's correlation. (**c**) Raw tapestation gel images of *UNC13A* CE products in 2H and 2R minigines and quantification of the PCR products. Graphs represent the means ± SEM (n = 3 biological replicates); Two-way ANOVA (**d**) Raw tapestation gel images corresponding to Fig. 4e. Two sets of primers were used to amplify either control (top row) or mutant minigene (bottom row). Left panel: single transfections were performed to ensure primer specificity. Right panel: three biological replicates of the double transfections. (**e**) Fractional changes at iCLIP peaks for 2R versus 2H minigene (mean and 75% confidence interval shown). Peaks that are within 50nt of each SNP are highlighted. (**f**) Mean crosslink density around the exonic (top) and intronic (bottom) SNPs in the 2H (red) and 2R (blue) minigenes, relative to the 5′ end of minigene (error bars = standard deviation; dashed lines show SNP positions). (**g, h**) Individual TDP-43 E-scores for the CE (**g**) and intronic (**h**) heptamers for which there was data[27] (**i**) Average change in E-value (measure of binding enrichment) across proteins for heptamers containing risk/healthy intronic SNP allele; TDP-43 is indicated in red. Significance levels reported as * (p < 0.05) ** (p < 0.01) *** (p < 0.001) **** (p < 0.0001).

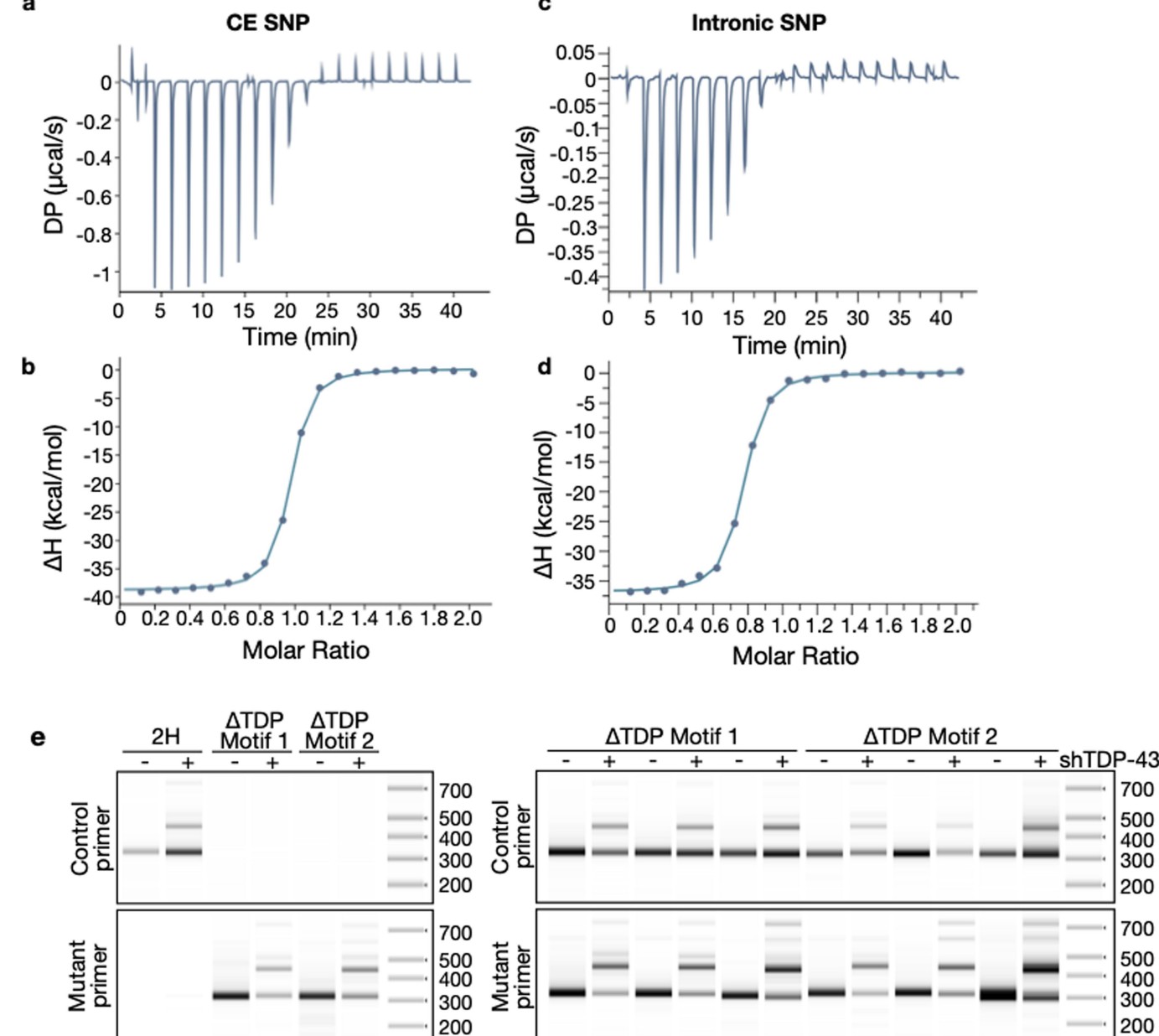

**Extended Data Fig. 9 | Binding of TDP-43 to SNP-containing intronic RNA.**
(**a**–**d**) ITC measurement of the interaction of TDP-43 with 14-nt RNA containing the CE SNP (**a, b**) and intronic SNP (**c, d**) healthy sequence. A representative data set is reported, with raw data (**a, c**) and integrated heat plot (**b, d**). Circles indicate the integrated heat; the curve represents the best fit. (**e**) Raw Tapestation gel images corresponding to Fig. 4j. For each experiment, two

RT-PCRs were performed with a different primer set which either amplified a control minigene (top row; minigene 2H) or a mutant minigene (bottom row). Left: single transfections to ensure specificity of primers for either the control or the mutant minigene. Right: Three replicates of double transfections with control minigene 2H and either mutant minigene.

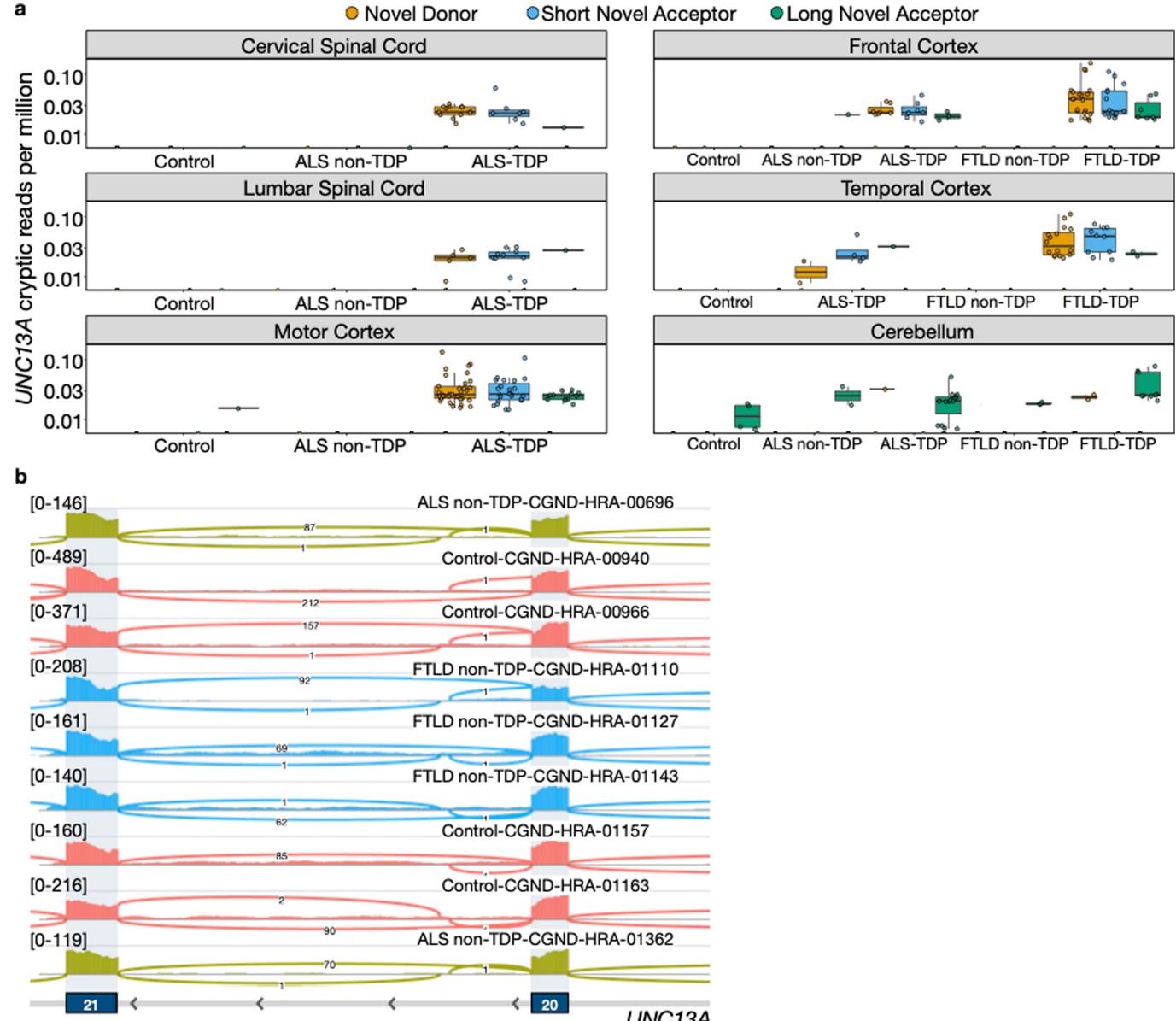

**Extended Data Fig. 10 | One of the splice junctions for UNC13A CE overlaps with an unannotated exon expressed in control cerebellum.** (**a**) Expression of splice junction reads supporting the *UNC13A* CE across tissues and disease subtypes. Junction counts are normalised by library size in millions (junctions per million). The long novel acceptor junction is expressed across all disease subtypes in the cerebellum. Box plots: boundaries 25–75th percentiles; midline, median; whiskers, Tukey style. (**b**) Example RNA-seq traces from IGV showing *UNC13A* cerebellar exon which shares the long novel acceptor junction as the *UNC13A* CE.

# Reporting Summary

Nature Research wishes to improve the reproducibility of the work that we publish. This form provides structure for consistency and transparency in reporting. For further information on Nature Research policies, see our Editorial Policies and the Editorial Policy Checklist.

## Statistics

For all statistical analyses, confirm that the following items are present in the figure legend, table legend, main text, or Methods section.

| n/a | Confirmed | |
|---|---|---|
| ☐ | ☒ | The exact sample size ($n$) for each experimental group/condition, given as a discrete number and unit of measurement |
| ☐ | ☒ | A statement on whether measurements were taken from distinct samples or whether the same sample was measured repeatedly |
| ☐ | ☒ | The statistical test(s) used AND whether they are one- or two-sided *Only common tests should be described solely by name; describe more complex techniques in the Methods section.* |
| ☒ | ☐ | A description of all covariates tested |
| ☒ | ☐ | A description of any assumptions or corrections, such as tests of normality and adjustment for multiple comparisons |
| ☐ | ☒ | A full description of the statistical parameters including central tendency (e.g. means) or other basic estimates (e.g. regression coefficient) AND variation (e.g. standard deviation) or associated estimates of uncertainty (e.g. confidence intervals) |
| ☐ | ☒ | For null hypothesis testing, the test statistic (e.g. $F$, $t$, $r$) with confidence intervals, effect sizes, degrees of freedom and $P$ value noted *Give P values as exact values whenever suitable.* |
| ☒ | ☐ | For Bayesian analysis, information on the choice of priors and Markov chain Monte Carlo settings |
| ☒ | ☐ | For hierarchical and complex designs, identification of the appropriate level for tests and full reporting of outcomes |
| ☒ | ☐ | Estimates of effect sizes (e.g. Cohen's $d$, Pearson's $r$), indicating how they were calculated |

*Our web collection on statistics for biologists contains articles on many of the points above.*

## Software and code

Policy information about availability of computer code

| Data collection | Xcalibur v4.3 |
|---|---|
| Data analysis | https://github.com/ulelab/ultraplex<br>https://github.com/frattalab/unc13a_cryptic_splicing<br>https://github.com/frattalab/rna_seq_snakemake<br>https://github.com/frattalab/splicing<br>https://github.com/frattalab/bedops_parse_star_junctions<br>https://github.com/frattalab/pipeline_iclip<br>MicroCal PEAQ-ITC<br>BWA-MEM v0.7.15<br>Genome Analysis Toolkit v3.5<br>LeafCutter - GitHub branch - 'psi_2019'<br>MAJIQ v2.1<br>Integrative Genomics Viewer v 2.9.1<br>Salmon v1.5.1<br>iCount v2.0.1.dev<br>STAR v2.7.0f & 2.7.2a<br>fastp v0.19.11<br>DESeq2 v1.30.1<br>Snakemake v5.5.4<br>IRFinder v1.3.0<br>bedtools v2.29.2<br>R v4.0.3 |

Proteome Discoverer (v2.4)
featureCounts v1.6.4
ultraplex v1.1.2
Bowtie2 v2.4.2
UMI-tools v1.0.1
LocusZoom
regtools v0.5.1
samtools v1.9
Picard Tools v2.4.1
ImageJ v1.52p

For manuscripts utilizing custom algorithms or software that are central to the research but not yet described in published literature, software must be made available to editors and reviewers. We strongly encourage code deposition in a community repository (e.g. GitHub). See the Nature Research guidelines for submitting code & software for further information.

## Data

Policy information about availability of data

All manuscripts must include a data availability statement. This statement should provide the following information, where applicable:
- Accession codes, unique identifiers, or web links for publicly available datasets
- A list of figures that have associated raw data
- A description of any restrictions on data availability

Minimum data to reproduce results freely available: https://github.com/frattalab/unc13a_cryptic_splicing/tree/main/data
RNA-Seq Data for i3Neurons, SH-SY5Y and SK-N-DZa are available through the European Nucleotide Archive (ENA) under accession PRJEB42763(Fig 1,2, fig S1).
Public data was obtained from Gene Expression Omnibus (GEO): iPSC MNs (Klim et al., 2019)-GSE121569, SK-N-DZb-GSE97262, and FACS-sorted frontal cortex neuronal nuclei-GSE126543(Fig 1,2,3, fig S1).
Riboseq: E-MTAB-10235(Fig 2, fig S2).
Targeted RNA seq: E-MTAB-10237(Fig 4)
Minigene iCLIP: E-MTAB-10297(Fig 4)
NYGC ALS Consortium RNA-seq: RNA-Seq data generated through the NYGC ALS Consortium in this study can be accessed via the NCBI's GEO database (GEO GSE137810, GSE124439, GSE116622, and GSE153960). All RNA-Seq data generated by the NYGC ALS Consortium are made immediately available to all members of the Consortium and with other consortia with whom we have a reciprocal sharing arrangement. To request immediate access to new and ongoing data generated by the NYGC ALS Consortium and for samples provided through the Target ALS Postmortem Core, complete a genetic data request form at ALSData@nygenome.org.(Fig 3,4, fig S5,6,S8-11,S13)
NYGC ALS Consortium Whole Genome Seq: to be released later with companion manuscript.
NYGC ALS Consortium genotypes on common SNPs in this study rs129731921 and rs12608932 are present at https://github.com/frattalab/unc13a_cryptic_splicing/blob/main/data/nygc_junction_information.csv

# Field-specific reporting

Please select the one below that is the best fit for your research. If you are not sure, read the appropriate sections before making your selection.

☒ Life sciences          ☐ Behavioural & social sciences          ☐ Ecological, evolutionary & environmental sciences

For a reference copy of the document with all sections, see nature.com/documents/nr-reporting-summary-flat.pdf

# Life sciences study design

All studies must disclose on these points even when the disclosure is negative.

| | |
|---|---|
| Sample size | Sample size in NYGC ALS Consortium was not determined in advance, the consortium data collection is ongoing, and sample size was determined by the number of available RNA-seq and genomically matched samples available at start time of analysis (2020-07). Sample size for RNA-seq on cell lines new to this study was determined by prior literature using similar experimental approaches rather than by power analysis. Examples of similar RNA-seq studies using similar (n of at least 3 in each condition) samples size include "Major hnRNP proteins act as general TDP-43 functional modifiers both in Drosophila and human neuronal cells" and "Premature polyadenylation-mediated loss of stathmin-2 is a hallmark of TDP-43-dependent neurodegeneration" |
| Data exclusions | Tissue samples with discordant genotype on rs12608932 and rs12973192 (2) were excluded from the analysis on genotype effect on the ratio of UNC13A/STMN2 cryptic PSI. |
| Replication | Expression of the UNC13A CE was replicated in multiple independent cell lines, WTC11 iPS and NCRM5 iPS cell lines. Minigene experiments were replicated in 3 biological replicates. SH-SY5Y doxycycline knock-downs were replicated in 3 biological replicates. |
| Randomization | Not relevant to this study as no randomization is required due to the homogeneous nature of the cell lines. Furthermore, our RNA-seq discovery assays are high-throughput and were initially done in a hypotheses-free analysis. Observations in the RNA-seq were validated and confirmed by biochemical assays to bolster initial observations from the high-throughput assays. |
| Blinding | BaseScope cortical sections were imaged and analysed blinded to disease status. |

# Reporting for specific materials, systems and methods

We require information from authors about some types of materials, experimental systems and methods used in many studies. Here, indicate whether each material, system or method listed is relevant to your study. If you are not sure if a list item applies to your research, read the appropriate section before selecting a response.

## Materials & experimental systems

| n/a | Involved in the study |
|---|---|
| ☐ | ☒ Antibodies |
| ☐ | ☒ Eukaryotic cell lines |
| ☒ | ☐ Palaeontology and archaeology |
| ☒ | ☐ Animals and other organisms |
| ☐ | ☒ Human research participants |
| ☒ | ☐ Clinical data |
| ☒ | ☐ Dual use research of concern |

## Methods

| n/a | Involved in the study |
|---|---|
| ☒ | ☐ ChIP-seq |
| ☒ | ☐ Flow cytometry |
| ☒ | ☐ MRI-based neuroimaging |

## Antibodies

**Antibodies used**

Rabbit anti-UNC13A (Synaptic Systems 126 103) dilution 1:2,000
Rabbit anti-UNC13B (abcam ab97924) dilution 1:1,000
Rat anti-Tubulin (abcam ab6161 clone YOL1/34) dilution 1:5,000
Mouse anti-TDP-43 (abcam ab104223 clone 3H8) dilution 1:5,000
Rabbit anti-TDP-43 10782-2-AP (Proteintech) - lot 00065465
Goat anti-Rabbit HRP (Bio-Rad 1706515) 1:10,000
Goat anti-Mouse HRP (Bio-Rad 1706516) 1:10,000
Rabbit anti-Rat HRP (Dako P0450) 1:10,000
Donkey anti-Rabbit 488 (Jackon Immuno 711-545-152)
Donkey anti-Mouse 647 (Jackson Immuno 715-605-151)
Rabbit anti-TDP-43 (Proteintech 12892-1-AP)
Mouse anti-TUBB3 (Biolegend 801201)

**Validation**

Mouse anti-TDP43 abcam ab104223 was validated via shRNA experiments against TDP-43

Rb anti-UNC13A (synaptic systems 126 103) has been previously verified in KO mouse [Lai Y, Choi UB, Leitz J, Rhee HJ, Lee C, Altas B, Zhao M, Pfuetzner RA, Wang AL, Brose N, Rhee J, et al. Molecular Mechanisms of Synaptic Vesicle Priming by Munc13 and Munc18. Neuron (2017) 953: 591-607.e10.]. The antibody was additionally validated by western blot analysis of human brain lysate and U2OS cell lysate which do not express UNC13A. As well as overexpression experiments with GFP tagged UNC13A.

Rb anti-UNC13B (abcam ab97924) was validated with overexpression experiments with GFP tagged UNC13B.

Rat anti-Tubulin (abcam ab6161) has been cited in > 126 publications.
Rabbit anti-TDP-43 10782-2-AP (Proteintech) has been cited in >1200 publications. Rabbit anti-TDP-43 10782-2-AP (Proteintech) has postive WB detection in SH-SY5Y cells, HeLa cells, K-562 cells, C2C12 cells, Neuro-2a cells, Positive IHC detected in mouse brain tissue, human brain tissue, human brain (FTLD-U) tissue, human gliomas tissue, human pancreas tissue, rat brain tissue. Positive IF detected in HeLa cells, SH-SY5Y cells,Positive FC detected in HeLa cells.

Goat anti-Rabbit HRP (Bio-Rad 1706515) according to manufacturer's website has been double-affinity purified with human IgG adsorbed.
Goat anti-Mouse HRP (Bio-Rad 1706516) has been used in >1000 citations.
Rabbit anti-Rat HRP (Dako P0450) has been used in 120 citations.
Donkey anti-Rabbit 488 (Jackon Immuno 711-545-152) has been used in 705 citations - https://www.jacksonimmuno.com/catalog/products/711-545-152
Donkey anti-Mouse 647 (Jackson Immuno 715-605-151) has been used in 121 citations - https://www.jacksonimmuno.com/catalog/products/715-605-151
Rabbit anti-TDP-43 (Proteintech 12892-1-AP) has been used in 321 citations https://www.ptglab.com/Products/TARDBP-Antibody-12892-1-AP.htm
Mouse anti-TUBB3 (Biolegend 801201)has been used in 597 citations https://www.biolegend.com/en-us/search-results/purified-anti-tubulin-beta-3-tubb3-antibody-11580

## Eukaryotic cell lines

Policy information about cell lines

**Cell line source(s)**

iPS-derived cortical neurons are from the WTC11 line, which was derived from a healthy human male participant and NCRM5, which was derived from a healthy human male's cord blood and obtained from the Coriell cell repository. All policies of the NIH Intramural Research Program for the registration and use of this iPS cell line were followed. HEK293T and

SH-SY5Y cells were obtained from ATCC. SK-N-DZ cells were obtained from the International Centre for Genetic Engineering and Biotechnology in Trieste, Italy.

| Authentication | WTC11 iPS cell line was validated to have a normal male karyotype<br>NCRM5 iPS cell line was validated to have a normal male karyotype |
| --- | --- |
| Mycoplasma contamination | WTC11 iPS cell line was confirmed to be mycoplasma free based on the Lonza MycoAlert mycoplasma testing kit.<br>NCRM5 iPS cell line was confirmed to be mycoplasma free based on the Lonza MycoAlert mycoplasma testing kit. |
| Commonly misidentified lines<br>(See ICLAC register) | No commonly misidentified cell lines were used in this study. |

# Human research participants

Policy information about studies involving human research participants

| Population characteristics | 1349 tissue samples from 377 unique participants (164 female). 77 controls - Median age 67, 239 ALS - Median age 66, 61 FTLD - Median age 67 |
| --- | --- |
| Recruitment | In NYGC ALS Consortium recruitment and contribution postmortem samples and clinical information was performed by members using their recruitment criteria and strategy. |
| Ethics oversight | The NYGC ALS Consortium samples presented in this work were acquired through various IRB protocols from member sites and the Target ALS postmortem tissue core and transferred to the NYGC in accordance with all applicable foreign, domestic, federal, state, and local laws and regulations for processing, sequencing, and analyses. The Biomedical Research<br>3<br>Alliance of New York (BRANY) IRB serves as the central ethics oversight body for NYGC ALS Consortium. Ethical approval was given and is effective through 08/22/2022. |

Note that full information on the approval of the study protocol must also be provided in the manuscript.

