## [Peer Review File · Nature]

Manuscript Title: TDP-43 loss and ALS-risk SNPs drive mis-splicing and depletion of UNC13A

Redactions – unpublished data

Reviewer Comments & Author Rebuttals

Reviewer Reports on the Initial Version:

Referee #1

Common ALS/FTD risk variants in UNC13A exacerbate its cryptic splicing and loss upon TDP-43 mislocalization (Brown et al) is a well written and interesting manuscript, which potentially provides a mechanistic understanding of one of the established risk loci for FTD/ALS and directly links it to the function and deposition of TDP43 within the brain tissue of affected individuals. There are relatively few examples where this form of molecular understanding has been achieved for complex diseases, especially to this level of detail, and therefore this paper will generate considerable interest. However, it will be of greatest interest to those focused on TDP43-related diseases and those interested in the development of novel therapies in the field. It is not clear to me the extent to which these findings or the approach taken are generalisable beyond this specific disease and within that the subtype characterised by TDP43 deposits. The authors do raise the point that TDP43 is not unique to ALS/FTD but this is a complex field and I think it will be hard to easily extrapolate.

A range of methods and approaches have been used to connect TDP43 function to the UNC13A locus, and I think in places additional detail is required to fully assess the validity of the data and the conclusions reached.

In terms of further experimental work which I think would improve this study, I think one area is the splicing defect in UNC13A. This is complex making it hard to really know what abnormal transcripts are being produced, their true usage and consequently the exact mechanism by which they cause disease. I think this is most concerning around the relationship between CE inclusion, the overall levels of UNC13A expression and the protein expression. Now that it is possible to run targeted long read RNAseq which could be performed across samples and which could allow more accurate delineation of the proposed abnormal UNC13A transcript/s containing the CE and the relative usage of transcripts. I think the inclusion of this form of data would significantly add to the work.

I also think it would be valuable to go back to iPSC/mouse models of pathogenic Mendelian forms of FTD/ALS due to various TDP43 mutations. It strikes me that variability in the severity of the disease could be explained amongst these patients by variability at the UNC13A locus and this is very testable.

In summary, I think this is a very valuable and interesting piece of work and the efforts of the authors to really try and nail the molecular processes is admirable.

In terms of specific comments:

To discover novel CEs induced by TDP-43 depletion, we performed RNA-seq on human induced pluripotent stem cell (iPSC)-derived cortical-like i3Neurons in which we reduced TDP-43 expression through CRISPR inhibition (CRISPRi)^{10–13}. We identified 179 CEs, including several previously reported, such as AGRN, PFKP and STMN26–9 (Fig. 1A; data S1) (Fig. 1B; data S2).

- I may have missed it in the methods, but it wasn't clear to me how many independent iPSC lines this was actually done with. Related to this point have the iPSC-lines used been assessed for ALS/FTD relevant mutations?

Inspection of the UNC13A gene revealed a previously unreported CE after TDP-43 knockdown (KD), with both a shorter and longer form, between exons 20 and 21 (Fig. 1C), and increased IR between exons 31 and 32 (fig. S1B).

- The complexity of the events detected makes it hard to know from short read data the actual transcripts being produced and I think this is important in terms of understanding mechanism and is soluble with targeted long read sequencing. Therefore I would suggest these experiments are performed.

TDP-43 KD also decreased expression of UNC13A and UNC13B at the protein level, as assessed by quantitative

proteomics with liquid chromatography tandem-mass spectrometry and western blot (Fig. 2D,E). These data suggest that the mis-splicing in UNC13A and UNC13B after TDP-43 KD reduces their transcript and protein abundance in neurons.

- Given that genuinely quantitative methods have been used to assess the UNC13A-CE inclusion transcripts, the UNC13A canonical transcript levels and protein levels, I think it would be important to really assess how well they correlate and formally model whether the protein level changes can be fully explained by the CE levels. Looking at the plots it's not clear to me that this is the case and perhaps this is unsurprising given the complex relationship between RNA and protein. However, this does raise the possibility of additional mechanisms and at the very least this should be explored and recognised.

We compared levels of UNC13A CE to levels of a CE in STMN2 known to be regulated by TDP-43. Both STMN2 and UNC13A CEs were exclusive to TDP-43-depleted nuclei, and, strikingly, in some cases the UNC13A CE percent spliced in (PSI) reached 100% (Fig. 3A).

- I think the validity of studying splicing using nuclear RNA extractions needs to be addressed. After all the implication is that the authors are studying pre-mRNA/ partially processed RNA species and so there is a possibility that this isn't mis-splicing but a change in splicing dynamics.

Next, we quantified UNC13A CE inclusion in bulk RNA-seq from the NYGC ALS Consortium, a dataset containing 1,349 brain and spinal cord tissues from a total of 377 ALS, FTLN, and control individuals. The UNC13A CE was detected exclusively in FTLN-TDP and ALS-TDP cases (89% and 38% respectively), with no detection in ALS-non-TDP (SOD1 and FUS mutations), FTLN-non-TDP (FTLN-TAU and FTLN-FUS), or control cases.

- I think what is meant by detection of the UNC13A CE should be clarified with some account taken of the likely change in cellular composition of the tissues between cases and controls which could be a confounding factor as well as read depth, RIN differences etc. From the methods, it is not clear to me that this has occurred.

The lower detection rate in ALS versus FTLN is likely due to the lower expression of UNC13A in the spinal cord (fig. S3A). Thus, pathological UNC13A CEs occur in vivo and are specific to neurodegenerative disease subtypes in which mislocalization and nuclear depletion of TDP-43 occurs.

- I am not sure that this is a valid conclusion since it doesn't consider the role of NMD in removing UNC13A- CE containing transcripts and NMD also has regional differences in expression.

- Also when something is not detected much more controls on what that means

To test whether the ALS/FTLN UNC13A risk SNPs promote cryptic splicing, which could explain their link to disease, we assessed UNC13A CE levels across different genotypes, and found significantly increased levels in cases homozygous for CE rs12973192(G) and intronic rs12608932(C) SNPs (fig. S4A-B). To ensure that this was not simply due to more severe TDP-43 pathology in these samples, we normalised by the level of STMN2 cryptic splicing, and again found a significantly increased level of the UNC13A CE in cases with homozygous risk variants (Wilcoxon test, $p < 0.001$) (Fig. 4A; fig. S4C,D).

- I am not sure about the validity of using STMN2 cryptic splicing to normalise. If it is a proxy for the levels of TDP43 pathology why not just quantify that directly and include it within a single linear module. In addition, again it isn't clear whether the authors are taking into account common known covariates (ethnicity, sex, age etc) assessed for unknown covariates and tissue composition. This is in effect eQTL analysis and this is all very standard for the field. With this in mind, I'm not sure why a Wilcoxon test has been used and the p-value doesn't seem particularly impressive in that context.

Next, we performed targeted RNA-seq on UNC13A CE from temporal cortices of ten heterozygous risk allele cases and four controls. We detected significant biases towards reads containing the risk allele ($p < 0.05$, single-tailed binomial test) in six samples, with a seventh sample approaching significance (Fig. 4B), suggesting that the two ALS/FTLN-linked variants promote cryptic splicing in vivo.

- This is an ASE analysis and so of course it's not just the p-val that matters but the direction of effect and it is highly prone to mapping biases or biases in primer design when a capture is done so more information is needed to ensure this result is valid. In addition the exact p-value is not provided and there is some controversy over the use of a binomial test here.

To specifically examine whether the CE or the intronic SNP of UNC13A promote CE splicing, we generated four variants of minigenes containing UNC13A exon 20, intron 20, and exon 21, featuring both risk alleles (2R), both

non-risk alleles (2H), the risk allele within the CE (rs12973192) (RE), or the risk allele in the intron (rs12608932) (RI) (Fig. 4C). We then expressed these minigenes in SH-SY5Y cells with doxycycline-inducible TDP-43 knockdown. We found that both the CE SNP and, to a lesser extent, intronic SNP independently promoted CE inclusion, with the greatest overall levels detected for the 2R minigene (Fig. 4D,E).

- I may have missed it, but have the authors checked to assess other SNPs/variants in LD in the region and that could include repeats.

Figure 1.

Panels A) and B): The impact on splicing looks much more significant than the impact on expression and it's not at all clear to me that UNC13A would have been flagged through expression. In fact there appear to be better candidates, what are they and have they been investigated or assessed in any formal way eg. just intersecting with genes linked to ALS/FTD through GWAS or Mendelian disease.

(H) Correlation between relative TARDBP RNA and UNC13A CE PSI across five TDP-43 knockdown datasets : I find this plot unconvincing. If the authors accounted for differences in cell type, I don't think there would be any correlation and for the iPSC derived motor neurons it would be flat.

Fig. 3. UNC13A CE is highly expressed in ALS/FTLD patient tissue and correlates with known markers of TDP-43 loss of function. (E) Correlation in ALS/FTLD-TDP cortex between UNC13A and STMN2 CE PSI in patients with at least 30 spliced reads across the CE locus.

- Couldn't this form of analysis be broadened in order to look for correlations between STMN2 CE PSI and other apparently abnormal splicing events? Has this been done? It could throw up other interesting candidates and would also enable the correlation for UNC13A to be assessed more robustly.

iPSC-derived i3Neuron differentiation and culture

i3Neurons were then fed three times a week by half media changes. i3Neuron were then harvested on day 17 post addition of doxycycline or 14 days after re-plating.

- Is 17 days sufficient for neuronal maturation? It seems short to me, but I am not familiar with iPSC protocols.

RNA-sequencing, differential gene expression and splicing analysis

For RNA-seq experiments of i3Neurons, the i3Neurons were grown on 96-well dishes. To harvest on day 17, media was completely removed, and wells were treated with tri-reagent (100 μ L per well) (Zymo research corporation, Cat. No. R2050-1-200). Then 5 wells were pooled together for each biological replicate: control (n=3); TDP-43 knockdown (n=4). To isolate RNA, we used a Direct-zol RNA miniprep kit (Zymo Research Corporation, Cat. No. R2052), following manufacturer's instructions including the optional DNase step. Note: one control replicate did not pass RNA quality controls and so was not submitted for sequencing. Total RNA was then enriched for polyA and sequenced 2x75 bp on a HiSeq 2500 machine.

- The read lengths used are rather short for a splicing assessment and I see no information on the actual library construction method used. Also I note data pooling was performed across biological replicates, why?

DESeq2's median of ratios, which controls for both sequencing depth and RNA composition, was used to normalize gene counts. Differential expression was defined at a Benjamini-Hochberg false discovery rate < 0.1 .

- This seems a very generous definition of differential expression. If a more convention FDR of at least < 0.05 was used would UNC13A be differentially expressed at all?

Differential splicing was performed using MAJIQ (v2.1)³⁷ using the GRCh38 reference genome. A threshold of 0.1 Δ PSI was used for calling the probability of significant change between groups.

- MAJIQ is used for this analysis but then a later analysis uses LeafCutter, why when both these methods address the same question?

Analysis of published iCLIP data

Cross-linked read files from TDP-43 iCLIP experiments in SH-SY5Y and human neuronal stem cells²² were processed using iCount v2.0.1.dev implemented in Snakemake version 5.5.4, available at

https://github.com/frattalab/pipeline_iclip . Sites of cross-linked reads from all replicates were merged into a single file using iCount group command. Significant positions of cross-link read density with respect to the same gene (GENCODE v34 annotations) were then identified using the iCount peaks command with default parameters.

- The GENCODE build is different here to that used in the RNAseq analyses. I think it would be important to maintain consistency.

Nonsense-mediated decay (NMD) inhibition

Ten days post induction of shRNA against TDP-43 with 1 µg/ml doxycycline hyclate (Sigma D9891-1G), SH-SY5Y cells were treated either with 100 µM cycloheximide (CHX) or DMSO for 6 hours⁴¹ before harvesting the RNA through RNeasy Minikit (Qiagen).

- Cycloheximide will not just affect NMD and therefore I think an alternative approach would be beneficial.

Reverse transcription was performed using RevertAid cDNA synthesis kit (Thermo), and transcript levels were quantified by qPCR (QuantStudio 5 Real-Time PCR system, Applied Biosystems) using the $\Delta\Delta C_t$ method and GAPDH as reference⁴⁰. Since it proved to undergo NMD⁴², hNRNPL NMD transcript was used as a positive control.

- It is not clear to me that GAPDH would be an appropriate reference, has this been checked?

Ribosome footprints from 3x TDP-43 knockdown and 3x control samples were generated and purified as described, using a sucrose cushion (McGlincy and Ingolia, 2017) and a customised library preparation method based on revised iCLIP⁴³.

- Are the replicates biological or technical?

NYGC ALS Consortium RNA-seq cohort

Sample processing, library preparation, and RNA-seq quality control have been extensively described in previous papers^{10,49}. In brief, RNA was extracted from flash-frozen postmortem tissue using TRIzol (Thermo Fisher Scientific) chloroform, and RNA-Seq libraries were prepared from 500 ng total RNA using the KAPA Stranded RNA-Seq Kit with RiboErase (KAPA Biosystems) for rRNA depletion. Pooled libraries (average insert size: 375 bp) passing the quality criteria were sequenced either on an Illumina HiSeq 2500 (125 bp paired end) or an Illumina NovaSeq (100 bp paired end). The samples had a median sequencing depth of 42 million read pairs, with a range between 16 and 167 million read pairs.

- This data strikes me as relatively complex from the point of view of splicing. There is clearly batching with different read lengths and total RNA has been used with ribodepletion which means you expect more pre-mRNA reads which complicate analyses. It is not clear to me how these issues are accounted for in the analysis. Furthermore, the range in read depth is huge and for some samples I would question whether this is indicative of a library/technical failure – certainly I find it hard to see how reliable the data would be. These issues need to be formally addressed and this may require new analyses.

Samples were uniformly processed, including adapter trimming with Trimmomatic and alignment to the hg38 genome build using STAR (2.7.2a)³² with indexes from GENCODE v30.

- This is the third GENCODE version being used in the paper. Either the authors should standardise their use of GENCODE versions or demonstrate that this does not impact on their findings.

Uniquely mapped reads within the UNC13A locus were extracted from each sample using SAMtools. Any read marked as a PCR duplicate by Picard Tools was discarded. Splice junction reads were then extracted with RegTools⁵² using a minimum of 8 bp as an anchor on each side of the junction and a maximum intron size of 500 kb. Junctions from each sample were then clustered together using LeafCutter⁵³ with relaxed junction filtering (minimum total reads per junction = 30, minimum fraction of total cluster reads = 0.0001). This produced a matrix of junction counts across all samples.

- LeafCutter is the second splicing tool being used what is the justification? Furthermore, it isn't clear to me that the exon-exon junction read filtering for MAJIQ and LeafCutter are the same and this is also an issue.

Hybridised sections were graded, blinded to disease status, according to the relative frequency of red foci which should identify single transcripts with the UNC13A CE event. Grades were prescribed by relative comparison with the negative control slide. - = Less signal than negative control probe; + = similar signal strength to negative control; ++ = visibly greater signal than negative control, +++ = considerably greater signal than negative control. We identified a signal level above background (++ or +++) in 4 of 5 FTLD-TDP cases and a signal

considerably above background (+++) level in 2 cases. All FTLD-Tau and control cases were graded as exhibiting either reduced (-) or comparable (+) signal relative to background.

- This is a semi-quantitative analyses, but there are methods available to make this quantitative. Those methods should be used.

UNC13A genotypes in the NYGC ALS Consortium

Whole Genome Sequencing (WGS) was carried out for all donors, from DNA extracted from blood or brain tissue. Full details of sample preparation and quality control will be published in a future manuscript.

- I am not sure how this sits with the editorial policy on data release though I am sympathetic to the issues around data release timing from big consortium projects. Maybe genotyping has been performed and this is available in the public domain?

Targeted RNA-seq

RNA was isolated from temporal cortex tissue of 10 FTLD-TDP and four control brains (6M, 4F, average age at death $70.6 \pm 5.8y$, average disease duration $10.98 \pm 5.9y$).

- Given the small number of samples, I think precise data should be provided on all samples in a tabular form. Three rounds of nested PCR using Phusion HF 2x Master Mix (New England Biolabs) were used to obtain highly specific amplicons for the UNC13A cryptic, followed by gel extraction and a final round of PCR in which the full length P3/P5 Illumina sequences were added. Samples were sequenced with an Illumina HiSeq 4000 machine (SR100).

- Further information is required to convince me of the specificity of the amplicons produced and that this process was not directly biased by the SNPs of interest which would make an ASE analysis potentially invalid.

Raw reads were demultiplexed, adaptor/quality trimmed and UMIs were extracted with Ultraplex (<https://github.com/ulelab/ultraplex>), then aligned to the hg38 genome with STAR32; for the hexamer data, a subsample of reads was used to reduce the number of PCR duplicates during analysis.

- Have the authors considered the problem of mapping biases, and if so how have they accounted for that given that these biases have the potential to create false positive ASE results.

- I do not understand the rationale behind the use of a subsample of reads in this way.

Referee #2

Amyotrophic lateral sclerosis (ALS) and frontotemporal dementia (FTD) are two neurodegenerative disorders with a common etiology; the RNA binding protein TDP-43 undergoes a shift from a nuclear- to a cytosolic localization, thereby losing its ability to function as a splicing repressor. The downstream effect involves mis-splicing events, for example the unwanted inclusion of so-called cryptic exons (CEs) in mature RNA. These CEs would normally be spliced out, and their inclusion results in downstream errors in protein homeostasis.

Munc13-1 is a central presynaptic protein. It is involved in a preparation step synaptic vesicles must undergo to make them release competent. Munc13-1 is the central isoform in the majority of synapses at the central nervous system (CNS), as well as in the neuromuscular junction. Its knock-out in mice is a cause for one of the most severe phenotypes in synaptic biology; in excitatory hippocampal neurons, it results in a 90% block of neurotransmission, and the concomitant removal of Munc13-2 leads to a complete block of neurotransmitter release, in both excitatory and inhibitory CNS neurons (Augustin et al., 1999; Varoqueaux et al., 2002). Recently, Munc13-1 has been shown to be arranged in clusters that are thought to represent discrete SV release sites (Reddy-Alla et al., 2017; Sakamoto et al., 2018). The number of clusters is associated with the strength of the synapse; synapses with many clusters will release more glutamate, making them stronger, whereas synapses with less clusters will release less glutamate, making them weaker. Work in the *Drosophila* neuromuscular junction also showed that the amount of the Munc13-1 homolog UNC13A is scalable under plasticity (Bohme et al., 2019). Current interpretations of these data suggests that the levels of Munc13-1 in synapses may regulate the strength of neurotransmitter release. In addition to synaptic strength, Munc13-1 is a regulator of synaptic plasticity, translating elevations in Ca^{2+} and dynamic changes in membrane lipid concentrations into changes in the efficacy of neurotransmission during activity (Lipstein et al., 2013; Shin et al., 2010). Finally, two cases of human brain disease have been described, with the loss of Munc13-1 leading to a severe brain disease and premature death (Engel et al., 2016), and one variation described to result in a neurodevelopmental and a neuropsychiatric disorder (Lipstein et al., 2017).

A genetic association between non-coding SNPs in the UNC13A gene locus and ALS has emerged over a decade ago. Interestingly, the SNPs have been associated with increased disease risk and with increased disease progression, which makes the mechanism by which these contribute to disease significant and important to elucidate. To date, however, the molecular and cellular mechanisms linking these SNPs to the disease, and, more importantly, whether these indeed involve the Munc13 gene product, remained elusive.

In this exciting manuscript, Brown and colleagues establish these mechanisms and links. They demonstrate that the nuclear loss of TDP-43 leads to a CE inclusions in the UNC13A and UNC13B RNA, resulting in substantial reduction in Munc13 protein expression. The supporting data includes:

- A screen in cortical iPSC under reduced TDP-43 expression that identifies splicing abnormalities in UNC13A and UNC13B, as well as in additional 177 transcripts. The previously identified disease-related SNPs lie in, or close to one of the identified CEs (Figure 1). TDP-43 binding sites are found in, or in close proximity to the SNPs and identified CEs.
- Demonstration that the splicing defect results in reduced RNA levels, and in reduced Munc13-1 and Munc13-2 protein levels. UNC13A RNA levels are decreased due to nonsense-mediated decay (Figure 2).
- Evidence that mis-spliced RNA of UNC13A is found in extremely high levels in patient-derived material, exclusively in patients with TDP-43-dependent ALS/FTLD (rather than in ALS/FTLD caused by other genetic reasons), and in the disease-relevant tissue only (Figure 3).
- A semi-quantitative, indirect analysis indicating relationship between TDP-43 pathology and UNC13A mis-splicing levels (Figure 4).
- A minigene splicing analysis showing that the risk SNPs exerts an additive, deleterious effect on splicing (Figure 4 and 4S). This may be due to the SNPs inhibiting binding of TDP-43 to the RNA.

The paper is clear and well-written. The experimental array used and the conclusions drawn from the data appear justified and robust to me, although I am not an expert in the full spectrum of methodologies used here, and others may be better suited to determine whether all controls are included. The presentation of the data is mostly clear (see comments below regarding Figure 4).

The major novelty is the univocal demonstration of a functional relationship between TDP-43 and Munc13 proteins. This exciting finding is highly significant, as it is associated with SNPs that exert a substantial effect on disease course. Because Munc13s are primarily synaptic proteins, the data presented here draws a strong connection between presynaptic dysfunction and the neurodegenerative disorders ALS and FTLD, expanding the already identified link between presynaptic function and neurodegeneration in tauopathies and alpha-synucleinopathies. Notably, the clear demonstration that the RNA irregularities in UNC13 transcripts can be found in patient-induced material is extremely convincing of the relevance of the finding. This paper opens a new avenue for mechanistic studies of ALS/FTLD in synaptic biology and its findings are likely to be followed up by many.

A major element that the current paper does not address is whether and how reducing the expression levels of the Munc13 proteins may lead to cellular pathology and/or to disease course modulation. It is important to note, that while a relationship between the reduction of Munc13-1/2 expression has been functionally established in hippocampal and striatal neurons (no Munc13s = no synaptic transmission (Varoqueaux et al., 2002)), and in the neuromuscular junction (no Munc13s = changes in synaptic transmission (Varoqueaux et al., 2005)) no such relationship has been established with regards to cell survival. In fact, while mice lacking the two isoforms die at birth due to a block of synaptic transmission, their CNS appears fully intact (Varoqueaux et al., 2002). In CNS-derived cultures or in organotypic slices in vitro, no degenerative effects have been observed (Broeke et al., 2010; Imig et al., 2014). In a single human patient with a Munc13-1 protein loss, neuromuscular architecture was fully preserved (Engel et al., 2016). Moreover, in mice, heterozygosity results in 50% reduction in protein abundance, but not in observable changes in synaptic function or in animal behavior, although this has not been studied in great detail (Augustin et al., 1999; Varoqueaux et al., 2005). Because conditional mouse models to reduce the expression of Munc13 isoforms in the adult nervous system (to lower levels than 50%) have not been published yet, the effects of such reduction at the adult nervous system have not been studied yet and are difficult to address. As such, this goes beyond the scope of the current manuscript. I would therefore recommend the paper for publication, provided

that the following major points will be addressed:

1. Figure 4: in what samples was the analysis in Figure 4A, 4SA-D made?
2. Figure S4 A-D: There seems to be some issue with the figures, or else I do not follow. Panels A and B have different titles, but refer to the same SNPs (title of A: 'CE SNP (rs12973192)', title of B: 'Intronic SNP (rs12973192)'). They are supposed to refer to different SNPs, which is also indicated in the legends. But these graphs appear to contain, essentially, the same data, only that the data points are scattered differently on the x-axis, making them look somewhat different. The P values are also different. Graphs C and D appear identical. Nonetheless, the P-values are not the same. This should be carefully checked.
3. While NMD is proposed as the mechanism to explain reduced RNA levels of UNC13A, no such analysis exists for UNC13B. UNC13B analysis in patient-derived material is missing. In fact, after the first section, UNC13B is ignored. For the completeness of the presentation and for a better mechanistic understanding, I find this is necessary.
4. Can the authors provide some evidence for reduced protein expression in patient-derived material?
5. For clarity of the proposed model: the risk alleles are also present in healthy people, with no TDP-43 pathology. TDP-43 reduction in binding affinity to UNC13A RNA is therefore expected to occur in them as well. However, the authors did not detect CE inclusion in healthy individuals. Why in healthy people, who carry the risk alleles leading to impaired TDP-43 binding, no effects are seen? The authors propose that the CE inclusion occurs below a certain TDP-43 threshold. Can this be substantiated at the RNA or protein level? For example, by providing data for the time course of CE inclusion in patients? This will aid in supporting the statement that 'severe nuclear depletion of TDP-43 in the end stage disease induces CE inclusion...' (discussion). It would be fascinating to align such data to the time course of cell degeneration.
6. The authors conclude by stating 'Excitingly, UNC13A provides a generalizable therapeutic target for 97% of ALS and approximately half of FTD cases.' Considering the devastating diseases involved, and the impact such a sentence may have, such a statement requires more support – what are these numbers based on? Munc13s are terrible targets for pharmacology - manipulating the function of the Munc13-1 protein is likely to affect the vast majority of synapses in the brain and out of the brain, potentially leading to a massive imbalance in neurotransmission. Genetic strategies to correct for the deleterious SNPs is not likely to prevent disease, but rather to delay disease progression. Are the authors proposing something in this direction? Please clarify.

Referee #3

In this study, Brown, Wilkins, Keuss, Hill, and colleagues report the inclusion of a novel cryptic exon (CE) in the gene UNC13A, an RNA target of TDP-43, in a subset of FTD and ALS postmortem patient samples, which results in a dramatic decrease of UNC13A mRNA and protein levels. Using a bioinformatic approach and experimental validation in cells, they show that CE inclusion is dependent on TDP-43 loss of function and is increased by two UNC13A small nucleotide polymorphisms (SNPs), previously shown to be associated with increased ALS and ALS-FTD risk. They also provide evidence that the CE-containing UNC13A transcript is subject to nonsense-mediated decay and that the ALS-associated SNPs directly distort TDP-43 binding on the region. Collectively, the work shows that the UNC13A CE inclusion is a marker of TDP-43 dysfunction in TDP-43 proteinopathies and indicates a direct link between TDP-43 loss of function and disease.

Following the recent discovery of the abnormal inclusion of STMN2 CE upon TDP-43 loss of function and the resulting loss of STMN2 mRNA and protein with detrimental effects for neurons (Melamed et al, 2019; Klim et al, 2019), the current work supports a mechanistic link between TDP-43 pathology and defects in key neuronal proteins. The findings are novel and exciting as they set the stage for several applications, including potentially UNC13A-based diagnostic or patient stratification tests, as well as potential therapeutic approaches for these diseases.

The manuscript is clearly written and the experiments are carefully designed and well-controlled. Due to the strong relevance of these findings for human disease and the high quality of the study, I enthusiastically recommend this work for publication in Nature, after revision to address the following concerns:

1. My main concern is the lack of functional validation of the effect of UNC13A decrease in adult human neurons. While I appreciate that this is technically challenging, I think it is important to show that this direct effect of TDP-43

loss of function on UNC13A levels has significant consequences for neuronal maintenance and function. In my view, this is necessary to support the claimed therapeutic potential of the work. Is the loss of UNC13A the culprit of neuronal death? Since STMN2 CE inclusion levels are always at least as high as those for CE inclusion in UNC13A and loss of STMN2 has been proven to be deleterious (Melamed et al., 2019), how could the relevance of UNC13A loss for cell survival be differentiated from STMN2 loss? Can reintroduction of UNC13A levels restore neuronal survival? Without answering some of these key questions, I think UNC13A levels may be a potential diagnostic tool, but not (yet) a therapeutic target.

2. Related to the above point, I find the statement: "That genetic variation influencing the UNC13A CE inclusion can lead to changes in ALS/FTD susceptibility and progression strongly supports UNC13A downregulation to be one of the critical consequences of TDP-43 loss of function" somewhat premature. Based on the current data, UNC13A CE inclusion could be a modifier of disease risk without necessarily be a critical event for neuronal survival, like STMN2 was shown to be.

3. The authors explored the effect of the ALS-associated SNPs on TDP-43 binding on this region, which I think is very important. In my view, however, these data need to be further explained and expanded to be conclusive. In particular:

i. The plot in Figure 4F indicates that TDP-43 binding in the UNC13A transcript is decreased in the presence of ALS-associated SNPs. However, it is unclear to me what kind of data and what kind of analysis was used to create this graph. The authors should clearly explain this in the results and methods parts of their manuscript.

ii. In the same graph (Figure 4F), what are the other few RBPs that show significantly reduced binding (even more than TDP-43) in the same region? Are they potentially linked to the mechanism of CE inclusion in the presence of the SNPs? I think this is really important to clarify and maybe also experimentally test.

iii. The authors showed that one of the previously identified SNPs (rs12973192) leads to a significantly lower binding affinity for TDP-43 than the wild type sequence (Figure 4F-I), while, in contrast, there is enhanced TDP-43 binding on rs12608932 SNP (Figure 4I). I find this result somewhat puzzling and I think that the authors need to further explore it. For instance, how does rs12608932 SNP perform on the in vitro assay shown in Figure 4H? Based on the iCLIP results, I would expect that the affinity will be higher in the risk SNP sequence, i.e. the opposite of what is shown for rs12973192 in the current Figure 4H.

iv. In the iCLIP experiment (Figure 4I), it is unclear to me how the authors normalized the levels of minigenes in the transfected cells to quantitatively compare TDP-43 binding on the different variants. This is necessary since I would expect some variability originating from transfection and which may significantly skew the conclusions of the iCLIP comparison.

4. Authors state that [...] although unlike the STMN2 CE, the UNC13A CE induces NMD, it was detected at similar levels to STMN2 CE in cortical regions, whilst STMN2 CE was more abundant in the spinal cord (Figure 3). What is the relevance of this finding? Could it be neuronal subtype-specific?

5. rs12973192 and rs12608932 SNPs are mainly associated with increased ALS risk (van Es et al., 2009; Nicolas et al., 2018) but also shown to contribute to FTD in sporadic ALS (Placek et al., 2019). However, CE inclusion seems to be more frequent in FTLD-TDP than in ALS-TDP (Figure 3). How can the authors explain this result?

6. I think that showing that UNC13A CE-containing transcript depends on nonsense-mediated decay for degradation is very important mechanistically (Figure 2F). What about the alternative splicing shown for UNC13B? Is this the same or a different mechanism of degradation?

Minor concerns:

1. I encourage the authors to reconsider or complete some of the references. For example:

- a. In the sentence "Genome-wide association studies (GWASs) have repeatedly demonstrated a shared risk locus between ALS and FTD within the crucial synaptic gene UNC13A, although the mechanism underlying this association has remained elusive⁴", reference 4 (van Es et al 2009 Nat Gen) states UNC13A locus is associated to ALS, but not FTD.
- b. In the sentence "ALS and FTD are pathologically defined by cytoplasmic aggregation and nuclear depletion of TAR DNA-binding protein 43 (TDP-43) in the vast majority (>97%) of ALS cases and in 45% of FTD cases (FTLD-TDP)⁵", reference 5 (Neumann et al 2005 Science) shows cytoplasmic mislocalization and aggregation of TDP-43, but the incidence of TDP-43 proteinopathies in ALS and FTD remains unreferenced.
- c. The sentence "The RNA-seq data used from TDP-43 depleted SH-SY5Y and SK-N-DZ neuronal lines (Figure 1I-L)" is missing a reference in the main text.
- d. In the legend of Figure 1, references to the latest ALS GWAS and the five TDP-43 knockdown datasets are missing.
2. I encourage the authors to clarify their methodology and to include all necessary details on the performed experiments in the main text whenever possible. For example:
- a. In how many samples/replicates was the RNA-seq on iPSC-derived cortical-like i3Neurons performed?
- b. How were the 179 CE identified or predicted?
- c. It is unclear to me where "the multiple binding peaks both downstream and within the body of the UNC13A CE (Figure 1D)" are coming from. Are those from previously published CLIP data?
- d. The authors state that "TDP-43 KD significantly reduced UNC13A RNA abundance in the three cell types with the highest levels of cryptic splicing." Which three cell types are they referring to?
- e. Unless I missed it, the cell type (HEK293T) in which iCLIP was performed is only mentioned in the methods and not in the main text/legend.
- f. When the authors analyzed a dataset of in vitro RNA heptamer/RBP binding enrichments, what was the followed pipeline? Did they take the heptamers that cover the SNP position and compare the RBP binding of heptamers with the risk SNP to heptamers without (healthy SNP)?
- g. In Figure 1B, authors say they used four controls and three TDP-43 depleted samples. However, in the methods sections, they claim it is three controls and four TDP-43 depleted samples.
- h. In Figure 1, are the sashimi plots showing these two representative samples, or a merge of all replicates?
- i. In Figure 1H, which kind of correlation test was performed? Spearman's? Information is missing in the corresponding legend.
- j. In Figure 4B, there are no error bars shown in the plot. Would it be possible to show the actual data points?
- k. In Figure 4I, where does the UGNNUG motif come from? Was it a prediction based on their own iCLIP data?
- l. In the Materials & Methods section:
- i. Did they use the STAR alignments as input to the splicing analysis with MAJIQ?
- ii. Analysis of published iCLIP data is unfinished. The section ends with an unfinished sentence: "The pipeline...".

iii. For the RNA-seq in i3Neurons, authors state that "one control replicate did not pass RNA quality controls and so was not submitted for sequencing." Does that mean they only had two controls instead of three, or three instead of four?

iv. In the iCLIP analysis of the minigene-transfected cells, how did the authors identify the cross-links? And why did they not use the iCount pipeline as they did for the published iCLIP data?

Referee #4

The manuscript titled "Common ALS/FTD risk variants in UNC13A exacerbate its cryptic splicing and loss upon TDP-43 mislocalization" by Brown and colleagues reported that loss of TDP-43 from the nucleus induces inclusion of a cryptic exon (CE) in the UNC13A transcript, which leads to the loss of UNC13A protein. Furthermore, ALS/FTD risk-associated SNPs within UNC13A genic region promote increase inclusion of the CE.

General comments:

Overall, this is a very nice study to link TDP43-dependent regulation of a CE within UNC13A transcript and strong risk-associated UNC13A SNPs, therefore elucidating the molecular mechanism behind the variants. The authors performed a series of carefully designed experiments to elucidate the molecular mechanism. The findings are novel and are of immediate interest to people in the neurodegeneration field. However, some of the statistical analyses need to be re-visited to support the conclusions.

Specific comments:

The statement of "UNC13A CE inclusion negatively correlated with TARDBP RNA levels" Figure 1H is questionable. First of all, the calculated p-value is not significant ($p=0.077$). More importantly, is this model suitable for the data points that were collected from multiple cell lines? Let's say if only iPSC MN data (6 data points) are used to run the regression, can the authors still conclude with a negative correlation?

Figure 1 (I, J, K, L) Can the authors add the statistical test? What can be concluded for UNC13A IR, UNC13B fsE and UNC13B IR in SH-SY5Y cell line (Figure 1J, K, L)? Some discussion could help.

"TDP-43 KD also decreased expression of UNC13A and UNC13B at the protein level, as assessed by quantitative proteomics with liquid chromatography tandem-mass spectrometry and western blot (Fig. 2D,E)." From Figure S1 (D, E), it appeared that both correct and UNC13A CE were detected at RNA level. Is there any evidence from LC/MS to show UNC13A CE at protein level?

Figure 3 (B, C) Can the authors add the statistical test?

Figure 3D Additional barplot of a quantitative measure of probe signals with the statistical test is recommended to show the significance.

Figure 4E May the authors switch to two-way ANOVA test (two categorical variables: shTDP43 treatment, minigene constructs) as the statistical test?

Minor:

Figure 2F, 4H: Please document the statistical test used.

In UNC13B, TDP-43 KD ...increased intron retention (IR) between exon 21 and 22 (Figure S1A). Some quantitative comparisons could help. Same for Figure S1B

Consistency of Figure 4A and Figure S4 (A,B). Any reason to include Kruskal-Wallis test results in FigureS4 (A,B), but not in Figure 4A (labeled with "Wilcoxon-Test" instead)?

Referee #5

- A. In this manuscript, authors identify an UNC13A cryptic exon inclusion as the direct consequence induced by TDP-43 pathology/depletion in ALS-FTD patients. The authors found that two known SNPs highly associated with ALS from previous GWAS studies directly affect the binding affinity and CE exon inclusion levels.
- B. This interesting finding links previous GWAS and genetics studies to the direct molecular functions and provides a very important advance in the field. The manuscript is extremely well done, and there are few comments.
- C. The approaches used in this study are valid. The data is of very high quality. The presentation is solid but could be further improved as suggested below.
- D. In general the statistical methods are well documented, with suggestions listed below.
- E. The conclusions seem robust, and involve large-scale validation using numerous orthogonal methods.
- F.
1. Fig 3B, the significant differences between control and ALS-TDP is striking, according to Fig S3A all motor cortex seem to express a similar level of UNC13A, however, many ALS-TDP patients do not show evidence of CE incorporation? Is the phenotype between those CE positive and CE negative patients' coloured differently? Are the samples showing no CE inclusion from patients without the UNC13A risk alleles?
 2. Page 5 paragraph 1, the quantitative traits look very strong, but rs12608932 has a population AF of 30% according to dbSNP. How can we reconcile such a high VAF with the rare ALS-TDP phenotype? Are there any rare variants found in any of the samples the authors analyzed?
 3. Fig S2D Did the authors sequence the bands with Sanger to confirm the splice junction?
 4. Fig. 3E there are many points at the left bottom corner. The number of samples in total should be indicated in the main text?
- G. Appropriate.
1. What was the rationale to choose UNC13 family proteins over the many other genes mis-spliced in FTLD-TDP iNeurons? Was there a joint probability or prior probability used to intersect genes with GWAS? It would appear from Fig. 1b that there are many other genes that would fulfill the same criteria.
 2. Check ref 14 and 49 format is different
 3. Fig. 2F indicate what test was used in the legend
 4. Fig. 3C makes the case that STMN2 and UNC13A cryptic PSI are cross correlated. Is the genomic architecture and sequence between the two genes?
 5. Fig. 3c also indicate the n numbers and p-value?
 6. Figs 3 and 4 only focused on UNC13A CE, did the authors evaluate other possible events in the UNC13 family as mentioned in Fig 2 and the discussion?
 7. Fig. 4 there is a tetranucleotide short tandem repeat (STR) polymorphism that is part of the UNC13B risk haplotype. The authors have shown that the 2R minigene is sufficient to mediate altered TDP-43 binding and CE incorporation, but how was the STR accounted for?

Author Rebuttals to Initial Comments:

We thank the reviewers for their constructive comments on our manuscript. We are delighted that all five reviewers appreciate the importance and potential impact of our work, and are pleased to report that we have made excellent progress in responding to nearly all of the reviewers' requests.

In this revised manuscript we have added the following new results and data:

- We have **confirmed the UNC13A CE specificity to TDP-43 knock-down with orthogonal techniques** in both iPSC neurons and patients:
 - We present evidence of the UNC13A CE in i³Neurons using Basescape *in situ* hybridisation;
 - We have extended our FTD/ALS post mortem *in situ* hybridisation experiments by doubling the sample number and performing quantitative analysis, confirming the CE specificity for TDP-43 pathology in patients.
- We have used **long-read RNA- sequencing to reveal a more comprehensive structure of the**

UNC13A CE transcripts. As raised by Reviewer One, although our Illumina-RNA-seq data show the presence of the CE throughout a wide range of cell models and patient tissues, a crucial outstanding question is whether the *UNC13A* CE co-occurs with an intron retention event downstream in the transcript. Use of Nanopore long-read RNA-seq demonstrates that the ***UNC13A* CE and intron retention events occur largely independently** in both cells and patient brains. This finding is key in order to consider potential future *UNC13A*-targeted splicing therapeutics.

- In our first submission we provided evidence that *UNC13A* CE transcripts were degraded by nonsense-mediated decay (NMD). Reviewer One appropriately asked us to confirm this important point by using an orthogonal approach. We have now performed *UPF1* knock-down (the gold standard approach for inhibiting NMD) in both neuronal cell lines and i³Neurons. Results **confirm *UNC13A* CE transcripts are degraded by NMD.**

- Although our work and the companion manuscript by Ma et al. provide strong evidence for loss of TDP-43 inducing the *UNC13A* CE, the threshold of TDP-43 loss required for this had remained unclear. We have now generated results from neuronal lines with different levels of TDP-43 knockdown, **clarifying the relationship between *UNC13A*, *UNC13B* and TDP-43 mRNA and protein levels.**

- Both our paper and Ma et al. show that the risk SNPs enhance *UNC13A* CE both in patients and by using minigenes in cells. One key question is the *mechanism by which the risk SNPs do this*. In our first submission we reported three lines of evidence showing that the SNPs directly impact on RNA binding: a) *in vitro* binding data of TDP-43 to heptamer sequences containing the risk SNP, b) isothermal titration calorimetry (ITC) experiments using recombinant TDP-43 RNA binding domains and the RNA sequences around the SNP, and c) iCLIP data showing TDP-43 binding to the *UNC13A* intron 20 in our minigenes. Results showed binding affinities perfectly in line with others reported in the literature (*Lukavsky PJ et al., Nat Struct Mol Biol 2013*), and **all showed specific and reduced binding when the CE risk SNP was present.**

We have now performed novel additional experiments to further test this hypothesis and add two further lines of supporting evidence. We have firstly performed novel iCLIP experiments that confirm **TDP-43 binding to the CE SNP site in endogenous *UNC13A***. Importantly, we then tested whether TDP-43 binding to the CE SNP region is important for splicing by manipulating the sequence surrounding the SNP. Since TDP-43 binding to RNA is dependent on RNA motifs, we mutated two single nucleotides adjacent to the SNP that are essential components of the TDP-43 binding motif. Both mutations independently enhanced CE splicing at levels similar to the risk SNP, **confirming that TDP-43 binding to this motif is directly involved in CE regulation.**

We are aware that our results from these TDP-43/RNA interaction experiments are somewhat different from experiments by Ma et al. It is important to note several methodological differences between our respective experiments: 1) We used shorter RNA sequences (14 and 18 nt). 2) Our ITC experiments were performed using truncated recombinant TDP-43 containing the two RNA recognition motifs. 3) We used ITC rather than EMSA. Importantly, whilst ITC was performed with the RNA binding region only of TDP-43 (aa 102-269), heptamer binding experiments with TDP-43 (aa 56-315) and our iCLIP, and minigene experiments make use of endogenous full length TDP-43 protein, all showed coherent results. These findings are the only minor difference with Ma et al., where they show a weaker binding affinity to the relevant regions. **As such, both sets of results will be valuable for readers interested in different aspects of these binding interactions.**

- Lastly, as requested by Reviewer Three, we have performed ITC experiments and show **TDP-43 binds tightly to the RNA containing the intronic SNP**, potentially with increased affinity for the risk variant.

Overall, our results in this revised manuscript are remarkably consistent with those of Ma et al., with only minor differences that do not impact upon the major findings presented in our manuscript: that **UNC13A harbours a cryptic exon strongly activated by loss of TDP-43, and that previously identified ALS/FTD-linked SNPs exacerbate this process.** We believe that our manuscript is now substantially improved and is essentially ready for publication.

In our point-by-point response we have kept **Reviewer comments in black, our response in blue, and new added text to the manuscript in red.**

Response to Reviewer 1

I may have missed it in the methods, but it wasn't clear to me how many independent iPSC lines this was actually done with. Related to this point have the iPSC-lines used been assessed for ALS/FTD relevant mutations?

The majority of the work was performed using the WTC11 line: “The iPSCs used in this study were from the WTC11 line, derived from a healthy thirty-year old male, and obtained from the Coriell cell repository.” The parent WTC11 lines for the i3Neurons have been fully genotyped and available on the Allen Cell Institute: <https://www.allencell.org/genomics.html>. The engineered daughter lines harboring TO-NGN2 and CAG-dCas9-BFP-KRAB were karyotyped to ensure no chromosomal rearrangements. We have checked the genotypes of WTC11 and confirmed they harbor no ALS or FTD relevant mutations and updated the methods to state this. WTC11 is heterozygous for the *UNC13A* common risk SNPs described in this study. We confirmed the presence of rs12973192 C/G with Sanger sequencing.

To confirm that our findings were not restricted to one iPSC line, we tested an independent iPSC line, NCRM5 (Luo et al. 2014 <https://doi.org/10.5966/sctm.2013-0212>, Tian et al. 2019 [https://www.cell.com/neuron/pdf/S0896-6273\(19\)30640-3.pdf](https://www.cell.com/neuron/pdf/S0896-6273(19)30640-3.pdf)). We confirmed via RT-qPCR that this line also produced *UNC13A* cryptic exons in the presence of CRISPRi depletion of TDP-43 that correlated with the degree of TDP-43 knockdown. These results are now reported in fig 1I,J and fig. S1D-G, and the following text has been added:

Results: “To confirm the CE was not restricted to neurons derived from a single iPSC line, we performed TDP-43 KD in independent i³Neurons using two different guides leading to different levels of TDP-43 KD (fig. S1D,E). CE expression was exclusive to TDP-43 KD in both lines, and correlated with the level of TDP-43 KD (Fig. 1I,J; fig. S1F,G).”

The complexity of the events detected makes it hard to know from short read data the actual transcripts being produced and I think this is important in terms of understanding mechanism and is soluble with targeted long read sequencing. Therefore I would suggest these experiments are performed.

We agree this is an important point, with the most relevant question being whether the two main splicing

events (cryptic exon and intron retention event) occur together or independently. We have performed targeted long read Nanopore sequencing, using RNA from SH-SY5Y cells with TDP-43 knockdown and FTLD-TDP patients. Our results show that in both cell lines and patient brain, the IR event is present in controls, whilst CE is not detected. When TDP-43 is knocked down, and in TDP-ALS/FTD cases, the CE is detected, and CE and IR mostly occur independently. We have updated the text as follows, as well as Fig 1K,L, fig S3A,B and fig S7A,B.

Results: “We next asked whether the *UNC13A* IR and CE events co-occur in transcripts. Using targeted long-read sequencing, we determined that although co-regulated, the *UNC13A* CE and IR occur largely independent from each other (fig 1K,L; fig. S3A,B).”

Results: “Targeted long-read sequencing of *UNC13A* in FTLD frontal cortex revealed that the CE and IR event can co-occur, but mostly are detected independently, similar to what was observed in SH-SY5Y (fig. S7A,B).”

Given that genuinely quantitative methods have been used to assess the UNC13A-CE inclusion transcripts, the UNC13A canonical transcript levels and protein levels, I think it would be important to really assess how well they correlate and formally model whether the protein level changes can be fully explained by the CE levels. Looking at the plots it's not clear to me that this is the case and perhaps this is unsurprising given the complex relationship between RNA and protein. However, this does raise the possibility of additional mechanisms and at the very least this should be explored and recognised.

The reviewer raises an important point, as indeed the *UNC13A* retained intron could also contribute to the decrease at the RNA/protein level, along with other yet to be uncovered regulatory mechanisms. We have now assessed *UNC13A* CE, *UNC13A* RNA and protein levels within SH-SY5Y cells with increasing amounts of TDP-43 knock-down. Results show that *UNC13A* CE starts being detected when TDP-43 decreases to 50%. The levels of CE are detectable and very low in this condition, and we have shown this is partly due to the fact that *UNC13A* CE is very efficiently cleared through NMD (fig S4J,K). We present these novel results in Fig 2C and fig S4 F,G,J,K and, as suggested by reviewer, we have toned down our statement and do not solely link *UNC13A* CE to *UNC13A* reduction in the text.

Results: “In order to assess the relation between TDP-43 reduction and *UNC13* splicing, RNA and protein levels, we assayed SH-SY5Y cells with increasing amounts of TDP-43 knockdown. We found that *UNC13A* loss paralleled TDP-43 loss, and when TDP-43 was no longer detected by Western blot analysis, less than 5% of *UNC13A* protein remained (Fig. 2C; fig. S4G), whilst *UNC13B* levels did not fall under 50%. *UNC13A* CE inclusion and IR increased after loss of TDP-43, but the CE could be detected only after TDP-43 KD is >50%, and *UNC13B* fsE and IR were not robustly detected until there was a greater than 90% loss of TDP-43 (Fig. 2C; fig. S4G).”

Results: “Interestingly, CHX treatment of SH-SY5Y cells with mild TDP-43 knockdown enabled detection of the *UNC13A* CE, supporting the notion that its occurrence can be underestimated due to its efficient degradation (fig. S4J,K). Taken together, our data suggests that TDP-43 is critical in ensuring the correct pre-mRNA splicing and maintaining normal expression of the presynaptic proteins *UNC13A* and *UNC13B*.”

I think the validity of studying splicing using nuclear RNA extractions needs to be addressed. After all the implication is that the authors are studying pre-mRNA/ partially processed RNA species and so there is a possibility that this isn't mis-splicing but a change in splicing dynamics.

We agree with the reviewer that interpretation of certain splicing changes based solely on nuclear

sequencing can make interpretation difficult. We do not believe this impacts our work, as throughout our paper all the sequencing data we have generated in the different cell culture experiments, as well as in the tissue bulk RNA-seq, have been performed on whole cell/whole tissue extracts. The only exception is when we re-analyse nuclear-RNA-seq data from Liu et al. in Figure 3A. We do agree that in this setting, due to the lack of NMD activity in the nucleus, the splicing ratio may not reflect that of the whole cell - we have added a clarification in the text.

Results: “Strikingly, while lack of NMD activity in the nucleus means that nuclear splicing ratio may not reflect that of the whole cell, in some cases the *UNC13A* CE percent spliced in (PSI) reached 100% (Fig. 3A).“

I think what is meant by detection of the UNC13A CE should be clarified with some account taken of the likely change in cellular composition of the tissues between cases and controls which could be a confounding factor as well as read depth, RIN differences etc. From the methods, it is not clear to me that this has occurred.

We have clarified in the methods the threshold for the cryptic exon’s detection in the bulk RNA-seq NYGC ALS/FTD dataset.

Further, in addition to providing the differences in *UNC13A* TPM between cases and controls, we have included fig S5B,C,D illustrating the differences between cases and controls on library depth, RIN, and cell type composition.

Methods: “CE was considered detected in a sample if there was at least one uniquely mapped spliced read supporting either the short CE acceptor or the CE donor.”

Results: “There were no systematic differences across tissues between controls and ALS/FTLD-non-TDP and ALS/FTLD-TDP cases on confounding factors such as library depth, RIN (RNA integrity number), or cellular composition which could explain *UNC13A* CE specificity (fig S5A-D).”

“The lower detection rate in ALS versus FTLT is likely due to the lower expression of UNC13A in the spinal cord (fig. S3A). “ I am not sure that this is a valid conclusion since it doesn’t consider the role of NMD in removing UNC13A- CE containing transcripts and NMD also has regional differences in expression.

We have updated this claim as follows, and included a citation (Zetoune, A. B. *et al.* Comparison of nonsense-mediated mRNA decay efficiency in various murine tissues. *BMC Genet.* **9**, 83, 2008) on the regional differences of NMD in mammalian tissues:

Results: “The lower detection rate in ALS versus FTLT is possibly due to the lower expression of *UNC13A* in the spinal cord (fig. S5A), although differences in NMD efficiency between cortical and spinal regions could also affect detection rate”

Also when something is not detected much more controls on what that means

We have included analyses on ALS/FTD-TDP cases comparing library depth, RIN, cellular composition, and read length by *UNC13A* CE detection status. These are presented in fig S6 A,B,C,D.

Results: “*UNC13A* CE was more likely to be detected in bulk samples which had been sequenced with 125 bp rather than 100 bp paired end, but other technical factors did not systematically effect detection(fig S6A-D)”

I am not sure about the validity of using STMN2 cryptic splicing to normalise. If it is a proxy for the levels of TDP43 pathology why not just quantify that directly and include it within a single linear model.

We do agree that quantitative TDP-43 pathology would be useful, but pathology assessment and RNA analysis require different tissue processing (FFPE vs tissue freezing) and therefore need to be carried out on different brain samples. Our approach, based on our previous work (Prudencio, 2020), has the advantage of assessing TDP-43 pathology and *UNC13A* mis-splicing within the same sample. Lastly, a standardized quantitative measure for TDP-43 pathology still needs to be developed, and the >1,300 brain samples included in our analysis were collected in 8 different brain banks making such an assessment outside the scope of this project. In the text, we now clarify that *STMN2* cryptic splicing is used to normalise for the level of TDP-43 loss of function - not pathology -, and therefore assess whether the risk SNPs impact on *UNC13A* splicing in ALS/FTD cases.

Results: “To ensure that this was not simply due to more severe TDP-43 loss of function in these samples, we normalised *UNC13A* CE by the level of *STMN2* cryptic splicing, which is a well-established product of TDP-43 loss of function.”

In addition, again it isn't clear whether the authors are taking into account common known covariates (ethnicity, sex, age etc) assessed for unknown covariates and tissue composition. This is in effect eQTL analysis and this is all very standard for the field. With this in mind, I'm not sure why a Wilcoxon test has been used and the p-value doesn't seem particularly impressive in that context.

We have performed an analysis on the levels of *UNC13A* CE level, including into a single linear model tissue composition, the number of risk alleles, *STMN2* CE PSI in a sample, age, sex, mutation status, and sequencing platform, e.g. read length. We find that *STMN2* CE PSI, astrocytes, and the number of risk alleles are significantly positively associated with the level of *UNC13A* CE PSI, whereas tissue - lumbar spinal cord - is significantly negatively associated with *UNC13A* CE PSI. We have included the results in Table S4 with text as follow:

“Linear regression on the *UNC13A* CE PSI in ALS/FTLD-TDP samples with known covariates including age, sex, RIN, and cell type composition revealed significant positive relationship between *STMN2* CE PSI, level of astrocytes, and the number of risk alleles at *rs12973192(G)* (Table S4).”

*(re: Targeted analysis of *UNC13A* risk SNP expression)*

This is an ASE analysis and so of course it's not just the p-val that matters but the direction of effect and it is highly prone to mapping biases or biases in primer design when a capture is done so more information is needed to ensure this result is valid. In addition the exact p-value is not provided and there is some controversy over the use of a binomial test here.

Given the small number of samples, I think precise data should be provided on all samples in a tabular form. Further information is required to convince me of the specificity of the amplicons produced and that this process was not directly biased by the SNPs of interest which would make an ASE analysis potentially invalid.

We did not perform a capture step, and none of the primers anneal to the SNP itself or its flanking nucleotides, making it unlikely that SNP-specific specific biases would have been introduced by our

approaches. Furthermore, our use of UMIs to remove PCR duplicates means that we control for any PCR amplification biases introduced by the SNPs.

We provide below a diagram and description of our approach: to generate specific amplicons, we developed two nested PCR approaches (see diagrams below) which greatly increased specificity, as assessed by DNA electrophoresis using RNA from dox-treated SH-SH5Y cells with accompanying controls, while retaining UMIs to enable removal of PCR duplicates. To improve purity further, we gel purified bands of the expected size before sequencing. The net result was an extremely pure amplicon library for both approaches, typically with more than 99% (in some cases more than 99.9%) of mappable reads aligning to UNC13A.

We agree that data regarding the specific samples should be reported, and we have now added this information as Table S5.

With our targeted sequencing approach, we are using a single-tailed binomial test to ask whether in heterozygous individuals, the risk SNP allele is more represented within CE transcripts; as such, the p-value of the test would not be significant in the other direction. We have now amended, as requested, Fig 4B to include the exact p-values of the single-tailed binomial test.

Have the authors considered the problem of mapping biases, and if so how have they accounted for that given that these biases have the potential to create false positive ASE results.

We have remapped the reads using the WASP option in STAR, including a VCF containing rs12973192 (the only relevant variant within the amplicons), and selected for reads passing the WASP filter. The final results were near-identical to before - this suggests that mapping biases do not significantly influence these results. We have added the following to the methods:

Methods: “To control for mapping biases, a VCF containing rs12973192 was included and alignments that failed to pass WASP filtering were ignored.”

I do not understand the rationale behind the use of a subsample of reads in this way.

We have repeated the analysis using all reads, finding identical results - this is expected due to the large number of PCR duplicates per unique cDNA in these libraries. We have updated the methods to reflect this.

have the authors checked to assess other SNPs/variants in LD in the region and that could include repeats.

We have now included analyses of another variant in LD with the risk SNPs: STR *rs56041637*. We have assessed whether it has an impact in enhancing the *UNC13A* CE inclusion and results of these new experiments are now included in novel Fig 4D,E, and text has been added as follows:

Results: “To directly assess whether the risk SNPs increase CE inclusion, we performed minigene experiments. Using two minigenes containing *UNC13A* exon 20, intron 20, and exon 21, with and without the two ALS/ FTLN-linked variants, we determined that the risk variants enhanced CE upon TDP-43 loss (fig. S11C). To specifically examine whether the CE SNP, intronic SNP or short tandem repeat expansion *rs56041637* – which is in linkage disequilibrium with the two SNPs²⁹ – are responsible for promoting the CE inclusion, we generated minigene variants featuring different combinations of the three aforementioned genomic variants (Fig. 4C; fig. S11D). Via quantitative analysis of RT-PCR products, we found that both the CE SNP and, to a lesser extent, the intronic SNP independently promoted CE inclusion, with the greatest overall levels detected for the 2R minigene (Fig. 4D,E). ”

Figure 1. Panels A) and B): The impact on splicing looks much more significant than the impact on expression and it's not at all clear to me that UNC13A would have been flagged through expression.

We agree with the reviewer that the impact on splicing is substantially larger than the impact on *UNC13A* transcript expression, and had *UNC13A* not been previously linked to ALS risk it would not have been flagged. *UNC13A* is significantly downregulated at the RNA level with a p-adjusted value of 1.141770×10^{-05} and a log2 fold change of -0.25 in the i³Neurons. It is also downregulated at the ribosome footprint level, with log2 fold change of -0.78. Importantly, the effect of TDP-43 depletion on protein expression of *UNC13A* is substantially greater than transcript expression, providing a plausible rationale for a loss of function mechanism despite modest reduction at the transcript levels. We have amended the text and Fig 2A caption to clarify that the downregulation is highly significant:

Fig. 2A: “Significance levels reported as * (p<0.05) ** (p<0.01) *** (p<0.001) **** (p <0.0001).”

Results: “TDP-43 KD significantly reduced *UNC13A* RNA abundance in the three cell types with the highest levels of cryptic splicing (FDR < 0.0001; Fig. 2A, Fig. 1I). Likewise, *UNC13B* RNA was significantly downregulated in four datasets (FDR < 0.0001) (Fig. 2B).”

In fact there appear to be better candidates, what are they and have they been investigated or assessed in any formal way eg. just intersecting with genes linked to ALS/FTD through GWAS or Mendelian disease.

We have now intersected all ALS genes (<https://panelapp.genomicsengland.co.uk/panels/263/>) and early onset dementia (<https://panelapp.genomicsengland.co.uk/panels/265/>), as well as all the genes in the most recently published ALS GWAS (Nicolas, 2018) with gene expression and cryptic splicing status in the i³Neurons, and found that *UNC13A* is the only gene with a cryptic splice junction. We have included this analysis as Table S1 and updated the text in the results and methods as follows:

“We intersected splicing, expression, ALS GWAS¹⁷ risk genes, and diagnostic panel genes for ALS/FTD¹⁸. Of the 179 CE-harboring genes, only the synaptic gene *UNC13A* was also an ALS/FTD risk gene (Table S1; Fig. 1C,D).”

(H) Correlation between relative TARDBP RNA and UNC13A CE PSI across five TDP-43 knockdown datasets: I find this plot unconvincing. If the authors accounted for differences in cell type, I don't think there would be any correlation and for the iPSC derived motor neurons it would be flat.

We have removed this panel. To address this question directly and in a controlled manner, we used two companion approaches in SH-SY5Y cells and i³Neurons. We assessed 8 unique CRISPRi sgRNAs targeting TDP-43 in i³Neurons and identified one guide with attenuated TDP-43 knockdown. We measured *UNC13A* cryptic splicing and *UNC13A* protein levels via RT-PCR and whole cell proteomics, and observed a robust correlation between TDP-43 knockdown and both *UNC13A* cryptic splicing and protein levels. We also used clonal dox-inducible TDP-43 knockdown SH-SY5Y lines with different levels of TDP-43 loss. We analysed TDP-43, *UNC13A* and *UNC13B* RNA and protein levels along with *UNC13A* splicing and show that *UNC13A* loss parallels that of TDP-43. These results are detailed below and shown in Fig. 1J, Fig. 2B,C and fig S4 F,G.

Results: “In order to assess the relation between TDP-43 reduction and *UNC13* splicing, RNA and protein levels, we assayed SH-SY5Y cells with increasing amounts of TDP-43 knockdown. We found that *UNC13A* loss paralleled TDP-43 loss, and when TDP-43 was no longer detected by Western blot analysis, less than 5% of *UNC13A* protein remained (Fig. 2C; fig. S4G), whilst *UNC13B* levels did not fall under 50%. *UNC13A* CE inclusion and IR increased after loss of TDP-43, but the CE could be detected only after TDP-43 KD is >50%, and *UNC13B* fsE and IR were not robustly detected until there was a greater than 90% loss of TDP-43 (Fig. 2C; fig. S4F).”

*Couldn't this form of analysis be broadened in order to look for correlations between *STMN2* CE PSI and other apparently abnormal splicing events? Has this been done? It could throw up other interesting candidates and would also enable the correlation for *UNC13A* to be assessed more robustly.*

While systematic correlations between the PSIs in the NYGC bulk cohort of all cryptic hits discovered in this analysis are outside the scope of this paper, it is indeed an intriguing question and an active line of work in the lab now. As additional proof of concept, we have included in fig S10A,B correlations between two additional well known TDP-43 regulated cryptic targets, *RAP1GAP* and *PFKP*.

Results: “*STMN2* CE PSI correlates with the cryptic PSI of other well-known TDP-43 induced CE, such as those in *RAP1GAP* and *PFKP*⁹⁻¹¹ (fig. S10A,B) and correlates with phosphorylated TDP-43 in patient samples¹³”

Is 17 days sufficient for neuronal maturation? It seems short to me, but I am not familiar with iPSC protocols.

Using a longitudinal RNAseq dataset from control i³Neurons, we analyzed the normal expression of a subset of genes that formed CEs in TDP-43 depleted conditions, and were reliably quantified over time. Following 17 days of differentiation, we find little further change in gene expression by RNA-seq, suggesting that additional maturation of neurons would not have resulted in a substantial difference in CE identification or quantification. In the heatmap below, we show longitudinal expression data from Tian et al., 2019 PMID 31422865. Expression values are averaged, normalized to day 17 and log

transformed. Genes shown are from the GO term “neuronal maturation,” highlighting how the transcriptome of our iPCS-derived neurons changes little after further maturation.

Globally, we also observe that RNA abundances show little fluctuation, as you can see in the below graph of the correlation between log transformed RNA abundance at day 17 and day 38 (Adjusted R-squared: 0.8987, p-value: < 2.2e-16).

The read lengths used are rather short for a splicing assessment and I see no information on the actual library construction method used.

Paired-end libraries of 75 bp may be slightly shorter than preferred for splicing assessment, but are well within acceptable lengths for quantification of differentially expressed genes, and detection and quantification of both annotated and novel splice junctions - please see <https://genomebiology.biomedcentral.com/articles/10.1186/s13059-015-0697-y>.

We have updated the methods to include library construction method:

Methods: "Sequencing libraries were prepared with polyA enrichment using a TruSeq Stranded mRNA Prep Kit (Illumina) and sequenced (2x75 bp) on an Illumina HiSeq 2500 machine"

Also I note data pooling was performed across biological replicates, why?

We apologize for the confusion. Different i³Neurons wells (grown in 96-well dishes) were used as survival is best in this setting, and multiple wells were needed to generate sufficient quantities of material for experimental requirements, but no data pooling was performed.

This seems a very generous definition of differential expression. If a more convention FDR of at least <0.05 was used would UNC13A be differentially expressed at all?

We have updated the main text to reflect that the FDR for UNC13A was below 0.0001 (this is also available in Data S1). The standard cutoff for differential expression with DESeq2 is a p-adjusted value of 0.1 - please see the vignette - <http://bioconductor.org/packages/release/bioc/vignettes/DESeq2/inst/doc/DESeq2.html#differential-expression-analysis>. We also note that reduction in *UNC13A* transcript was confirmed by QPCR in multiple independent experiments and in multiple knockdown cell types (fig S4A-D).

MAJIQ is used for this analysis but then a later analysis uses LeafCutter, why when both these methods address the same question?

In our paper, we use MAJIQ in a hypothesis free analysis on a small sample size for differential splicing analysis. When LeafCutter is used later on, as we describe in the methods, it is used to cluster junctions together across hundreds of samples, a task that MAJIQ does not scale well to. Here, we are not using LeafCutter to perform differential splicing analysis, but instead to cluster together and retrieve raw junction counts from the same introns.

The GENCODE build is different here to that used in the RNAseq analyses. I think it would be important to maintain consistency.

In the RNA-seq and iCLIP analysis, the reference annotation GTF is used to assign RNA-seq reads or cross-linked reads to a given gene. While it would be ideal to maintain consistency in exact versions of GENCODE annotations between all analysis, minor updates to GENCODE definition typically add minor isoform updates, but do not change the genomic locus of well studied genes, such as *UNC13A* or *UNC13B*. In addition, while a reference annotation is used during STAR's index building, STAR outputs splice junctions for both annotated and unannotated splice junctions. As we use all splice junctions, including novel

junctions, even though the reference may change, this would not affect our analysis. In any case, we note that both the ENSEMBL gene minor version and transcripts for *UNC13A* and *UNC13B* are identical between the three GENCODE annotations used across our team, and now specify this in Methods section.

gene	transcript	gencode version	transcript	gencode version	transcript	gencode version
UNC13A - ENSG00000130477.15	ENST00000519716.6	v30	ENST00000519716.6	v31	ENST00000519716.6	v34
	ENST00000551649.5		ENST00000551649.5		ENST00000551649.5	
	ENST00000552293.5		ENST00000552293.5		ENST00000552293.5	
	ENST00000550896.1		ENST00000550896.1		ENST00000550896.1	
	ENST00000523229.1		ENST00000523229.1		ENST00000523229.1	
	ENST00000517497.2		ENST00000517497.2		ENST00000517497.2	
	ENST00000601528.1		ENST00000601528.1		ENST00000601528.1	
	ENST00000619578.4		v30		ENST00000619578.4	
ENST00000617908.4	ENST00000617908.4	ENST00000617908.4				
ENST00000396787.5	ENST00000396787.5	ENST00000396787.5				
ENST00000635942.1	ENST00000635942.1	ENST00000635942.1				
ENST00000378495.7	ENST00000378495.7	ENST00000378495.7				
ENST00000634487.1	ENST00000634487.1	ENST00000634487.1				
ENST00000378496.8	ENST00000378496.8	ENST00000378496.8				
ENST00000485086.1	ENST00000485086.1	ENST00000485086.1				
ENST00000637271.1	ENST00000637271.1	ENST00000637271.1				
ENST00000636694.1	ENST00000636694.1	ENST00000636694.1				
ENST00000481299.1	ENST00000481299.1	ENST00000481299.1				

Cycloheximide will not just affect NMD and therefore I think an alternative approach would be beneficial. We agree a second approach is beneficial and have now confirmed our results using another well-established approach to inhibit NMD: the knock-down of the key NMD factor *UPF1*. We performed TDP-43/*UPF1* double knock-down in both i³Neurons and SH-SY5Y cells, and, similarly to our experiment with cycloheximide, we assessed the levels of *hnRNPL* (known NMD target serving as positive control), *UNC13A*, *UNC13B*, and *STMN2*. In line with our cycloheximide experiment, both *UNC13A* and *UNC13B* showed an increase after *UPF1* knock-down, whilst levels of *STMN2*, which is not expected to be an NMD target, did not increase. We present these results in Fig. 2D, and with novel supplementary figures (fig. S4H,I), and have also amended the Results and Methods sections accordingly.

Results: “To assess whether the degree of *UNC13A* CE expression was underestimated due to efficient transcript degradation, we investigated whether it promoted nonsense-mediated decay (NMD), as predicted by the presence of a novel premature termination codon (PTC). Knockdown of either the key NMD factor *UPF1* or cycloheximide (CHX) treatment –which stalls translation and impairs NMD – increased *UNC13A* CE and *UNC13B* fsE, which also leads to a PTC at the beginning of exon 11, confirming they were both targeted by NMD (Fig 2D;fig. S4H,I). Conversely, CHX and *UPF1* knockdown did not alter levels of the aberrant *STMN2* transcript, which was not predicted to undergo NMD (Fig. Fig 2D;fig. S4H,I).”

It is not clear to me that GAPDH would be an appropriate reference, has this been checked?

We selected *GAPDH* as endogenous control after the assessment, using RefFinder (<https://www.heartcure.com.au/reffinder/>), of *GAPDH* along with other commonly used transcripts *ACTB* and *HPRT*. We identified *GAPDH* as the most stable endogenous control across our conditions of interest. This has now been clarified in the Methods section, as follows:

Methods: “Using RefFinder (<https://www.heartcure.com.au/reffinder/>), we identified *GAPDH* as the most stable endogenous control across our conditions of interest.”

Furthermore, we also use a *GAPDH*-independent approach, where we assess CE vs correctly spliced bands, as reported in Figure S4I for i3Neurons, and reported below for SH-SY5Y experiments. When *hnRNPL*, *UNC13A*, *UNC13B* and *STMN2* RT-PCR performed after treatment with cycloheximide (CHX, right 4 lanes), the ratio of NMD-sensitive to NMD-insensitive isoforms of *hnRNPL* (positive control), *UNC13A*, and *UNC13B* are clearly increased compared to control (DMSO), whereas the ratio is unchanged for *STMN2*, whose aberrant transcript is not expected to undergo NMD. These results are concordant with the results obtained using *GAPDH* as endogenous control, supporting the validity of the experiment - and are shown below.

Are the replicates biological or technical?

They are biological, and we have clarified this in the text.

There is clearly batching with different read lengths and total RNA has been used with ribodepletion which means you expect more pre-mRNA reads which complicate analyses. It is not clear to me how these issues are accounted for in the analysis. Furthermore, the range in read depth is huge and for some samples I would question whether this is indicative of a library/technical failure – certainly I find it hard to see how reliable the data would be. These issues need to be formally addressed and this may require new analyses.

We have now addressed the point raised in this comment with the additional fig. S6.

This is the third GENCODE version being used in the paper. Either the authors should standardise their use of GENCODE versions or demonstrate that this does not impact on their findings.

We have now clarified in the methods that annotations are identical across the three GENCODE annotations used.

Methods: “To ensure consistency between RNA-seq, re-analysis of published iCLIP data, and the NYGC ALS Consortium RNA-seq cohort, we confirmed that both the ENSEMBL gene minor version and transcripts for *UNC13A* and *UNC13B* are identical between the three GENCODE annotations used across our team.”

LeafCutter is the second splicing tool being used what is the justification? Furthermore, it isn't clear to me that the exon-exon junction read filtering for MAJIQ and LeafCutter are the same and this is also an issue.

Please see the previous answer, where we clarify that the tools are being used to accomplish different tasks.

This is a semi-quantitative analyses, but there are methods available to make this quantitative. Those methods should be used.

We agree this is an important point. We have doubled the number of cases analysed and made these analysis quantitative. We have updated the results, Fig. 3 and the methods.

Results: “We next investigated whether *UNC13A* CEs could be visualised by *in situ* hybridisation (ISH) in FTLD patient brains, using the same probe that detected *UNC13A* CEs in iPSC-derived neurons, We detected red foci in cortical neurons at a significantly higher frequency in FTLD-TDP cases relative to both neurologically normal controls (Kruskal-Wallis test, $p = 0.021$) and non-TDP (tau) FTLD cases ($p = 0.010$) (Fig. 3D).”

Methods: “Hybridised sections were imaged and analysed blinded to disease status. Slides were scanned using an Olympus VS120 slide scanner at x20 magnification and equal sized (34.5 mm^2) regions of interest were extracted from the centre of each section. The total number of red foci, which should identify single transcripts harbouring the *UNC13A* CE event, were manually counted in ImageJ. Foci frequency was background-corrected by subtracting the signal obtained with the negative control probe in the same experiment.”

I am not sure how this sits with the editorial policy on data release though I am sympathetic to the issues around data release timing from big consortium projects. Maybe genotyping has been performed and this is available in the public domain?

While full genotyping is not publically available for these samples, the relevant genotypes for each of the samples are available on our GitHub repo - see https://github.com/frattalab/unc13a_cryptic_splicing/tree/main/data/nygc_junction_information.csv

Response to Reviewer 2.

1. Figure 4: in what samples was the analysis in Figure 4A, 4SA-D made?

These were performed in ALS/FTLD-TDP cortical samples with at least 30 junction reads at the *UNC13A* cryptic locus. We have clarified this in the axis-label of Figure 4A - as well as in the methods:

Methods: “Only samples with at least 30 spliced reads at the exon locus were included for correlations.”
Figure 4A: “only samples which were concordant for genotype at *rs12973192* and *rs12608932*, had both *STMN2* and *UNC13A* CE detected, and had at least 30 spliced reads at the exon loci were included in the analysis.”

2. Figure S4 A-D: There seems to be some issue with the figures, or else I do not follow. Panels A and B have different titles, but refer to the same SNPs (title of A: ‘CE SNP (*rs12973192*)’, title of B: ‘Intronic SNP (*rs12973192*)’). They are supposed to refer to different SNPs, which is also indicated in the legends. But these graphs appear to contain, essentially, the same data, only that the data points are scattered differently on the x-axis, making them look somewhat different. The P values are also different. Graphs C and D appear identical. Non the less, the P-values are not the same. This should be carefully checked.

We apologize, there was a typo in fig. S4B. Additionally, as the data is almost the same, the differences are due to the single discordant sample between *rs12973192* and *rs12608932* in the cohort. As the result is duplicative - we have opted to only show samples who are concordant genotypes, as in Fig. 4A, and have replaced fig. S4A-D.

3a. While NMD is proposed as the mechanism to explain reduced RNA levels of *UNC13A*, no such analysis exists for *UNC13B*.

We agree this is an important point - we have extensively expanded our NMD analysis: we have added a second approach to inhibit NMD, *UPF1* silencing, and we have included i³Neurons in our analyses. The results confirm *UNC13A* CE undergoes NMD, and that this occurs also for *UNC13B*, as shown in Fig. 2D and fig. S4H,I. We also explained this in the Materials and Methods sections.

Results: “To assess whether the degree of *UNC13A* CE expression was underestimated due to efficient transcript degradation, we investigated whether it promoted nonsense-mediated decay (NMD), as predicted by the presence of a novel premature termination codon (PTC). Knockdown of either the key NMD factor *UPF1* or cycloheximide (CHX) treatment –which stalls translation and impairs NMD – increased *UNC13A* CE and *UNC13B* fsE, which also leads to a PTC at the beginning of exon 11, confirming they were both targeted by NMD (Fig 2D;fig. S4H,I)”

Methods: “The *UNC13B* experiment was subsequently performed, following the same method.”

3b. *UNC13B* analysis in patient-derived material is missing. In fact, after the first section, *UNC13B* is ignored. For the completeness of the presentation and for a better mechanistic understanding, I find this is necessary.

We have now added a novel figure and included *UNC13B* analysis in patient material in fig. S8 A,B,C. Whilst

the *UNC13A* CE is never detected in control brains, making the detection of even few reads in TDP-ALS/FTD brains very specific, the *UNC13B* fsE is present in transcripts that are normally expressed in brain, making it therefore hard to detect potential changes due to pathological neurons in bulk RNA-seq. In control iPSC neurons, this fsE in *UNC13B* does not occur; it is possible that baseline fsE in *UNC13B* is cell-type dependent, hence bulk-RNAseq is detecting baseline *UNC13B* fsE expression from diverse cell types, or alternatively that *in vitro* studies do not accurately model *UNC13B* fsE baseline expression, in contrast to the accurate *in vitro* modeling of baseline and pathological *UNC13A* CE expression. We have also complemented Figure 2 NMD and TDP-43 dosage analyses with *UNC13B* data. The focus of the paper remains *UNC13A*, as the genetic data directly supports the involvement of *UNC13A* CE in ALS and FTD.

Results: “We next assessed the expression of the *UNC13B* fsE across the NYGC dataset. We did not detect a specific increase in pathological ALS-TDP and FTLD-TDP tissues. However, the presence throughout control and ALS/FTD brains of a shorter isoform, which was absent in our *in vitro* experiments and includes the fsE, may be masking underlying changes (fig. S8A-C)”

4. Can the authors provide some evidence for reduced protein expression in patient-derived material?

UNC13A is expressed throughout neurons, and TDP-43 mislocalisation occurs in 5-10% of ALS/FTD neurons. (Liu EY et al., Cell Rep 2019). Hence, detecting a change in bulk tissue analyses may not be possible. Nonetheless, we have performed western blots on lysates from the frontal cortices of 4 controls, 4 FTD-A patients, 3 FTD-B patients, 5 FTD-C patients, and 4 Alzheimer's disease patients. There is a non-significant change in *UNC13A* when normalized to BIII-tubulin levels (left and middle panel, below). The composition of the cell types (neurona/non-neuronal) in the samples is not easily controllable and there are large variations in the amount of *UNC13A* detected between samples. We note that the levels of *UNC13A* closely match the levels of the 250 kDa brain specific isoform *UNC13B* which would not be affected by the fsE. Levels of stathmin-2 are also variable between samples and do not show a difference between the control and FTD groups. [Redacted]

5. For clarity of the proposed model: the risk alleles are also present in healthy people, with no TDP-43 pathology. TDP-43 reduction in binding affinity to *UNC13A* RNA is therefore expected to occur in them as well. However, the authors did not detect CE inclusion in healthy individuals. Why in healthy people, who carry the risk alleles leading to impaired TDP-43 binding, no effects are seen? The authors propose that the CE inclusion occurs below a certain TDP-43 threshold. Can this be substantiated at the RNA or protein level? For example, by providing data for the time course of CE inclusion in patients? This will aid in

supporting the statement that ‘severe nuclear depletion of TDP-43 in the end stage disease induces CE inclusion...’ (discussion). It would be fascinating to align such data to the time course of cell degeneration.

The reviewer raises a very important point. Although the risk SNPs decrease TDP-43 binding affinity to *UNC13A* pre-mRNA, when TDP-43 is present in normal levels it still is able to bind *UNC13A* RNA and inhibit CE inclusion. In support of this, we have now performed *UNC13A* CE splicing analysis in clonal SH-SY5Y lines with different levels of TDP-43 loss. We show that *UNC13A* reduction parallels that of TDP-43. We have added this information in the results section and in Fig. 2C. We have also made changes to our description of the model in the discussion to make this important point clearer.

Results: “In order to assess the relation between TDP-43 reduction and *UNC13* splicing, RNA and protein levels, we assayed SH-SY5Y cells with increasing amounts of TDP-43 knockdown. We found that *UNC13A* loss paralleled TDP-43 loss, and when TDP-43 was no longer detected by Western blot analysis, less than 5% of *UNC13A* protein remained (Fig. 2C; fig. S4F), whilst *UNC13B* levels did not fall under 50%. *UNC13A* CE inclusion and IR increased after loss of TDP-43, but the CE could be detected only after TDP-43 KD is >50%, and *UNC13B* fsE and IR were not robustly detected until there was a greater than 90% loss of TDP-43 (Fig. 2C; fig. S4G).”

Discussion: “In this model, when nuclear TDP-43 levels are normal in healthy individuals, TDP-43 efficiently binds to *UNC13A* pre-mRNA, even in the presence of risk SNPs, thus preventing CE splicing.”

6. The authors conclude by stating ‘Excitingly, *UNC13A* provides a generalizable therapeutic target for 97% of ALS and approximately half of FTD cases.’ Considering the devastating diseases involved, and the impact such a sentence may have, such a statement requires more support – what are these numbers based on? *Munc13s* are terrible targets for pharmacology - manipulating the function of the *Munc13-1* protein is likely to affect the vast majority of synapses in the brain and out of the brain, potentially leading to a massive imbalance in neurotransmission. Genetic strategies to correct for the deleterious SNPs is not likely to prevent disease, but rather to delay disease progression. Are the authors proposing something in this direction? Please clarify.

We agree that targeting the *Munc13* proteins themselves would be extremely difficult and potentially deleterious. We were instead suggesting the use of splice-switching antisense oligonucleotide (SSO) therapies to block the cryptic splicing and promote correct splicing of the *UNC13A* pre-mRNA, which conceptually would selectively restore *UNC13A* expression only in settings of CE formation in diseased neurons. SSOs are a clinically proven method for modulating splicing in neurodegenerative disorders; for example, Nusinersen is an FDA-approved SSO for the treatment of spinal muscular atrophy. We have updated the main text to clarify:

Discussion: “The *UNC13A* CE is thus a promising target for splice-switching antisense oligonucleotide therapies, potentially applicable to 97% of ALS and approximately half of FTD cases in which TDP-43 mislocalization occurs.”

Response to Reviewer 3

1. My main concern is the lack of functional validation of the effect of *UNC13A* decrease in adult human neurons. While I appreciate that this is technically challenging, I think it is important to show that this direct effect of TDP-43 loss of function on *UNC13A* levels has significant consequences for neuronal maintenance and function. In my view, this is necessary to support the claimed therapeutic potential of the work. Is the loss of *UNC13A* the culprit of neuronal death? Since *STMN2* CE inclusion levels are always at least as high as those for CE inclusion in *UNC13A* and loss of *STMN2* has been proven to be deleterious (Melamed et al., 2019), how could the relevance of *UNC13A* loss for cell survival be differentiated from *STMN2* loss? Can reintroduction of *UNC13A* levels restore neuronal survival? Without answering some of

these key questions, I think UNC13A levels may be a potential diagnostic tool, but not (yet) a therapeutic target.

We agree that direct functional evidence that decreased *UNC13A* mechanistically contributes to neurodegeneration would underline the importance of our study. However, for the reasons outlined below, we believe that generating such clear-cut data is not possible, and that any functional data we could feasibly obtain would not change the main conclusions of our manuscript. Of importance, *UNC13A* loss does not appear to affect neuronal survival in otherwise healthy neurons, as shown in i³Neurons (CRISPR screen data from Kampmann lab, Tian et al., 2021) and supported by mice lacking *Unc13a*, which have normal neuronal morphology (Varoqueaux et al., 2005). It would therefore not be possible to demonstrate *UNC13A*-mediated rescue via a survival assay.

UNC13A plays an essential role in synaptic vesicle priming and release. It is possible that extensive electrophysiological and/or synaptic vesicle imaging experiments, in the setting of *UNC13A* overexpression, may show restoration of these specialized synaptic functions in TDP-43 knockdown neurons. However, we note that multiple other synaptic proteins also form CEs or other pathological splicing events in settings of TDP-43 KD (SYT7, KCNQ2, CAMKII β , among others). We therefore anticipate that teasing apart the specific impact of *UNC13A* on neuronal function in TDP-43 deficient neurons - while certainly a critical question to address regarding therapeutic development - will be challenging to address experimentally, and beyond the scope of this manuscript.

Regarding the potential for *UNC13A* as a therapeutic target, what really sets the *UNC13A* cryptic exon apart from all other TDP-43 CEs is the striking abundance of genetic data implicating *UNC13A* in ALS risk. Human genetics data obtained from vast numbers of ALS patients show that disease-associated *UNC13A* SNPs significantly impact disease progression, therefore strongly supporting a pathogenic role for the cryptic exon in *UNC13A* that we identified. In our manuscript we include strong data linking TDP-43 pathology with *UNC13A* CE, and then demonstrate how the risk SNPs exacerbate this event, thus linking the CE abundance to patient disease outcome.

2. Related to the above point, I find the statement: "That genetic variation influencing the UNC13A CE inclusion can lead to changes in ALS/FTD susceptibility and progression strongly supports UNC13A downregulation to be one of the critical consequences of TDP-43 loss of function" somewhat premature. Based on the current data, UNC13A CE inclusion could be a modifier of disease risk without necessarily be a critical event for neuronal survival, like STMN2 was shown to be.

We have softened our claim; the sentence now reads:

Discussion: "That genetic variation influencing the *UNC13A* CE inclusion can lead to changes in ALS/FTD susceptibility and progression strongly supports *UNC13A* downregulation to be an important effector of TDP-43 loss-of-function-mediated neurotoxicity."

3i. The plot in Figure 4F indicates that TDP-43 binding in the UNC13A transcript is decreased in the presence of ALS-associated SNPs. However, it is unclear to me what kind of data and what kind of analysis was used to create this graph. The authors should clearly explain this in the results and methods parts of their manuscript.

We agree a clearer description is required: we have therefore added a description of the analysis in the methods section; the full code for this analysis is also available on our Github repo:

Methods: Heptamer analysis: Binding enrichment E-scores were downloaded from Ray et al., 2013. 7 nt sequences which overlapped with either the exonic or intronic SNPs were extracted using a sliding window approach. Using a custom R script (https://github.com/frattalab/unc13a_cryptic_splicing/), the average E-scores for each RBP were calculated for each set of 7-mers, and the RBPs were ranked by effect size of the

SNPs on average E-score.

3ii. In the same graph (Figure 4F), what are the other few RBPs that show significantly reduced binding (even more than TDP-43) in the same region? Are they potentially linked to the mechanism of CE inclusion in the presence of the SNPs? I think this is really important to clarify and maybe also experimentally test.

We agree it is possible that the binding of other RBPs is influenced by the SNPs, but the two RBPs that show a greater reduction in average E score are non-human (Sup-12 from *C. elegans* and PF10_0068 from *Plasmodium falciparum*). Because neither were sufficiently similar to a human homologue to assume that the binding preferences of the human homologue would be equivalent, we instead focused on the most decreased human RBP (TDP-43). We have stated in the main text that the two RBPs with a larger decrease are non-human.

3iii. The authors showed that one of the previously identified SNPs (rs12973192) leads to a significantly lower binding affinity for TDP-43 than the wild type sequence (Figure 4F-I), while, in contrast, there is enhanced TDP-43 binding on rs12608932 SNP (Figure 4I). I find this result somewhat puzzling and I think that the authors need to further explore it. For instance, how does rs12608932 SNP perform on the in vitro assay shown in Figure 4H? Based on the iCLIP results, I would expect that the affinity will be higher in the risk SNP sequence, i.e. the opposite of what is shown for rs12973192 in the current Figure 4H.

We have now performed the requested additional ITC experiments on the intronic SNP region. As expected due to the presence of a predicted TDP-43 binding motif and the enrichment of iCLIP signal in this region, we detected a strong binding affinity for both variants, with a potential slight increase in affinity for the risk variant. We emphasise, however, that while direct changes in binding affinity may help explain some of our results, these are highly artificial systems and in cells there may also be more complex mechanisms affecting binding, for example involving competition with other RBPs. The identification and validation of such competition could be an entire biochemistry/biophysics paper in itself, and we thus feel it is beyond the scope of the current study.

While it may appear surprising that increased binding might promote the cryptic exon, the reviewer will appreciate that splicing regulation is extremely complex due to the competition of different splicing regulators and the ability of TDP43 to act as both a splicing repressor and enhancer. We stress that although direct changes in binding affinity are an attractive mechanism to explore due to their relative simplicity, we do not rule out that additional, more complex mechanisms may be involved - we have clarified this in the manuscript.

Discussion: "Clarification of single versus additive effects of co-inherited SNPs regarding effects on CE inclusion, as well as contributions of other RBPs, will require future investigation."

The reviewer will have noted the differing reported binding affinities between Ma et al.'s revised manuscript and our own. For the reasons outlined below, we believe there is strong evidence that the regions harbouring the SNPs are TDP-43 binding sites:

- We used isothermal titration calorimetry, as it enables precise determination of binding constants.
- Our results are consistent with previously reported values for the binding of TDP-43 to UGNNUG-containing RNA.
- Differently to Ma et al., we used truncated TDP-43 without the prion-like domain, to reduce the risk that protein aggregation will interfere with K_d measurement.

- Differently from Ma et al., we used shorter RNAs (14 and 18 nt) in order to minimise RNA folding.

Additionally, we now report novel TDP-43 iCLIP experiments supporting that TDP-43 binds to this region when *UNC13A* is expressed endogenously (fig. S2B). We then tested whether TDP-43 binding to the CE SNP region is important for splicing by manipulating the sequence surrounding the SNP. Since TDP-43 binding to RNA is dependent on RNA motifs, we mutated two single nucleotides adjacent to the SNP that are essential components of the TDP-43 binding motif. Both mutations independently enhanced CE splicing at levels similar to the risk SNP, consistent with TDP-43 binding to this motif and playing a role in CE regulation. These results have been added to the text.

Results: “To explore whether these two SNPs directly influence TDP-43 binding, we analyzed a dataset of *in vitro* RNA heptamer/RBP binding enrichments. We examined the effect of the SNPs on relative RBP enrichment³⁰ by comparing healthy vs risk SNP-containing heptamers. Strikingly, when investigating which RBPs were most impacted in their RNA binding enrichment by the CE-risk SNP, TDP-43 had the third largest decrease of any RBP, with only two non-mammalian RBPs showing a larger decrease (Fig. 4G; fig. S11G), whilst the intronic SNP did not appear to strongly affect TDP-43 binding (fig. S11H,I). To verify that the CE SNP directly inhibited TDP-43 binding, we performed isothermal titration calorimetry using recombinant TDP-43 and 14-nt RNAs. We observed high-affinity binding and an increased Kd (lower binding affinity) for RNA containing the CE risk SNP. Similarly high affinity was observed from the intronic SNP region and the risk variant lead to a decrease in Kd (higher binding affinity) which approached significance (Two-sample t-test, $p = 0.052$). (Fig. 4H; fig. S12A-D; Data S4). Lastly, to test whether direct binding of TDP-43 to the region containing the CE SNP is critical for repressing the CE, we mutated the UGNNUG TDP-43 binding motif in this region, which led to significantly increased CE inclusion (Fig. 4I,J; fig. S12E). Together these data suggest that the risk SNPs modulate TDP-43 binding, in part via direct changes in binding affinity, exacerbating *UNC13A* CE inclusion.”

3iv. In the iCLIP experiment (Figure 4I), it is unclear to me how the authors normalized the levels of minigenes in the transfected cells to quantitatively compare TDP-43 binding on the different variants. This is necessary since I would expect some variability originating from transfection and which may significantly skew the conclusions of the iCLIP comparison.

We designed our experimental and analytical approach to reduce the impact of variable transfection efficiency. Experimentally, we normalised the amount of plasmid DNA by Nanodrop, and performed all transfections in parallel using identical reagents; additionally, each iCLIP sample was derived from two separate dishes of cells, reducing the impact of dish-to-dish variability in cell density or transfection efficiency. Analytically, we performed internal normalisation for each sample, calculating the number of reads at each position divided by the total number of reads aligning to the minigene. This internal normalisation means that a small difference in transfection efficiency, resulting in a global increase/decrease in reads aligning to the minigene for a given sample, will not be reported as a change in binding affinity. We have added the following to the methods section.

Methods: “1.25 μg of plasmid was used for each well, measured via Nanodrop (Thermo Fisher Scientific), combined with 2.5 μl of Lipofectamine 3000 and P3000 reagent diluted in 250 μl (2x 125 μl) of Opti-MEM I following the manufacturer protocol (Thermo Fisher Scientific).”

Methods: “All downstream analysis was performed using custom R scripts; to avoid biases due to differing transfection efficiencies, crosslink densities were normalised by the total number of minigene crosslinks for each sample. Raw data is available at E-MTAB-10297.”

4. Authors state that [...] although unlike the *STMN2* CE, the *UNC13A* CE induces NMD, it was detected at similar levels to *STMN2* CE in cortical regions, whilst *STMN2* CE was more abundant in the spinal cord (Figure 3). What is the relevance of this finding? Could it be neuronal subtype-specific?

The reviewer raises an interesting point. It is possible that the observed differences reflect the different baseline expression of *STMN2* and *UNC13A* in lower motor neurons vs cortical neurons, or the regional differences in NMD, as pointed out by Reviewer One. Another possibility is that *STMN2* plays a more important role in the spinal cord, whilst *UNC13A* has a more prominent effect in cortical neurons. Resolving these fascinating questions will likely uncover mechanistic roots of selective neuronal vulnerability in ALS/FTD spectrum disorders, perhaps revealing why certain patients develop ALS while others develop FTD (or co-occurrence of these disorders).

5. *rs12973192* and *rs12608932* SNPs are mainly associated with increased ALS risk (van Es et al., 2009; Nicolas et al., 2018) but also shown to contribute to FTD in sporadic ALS (Placek et al., 2019). However, CE inclusion seems to be more frequent in FTL-D-TDP than in ALS-TDP (Figure 3). How can the authors explain this result?

The reviewer raises a very interesting point, and indeed it is possible that *UNC13A* CE may have different relevance in FTD and ALS. As the reviewer alludes to, ALS and FTD are a continuum of related disorders, and the initial GWAS studies mentioned above did not rigorously assess ALS versus ALS/FTD in their clinical assessments. Newer studies, including the one mentioned above as well as by Tan et al 2020 (PMID: 32627229) that specifically assess for FTD co-morbidity, suggest that the risk-associated SNPs also lead to a higher incidence of ALS-FTD. It is nonetheless very difficult to draw conclusions from comparing different brain regions in different diseases, due to them having diverse cell populations and disease involvement, therefore affecting the detection of TDP-43 pathology and *UNC13A* CE. Specifically, in Figure 3, we have been able to compare ALS and FTD only using frontal and temporal cortex samples, the regions mostly affected in FTD. Involvement of these regions, as well as cognitive symptoms, can also occur in ALS, but it is understandable how these are more affected in FTD, and we believe this is the reason for FTD cases having more splicing changes in these regions.

6. I think that showing that *UNC13A* CE-containing transcript depends on nonsense-mediated decay for degradation is very important mechanistically (Figure 2F). What about the alternative splicing shown for *UNC13B*? Is this the same or a different mechanism of degradation?

Since the fsE in *UNC13B* leads to a PTC at the beginning of exon 11, NMD would be the most likely mechanism of degradation of this transcript. Therefore, we have replicated the NMD experiment on *UNC13B*. We have updated Fig. 2D (and added fig. S4 G,I), the Results section and the Methods.

Results: “To assess whether the degree of *UNC13A* CE expression was underestimated due to efficient transcript degradation, we investigated whether it promoted nonsense-mediated decay (NMD), as predicted by the presence of a novel premature termination codon (PTC). Knockdown of either the key NMD factor UPF1 or cycloheximide (CHX) treatment –which stalls translation and impairs NMD – increased *UNC13A* CE and *UNC13B* fsE, which also leads to a PTC at the beginning of exon 11, confirming they were both targeted by NMD (Fig 2D;fig. S4H,I). Conversely, CHX and UPF1 knockdown did not alter levels of the aberrant *STMN2* transcript, which was not predicted to undergo NMD (Fig. Fig 2D;fig. S4H,I).”

Methods: “The *UNC13B* experiment was subsequently performed, following the same method.”

a. In the sentence “Genome-wide association studies (GWASs) have repeatedly demonstrated a shared risk locus between ALS and FTD within the crucial synaptic gene *UNC13A*, although the mechanism underlying this association has remained elusive⁴”, reference 4 (van Es et al 2009 Nat Gen) states *UNC13A* locus is associated to ALS, but not FTD.

We have now added the following citations identifying *UNC13A* as an FTD GWAS risk locus:

- Pottier, C. et al. *Genome-wide analyses as part of the international FTLD-TDP whole-genome sequencing consortium reveals novel disease risk factors and increases support for immune dysfunction in FTLD. Acta Neuropathol. (Berl.) 137, 879–899 (2019).*
- Diekstra, F. P. et al. *C9orf72 and UNC13A are shared risk loci for ALS and FTD: a genome-wide meta-analysis. Ann. Neurol. 76, 120–133 (2014).*

b. In the sentence “ALS and FTD are pathologically defined by cytoplasmic aggregation and nuclear depletion of TAR DNA-binding protein 43 (TDP-43) in the vast majority (>97%) of ALS cases and in 45% of FTD cases (FTLD-TDP)⁵”, reference 5 (Neumann et al 2005 Science) shows cytoplasmic mislocalization and aggregation of TDP-43, but the incidence of TDP-43 proteinopathies in ALS and FTD remains unreferenced. We have now added the following citation listing the incidence of TDP-43 proteinopathies in ALS and FTD: Tan, R. H., Ke, Y. D., Ittner, L. M. & Halliday, G. M. *ALS/FTLD: experimental models and reality. Acta Neuropathol. 133, 177–196 (2017).*

c. The sentence “The RNA-seq data used from TDP-43 depleted SH-SY5Y and SK-N-DZ neuronal lines (Figure 1I-L)” is missing a reference in the main text.

In this case there is no reference for these samples as they are new to the study, we have clarified this in the text as well as in Table S2.

Results: “We also detected these splicing changes in RNA-seq data we generated from TDP-43 depleted SH-SY5Y and SK-N-DZ neuronal lines, and publicly available RNA-seq from iPSC-derived motoneurons (MNs)¹¹ and SK-N-DZ datasets²⁵ (fig. S1H-K, Table S2).”

d. In the legend of Figure 1, references to the latest ALS GWAS and the five TDP-43 knockdown datasets are missing

We have included the references to the ALS GWAS as well as the 2 previously published RNA-seq studies in the figure caption.

a. In how many samples/replicates was the RNA-seq on iPSC-derived cortical-like i3Neurons performed?

We have provided replicate information for all cell lines in Table S2.

b. How were the 179 CE identified or predicted?

The method for CE identification is described in the Methods. Briefly we used the splicing tool MAJIQ with cryptic splicing was defined as junctions with PSI < 5% in control samples, Δ PSI > 10%, and if the junction was unannotated in GENCODE v31. We have also clarified in the caption for Figure 1 as follows:

Results: “Fig. 1. TDP-43 depletion in neurons leads to altered splicing in synaptic genes *UNC13A* and *UNC13B*. (A) Differential splicing using MAJIQ”

c. It is unclear to me where “the multiple binding peaks both downstream and within the body of the *UNC13A* CE (Figure 1D)” are coming from. Are those from previously published CLIP data?

We have moved the citation from the end of the sentence to be make it more clear that these are binding sites from Tollervey et al. 2011.

d. The authors state that “TDP-43 KD significantly reduced *UNC13A* RNA abundance in the three cell types with the highest levels of cryptic splicing.” Which three cell types are they referring to?

We apologise for the confusion: it was not “Cell types”, but “experiments”. We have corrected this

sentence.

e. Unless I missed it, the cell type (HEK293T) in which iCLIP was performed is only mentioned in the methods and not in the main text/legend.

We have added this to the main text.

f. When the authors analyzed a dataset of *in vitro* RNA heptamer/RBP binding enrichments, what was the followed pipeline? Did they take the heptamers that cover the SNP position and compare the RBP binding of heptamers with the risk SNP to heptamers without (healthy SNP)?

This is precisely what we did, and we have clarified this in the main text as follows:

Results: “To explore whether these two SNPs directly influence TDP-43 binding, we analyzed a dataset of *in vitro* RNA heptamer/RBP binding enrichments. We examined the effect of the SNPs on relative RBP enrichment³⁰ by comparing healthy vs risk SNP-containing heptamers.”

g. In Figure 1B, authors say they used four controls and three TDP-43 depleted samples. However, in the methods sections, they claim it is three controls and four TDP-43 depleted samples.

We thank the reviewer for pointing this out. This was a typo and we have corrected in methods section to “control (n=4); TDP-43 knockdown (n=3)”.

h. In Figure 1, are the sashimi plots showing these two representative samples, or a merge of all replicates?

These are showing representative samples; we have clarified this in the figure legend.

i. In Figure 1H, which kind of correlation test was performed? Spearman’s? Information is missing in the corresponding legend.

We have removed this panel. However, as the reviewer correctly surmised from the use of “rho” rather than “r” this was a Spearman’s correlation.

j. In Figure 4B, there are no error bars shown in the plot. Would it be possible to show the actual data points?

4B shows counts of unique cDNA in barplot form to ease comparison between the number of reads with the risk versus healthy SNPs, and to allow quick visual comparison of the depth across the cryptic region on the multiple FTD samples sequenced. The height of the barplots represents the actual data points, and not a summary statistic.

k. In Figure 4I, where does the UGNNUG motif come from? Was it a prediction based on their own iCLIP data?

This is the core of the binding motif identified from NMR and iCLIP data (Lukavsky et al., 2013), also supported by more recent iCLIP-based studies (for example Hallegger et al., 2021) reflecting that, although TDP43 binds to UG repeats, the two bases positioned between RRM1 and 2 are not major determinants of binding affinity (hence “NN”).

i. Did they use the STAR alignments as input to the splicing analysis with MAJIQ?

Yes, we have clarified this in the methods:

Methods: “STAR aligned BAMs were used as input to MAJIQ (v2.1)³⁴ for differential splicing analysis using the GRCh38 reference genome.”

ii. Analysis of published iCLIP data is unfinished. The section ends with an unfinished sentence: "The pipeline...".

This was a typo and we have removed it.

iii. For the RNA-seq in i3Neurons, authors state that "one control replicate did not pass RNA quality controls and so was not submitted for sequencing." Does that mean they only had two controls instead of three, or three instead of four?

We had four controls and three knockdown replicates in the actual RNAseq experiment, after removing one knockdown replicate because of poor RNA quality. We have clarified this in these methods:

Methods: "control (n=4); TDP-43 knockdown (n=3). To isolate RNA, we used a Direct-zol RNA miniprep kit (Zymo Research Corporation, Cat. No. R2052), following manufacturer's instructions including the optional DNase step. Note: one knockdown replicate did not pass RNA quality controls and so was not submitted for sequencing, leaving a total of N=3 samples for this condition."

In the iCLIP analysis of the minigene-transfected cells, how did the authors identify the cross-links? And why did they not use the iCount pipeline as they did for the published iCLIP data?

Individual crosslink positions were defined at the first mapped position of each read, as is standard in the field. The iCount pipeline is designed for data with relatively low read depth, and therefore focuses on avoiding false positives. However, the TDP-43 binding region of our minigene had several orders of magnitude greater read depth than a typical intron in an iCLIP experiment (because the minigene was transgenically expressed using a strong promoter), meaning that the permutation method to call peaks in the iCount pipeline is not well suited to this data. To define peaks (for fig. S11F) we instead searched for positions with crosslinks more than 5x above the local average; this style approach will be an option in the new iMaps2 iCLIP pipeline, which is produced by the same group as the original iCount pipeline.

Response to Reviewer 4.

The statement of "UNC13A CE inclusion negatively correlated with TARDBP RNA levels" Figure 1H is questionable. First of all, the calculated p-value is not significant ($p=0.077$). More importantly, is this model suitable for the data points that were collected from multiple cell lines? Let's say if only iPSC MN data (6 data points) are used to run the regression, can the authors still conclude with a negative correlation?

We agree with the referee that combining data on different cell types is not suitable to address this point and have removed this panel. We now present data from i3Neurons with partial knockdown using CRISPRi, please refer to Fig 1I-J and S1D-G. We also present new data on the relation between UNC13A and TDP-43 protein levels and UNC13A CE levels obtained in SH-SH5Y cells where we were able to modulate TDP-43 knockdown. Please refer to Fig. 2C and fig. S4F,G.

Figure 1 (I, J, K, L) Can the authors add the statistical test?

We have included a Wilcoxon test comparing controls and TDP-43 knockdown in each experiment.

What can be concluded for UNC13A IR, UNC13B fsE and UNC13B IR in SH-SY5Y cell line (Figure 1J, K, L)? Some discussion could help.

These panels highlight the fact that TDP-43 loss induces also other splicing dysfunctions in the UNC13 genes. We now clarify through long-read RNA-seq, whether these events always co-occur, and find that this is not the case and they often appear independently (Fig. 1L; fig. S7). While comparing across different cell lines and different batches makes definite conclusions about the required levels of TDP-43 reduction for UNC13 cryptic exon to arise, the SH-SY5Y cell line had least efficient TDP-43 knockdown - this could be a potential reason for the relatively low expression of the *UNC13A/B* TDP-43 regulated splicing events we describe. We have included a supplementary figure illustrating this, as well as included the following into the main text:

Results: “We note that the expression of these events was lowest in the SH-SY5Y experiment, which also had the weakest TDP-43 KD (fig. S1L).”

Is there any evidence from LC/MS to show UNC13A CE at protein level?

We have found no evidence for cryptic peptides in our LC/MS data for any cryptic peptides originating from UNC13A. As we demonstrate in the paper, this transcript undergoes NMD, which degrades cytoplasmic transcripts that could undergo translation.

Figure 3 (B, C) Can the authors add the statistical test?

We have added a Wilcoxon test comparing means to B, C and updated the figure legend to state this.

Figure 3D Additional barplot of a quantitative measure of probe signals with the statistical test is recommended to show the significance.

We have now extended our ISH analysis, and added a quantitative measure of probe signal, which shows a significant difference between FTD-TDP and both controls and FTD-non-TDP. An extra panel has been added to Figure 3, and results description amended:

Results: “We detected red foci in cortical neurons at a significantly higher frequency in FTLTDP cases relative to both neurologically normal controls (Kruskal-Wallis test, $p = 0.021$) and non-TDP (tau) FTLTDP cases ($p = 0.010$) (Fig. 3D).”

Additionally, we have added ISH analysis from the i3Neurons:

Results: “We validated the *UNC13A* CE in i³Neurons by *in-situ* hybridization, which showed a primarily nuclear localisation and predominantly occurred in TDP-43 knockdown neurons (Fig. 1H; fig. S1C).”

Figure 4E May the authors switch to two-way ANOVA test (two categorical variables: shTDP43 treatment, minigene constructs) as the statistical test?

This figure has now been moved to fig. S11C. We have now switched to a two-way ANOVA.

Figure 2F, 4H: Please document the statistical test used.

The test used is a one-sample t-test. We updated the figure legend (please note the revised manuscript is reported in Fig. 2D)

Results: “Transcript expression upon CHX treatment suggests UNC13A and UNC13B, but not STMN2, are sensitive to nonsense-mediated decay. HNRNPL (heterogeneous nuclear ribonucleoprotein L) is a positive control. Grey bar indicated UNC13B was performed in separate experiment. One-sample t-test.”

Figure 4H: the test used is a two-sample t-test. We updated the figure legend according to the reviewer's suggestion:

Results: "Binding affinities between TDP-43 and 14-nt RNA containing the CE (n = 4) or intronic (n = 3) healthy or risk sequences measured by ITC; two-sample t-test."

In UNC13B, TDP-43 KD ...increased intron retention (IR) between exon 21 and 22 (Figure S1A). Some quantitative comparisons could help. Same for Figure S1B

We now indicate that the quantitative comparisons on IR for *UNC13A* (Figure S1A) and *UNC13B* (Figure S1B) are shown in fig. S1, K respectively. We have also included the quantification in the text of the supplementary figure caption for clarity.

Results: "(A,B) RNA-seq traces from IGV⁶⁹ of representative samples from control (top) and TARDBP KD (bottom) in i3Neurons showing intron retention in *UNC13A* (A) (mean 4.50 ± 1.50 increased IR in KD) and *UNC13B* (mean 1.86 ± 0.63 increased IR in KD)(B), overlaid with published TDP-43 iCLIP peaks²⁶"

Consistency of Figure 4A and Figure S4 (A,B). Any reason to include Kruskal-Wallis test results in Figure S4 (A,B), but not in Figure 4A (labeled with "Wilcoxon-Test" instead)?

We have changed the inconsistent reporting and removed the Kruskal-Wallis test results from the supplementary figure. Please note that in the revised manuscript S4A-B have been replaced with S11A (due to near-perfect redundancy, S4B is no longer shown).

Response to Reviewer 5.

1. *Fig 3B, the significant differences between control and ALS-TDP is striking, according to Fig S3A all motor cortex seem to express a similar level of UNC13A, however, many ALS-TDP patients do not show evidence of CE incorporation? Is the phenotype between those CE positive and CE negative patients' coloured differently?*

We believe *UNC13A* CE is not always detected in bulk CNS RNA-seq due to a combination of technical and biological reasons. Importantly, the *UNC13A* CE is targeted for degradation by nonsense mediated decay (as we show in Fig. 2D and the new fig. S4J,K), and occurs only in neurons with TDP-43 pathology, which reflect <1% of cells present in some of the tissues processed for bulk RNA-seq. These factors, in combination with the fact that RNA from post mortem CNS is often degraded, impact on overall coverage and detection of these splicing events. In support of this, when neurons with TDP-43 pathology are selectively targeted for RNA-seq (as shown in Fig. 3A), *UNC13A* CE is detected in all TDP-ALS/FTD samples. In Fig. 3B the colors refer to if the samples would be expected to show TDP-43 proteinopathy (orange ALS/FTLD-TDP) and those that would not (yellow - controls and ALS/FTLD-non TDP). Samples without the CE are the dots that lie along the x-axis at zero.

Are the samples showing no CE inclusion from patients without the UNC13A risk alleles?

According to our model, which we have now reworded in the discussion to improve clarity, *UNC13A* CE occurs when there is TDP-43 depletion in individuals with or without the risk SNPs. The risk SNPs enhance this process, but are not necessary for it to occur, as illustrated by our analyses in Fig. 4A and figS11A. Technical difficulties are likely the primary reason why the CE is not detected in some samples, relating to

the challenges associated with performing RNA-seq on *post-mortem* CNS samples (as outlined above), and exacerbated by the fact that only a subset of neurons will feature TDP-43 mislocalization, even in advanced-stage patients, thus diluting CE levels in bulk tissue. However, as the reviewer suggests, the lower level of the *UNC13A* CE in patients without the risk SNPs may also contribute to this: below, we show there is a non-significant trend for patients with both risk alleles to have higher detection rates of *UNC13A*.

1. Page 5 paragraph 1, the quantitative traits look very strong, but *rs12608932* has a population AF of 30% according to dbSNP. How can we reconcile such a high VAF with the rare ALS-TDP phenotype?

The *UNC13A* SNPs act as disease modifiers, rather than being directly causative of ALS or FTLD. The SNPs themselves will have no or little effect when there is enough TDP-43 present in the nucleus to prevent CE splicing, as would be found in the general population. It is only when TDP-43 nuclear depletion has begun, as in disease states of ALS/FTLD-TDP, that the SNPs effect of altering TDP-43 binding becomes relevant to inducing the *UNC13A* CE. We have changed the wording in the discussion to make this point clear:

Discussion: “In this model, when nuclear TDP-43 levels are normal in healthy individuals, TDP-43 efficiently binds to *UNC13A* pre-mRNA, even in the presence of risk SNPs, thus preventing CE splicing.”

Are there any rare variants found in any of the samples the authors analyzed?

Some of these samples contain disease-causing mutations in *C9orf72*, *SOD1*, *OPTN*, *MATR3*, *ANG*, *TBK1*, *MAPT*, or *FUS*, and this genotype information is available on our GitHub repo (https://github.com/frattalab/unc13a_cryptic_splicing).

2. Fig S1D Did the authors sequencing the bands with Sanger to confirm the splice junction?

We have confirmed the splice junctions via Sanger sequencing for the short and long form of the *UNC13A* cryptic exons in both SH-SY5Y and SK-N-DZ cell lines and included this in fig. S2A. We have added the following to the methods.

Methods: “For Sanger sequencing, *UNC13A* CE was amplified with exon 19 forward primer 5'-GACATCAAATCCCGCGTGAA-3' and exon 22 reverse primer 5'-CATTGATGTTGGCGAGCAGG-3'. Amplicons were resolved by agarose gel and the bands

corresponding to the short and long form of the cryptic exon were excised and purified (NEB T1030L). The *UNC13A* exon 22 reverse primer 5'-ATACTTGGAGGAGAGGCAGG-3' was used for sequencing reactions.”

3. Fig. 3E there are many points at the left bottom corner. The number of samples in total should be indicated in the main text?

We changed this image to only include the same samples plotted in figure 4A (cortical samples, with at least 30 junction reads, and both *STMN2* and *UNC13A* CE detected) We have updated the figure and the text accordingly with the N

Results: “As expected, across the NYGC ALS Consortium samples we observed a significant positive correlation between the level of *STMN2* CE PSI and *UNC13A* CE PSI ($\rho = 0.56$, $p = 2.9e-7$, $N = 72$ cortical samples) (Fig. 3E).”

1. What was the rationale to choose *UNC13* family proteins over the many other genes mis-spliced in FTLTDP iNeurons? Was there a joint probability or prior probability used to intersect genes with GWAS? It would appear from Fig. 1b that there are many other genes that would fulfill the same criteria. We focused initially on the *UNC13* family proteins because of the wealth of genetic associations of *UNC13A* with ALS and FTD and because one of the disease SNPs lay directly inside the TDP-43 regulated cryptic exon. We have additionally included Table S1 showing the gene expression and cryptic splicing

status of ALS and FTD associated genes in the i³Neurons, showing that only *UNC13A* contains a cryptic splice event. We have added the following explanation in the manuscript:

Results: “We intersected splicing, expression, ALS GWAS¹⁷ risk genes, and diagnostic panel genes for ALS/FTD¹⁸. Of the 179 CE-harboring genes, only the synaptic gene *UNC13A* was also an ALS/FTD risk gene (Table S1; Fig. 1C,D).”

2. Check ref 14 and 49 format is different

Thank you, we have fixed this.

3. Fig. 2F indicate what test was used in the legend

The test used was a one-sample t-test. We updated the legend of the figure according to the reviewer’s suggestion. Note that the equivalent figure is now Figure 2D.

Results: “Transcript expression upon CHX treatment suggests *UNC13A* and *UNC13B*, but not *STMN2*, are sensitive to nonsense-mediated decay. *HNRNPL* (heterogeneous nuclear ribonucleoprotein L) is a positive control. Grey bar indicated *UNC13B* was performed in separate experiment. One-sample t-test.”

4. Fig. 3C makes the case that *STMN2* and *UNC13A* cryptic PSI are cross correlated. Is the genomic architecture and sequence between the two genes?

STMN2 and *UNC13A* are on separate chromosomes (chr8 and chr19 respectively) with no clear evolutionary relationship. However, they both contain UG-rich regions and are thus predicted to be bound by TDP-43.

5. Fig. 3c also indicate the n numbers and p-value?

We have included the number of tissue samples in each condition on Fig. 3C and included a Wilcoxon test to compare the levels *STMN2* and *UNC13A* CE PSI. This has been stated in the figure caption.

6. Figs 3 and 4 only focused on *UNC13A* CE, did the authors evaluate other possible events in the *UNC13*

family as mentioned in Fig 2 and the discussion?

We thank the reviewer for highlighting this. To parallel the *UNC13A* data presented in Fig. 3, we have now included the analysis of the *UNC13B* fsE in the NYGC dataset and added a novel Fig. S8 and S9. Unfortunately, in both control and disease post mortem brains a short *UNC13B* isoform that includes the fsE is constitutively expressed. The presence of a basal level of *UNC13B* fsE does not allow us to assess whether, in the presence of TDP-43 pathology, there is an increase of altered splicing. We explain this in the result section with new text:

Results: “We next assessed the expression of the *UNC13B* fsE across the NYGC dataset. We did not detect a specific increase in pathological ALS-TDP and FTLTDP tissues. However, the presence throughout control and ALS/FTD brains of a shorter isoform, which was absent in our *in vitro* experiments and includes the fsE, may be masking underlying changes (fig. S8A-C). We also evaluated the both *UNC13A* and *UNC13B* IR events from bulk RNA-seq. Similar to the neuronal cell lines, both IR events were also in control brains, making it difficult to determine whether TDP-43 pathology increased IR in patients (fig. S9A,B)”

We do not carry out analyses on *UNC13B* in Fig. 4, as the work reported there aims to understand how the risk SNPs impact on *UNC13A* CE splicing, whilst there are no known risk SNPs in *UNC13B*.

7. Fig. 4 there is a tetranucleotide short tandem repeat (STR) polymorphism that is part of the *UNC13B* risk haplotype. The authors have shown that the 2R minigene is sufficient to mediate altered TDP-43 binding and CE incorporation, but how was the STR accounted for?

The reviewer refers to rs56041637 which leads to GAUG repeats in *UNC13A* pre-mRNA. This STR is within the TDP-43 binding region and could indeed impact on TDP-43 binding. We have now experimentally tested whether the number of repeats impact on *UNC13A* CE splicing and found that the effect of the STR on CE inclusion is minimal compared to the two SNPs. We now report these novel results in Fig. 4D, E and also illustrate the location of rs56041637 in Fig. 4C. We have added a description of this in the Results section:

Results: “To specifically examine whether the CE SNP, intronic SNP or short tandem repeat expansion rs56041637 – which is in linkage disequilibrium with the two SNPs²⁹ – are responsible for promoting the CE inclusion, we generated minigene variants featuring different combinations of the three aforementioned genomic variants (Fig. 4C; fig. S11D). Via quantitative analysis of RT-PCR products, we found that both the CE SNP and, to a lesser extent, the intronic SNP independently promoted CE inclusion, with the greatest overall levels detected for the 2R minigene (Fig. 4D,E).”

Reviewer Reports on the First Revision:

Referee #1

The authors have added a considerable amount of additional material, including new experimental work, as well as new *in silico* analyses which both increase my confidence in the accuracy of the claims and develop it further. In particular, I would highlight the additional long read sequencing data, and the use of variable TDP43 knockdowns to clarify the relationship between TDP43, *UNC13A* and *UNC13B* mRNA and protein levels. Overall, I think this is an excellent piece of work, which will generate a huge amount of interest.

Referee #2

I congratulate the authors for successfully revising the manuscript and for providing a convincing and clear demonstration of a novel mechanism in TDP43 pathology. My concerns have been addressed and I recommend the paper for publication in Nature.

Referee #3

The revised manuscript by Brown et al is significantly improved and addresses most of the important points raised by all five referees. I congratulate the authors for the thorough rebuttal and the additional data that they present clearly in their revised manuscript. They clarify all the points that have been ambiguous in the earlier version. In particular, they added new experimental data that convincingly demonstrates that TDP-43 binding affinity on the risk UNC13A alleles is altered. Moreover, the authors provide new data supporting that the CE-containing allele is degraded via NMD. I think the work is truly exciting and thorough and should be published in Nature.

Without attempting to significantly delay publication, I would like to insist on the only important point that they have not addressed, which is the functional consequence of UNC13A loss for adult neurons. This has been my main concern and I see that some of the other referees raised the same point. The authors argue that since there are several other synaptic targets of TDP-43 with possibly similar effects, it might be difficult to detect an UNC13A-specific effect. I would counterargue that exactly because TDP-43 has so many important targets in neurons, it is critical to test this. In my view, depending on the impact of UNC13A loss in the context of TDP-43 pathology/misregulation, this target might be anywhere between "the" key effector of neurotoxicity to an inert bystander (in reality it is probably somewhere in-between). I understand that the full mechanistic elucidation is complex and may be beyond the scope of the current study and I do not propose to significantly delay the publication of this exciting finding. However, I think that an initial test of the functional consequences of UNC13A loss for adult neurons and/or rescue experiments by combining TDP-43 loss with co-expression of a TDP-43-insensitive UNC13A construct may be more revealing than the authors predict. I agree that the genetic implications of this gene as an ALS risk argue for an important role of UNC13A in neurodegeneration. I am therefore more optimistic that this direct link would be demonstratable experimentally.

Referee #4

All my concerns related to statistics have been addressed. I don't have further comments.

Referee #5

The authors have adequately addressed our concerns with a detailed response and changes to the manuscript.

nature portfolio